# Fast analysis and engineering of protein function by microbe-independent deep assembly and screening

Yan Wu[1,10,11], Pengli Wang [ID][2,11], Lan Xiang Liu[3], Daesun Song [ID][3], Qin Qin[4], Chao Gao [ID][5], Matt Hageman[5], Thomas A Kirkland [ID][6], Yichi Su [ID][4,7] & Michael Z Lin [ID][1,3,8,9][✉]

## Abstract

Traditional methods for engineering and sequence-fitness analysis of proteins in mammalian cells are limited by the time, cost, and labor associated with plasmid cloning and preparation. Here we present Microbe-Independent Deep Assembly and Screening (MIDAS), a deterministic platform for rapid protein variant expression and characterization in mammalian cells that bypasses microbial cloning by directly transfecting PCR-assembled genes. MIDAS enables high-quality sequence-fitness assessment of arbitrary mutational spaces, including truly deep saturation mutagenesis and combinatorial variant assembly, requiring less than one workday from initial PCR to cell transfection. Using MIDAS, we engineer a high-performance acetylcholine neurotransmitter bioluminescent indicator (ACh-NeuBI), achieving stepwise improvements in responsivity through linker engineering, single-site, and multi-site mutagenesis. We also apply MIDAS to engineer improved NanoLuc luciferase variants for multiple substrates, and to characterize the structural basis of mutational tolerance and substrate specificity. Thus, MIDAS is a versatile method for rapid plasmid-free protein engineering and sequence-fitness analysis in mammalian cells, offering a practical alternative to cloning-based approaches for many protein optimization and characterization tasks.

**Subject Category** Biotechnology & Synthetic Biology

## Introduction

A major goal in biotechnology and biomedical research is the engineering of proteins with enhanced or novel functions, especially those with industrial or medical applications. While machine-learning algorithms have made large strides toward designing primary sequences to create desired three-dimensional folds or protein–protein interfaces, current machine-learning algorithms do not reliably identify the most optimal protein sequence or small sets of near-optimal sequences. Rather, they typically generate hundreds of candidates, which then need to be constructed and screened. Thus, even computationally guided protein engineering still relies extensively on empirical experimentation. In addition, machine-learning approaches lack training for optimizing the multistep conformational changes required for enzyme catalysis or analyte-induced biosensor output changes. Correspondingly, additional data on sequence-fitness (or structure-function) relationships in proteins will be needed to train computational models to better predict mutational effects (Yang et al, 2024). Thus, methods to accelerate the generation of protein variants and characterizing their function would greatly facilitate protein engineering, either directly through faster screening or indirectly through generating data on protein sequence-fitness relationships.

When the intended uses for improved proteins are in mammalian cells, rigorous measurement of function in mammalian cells is critical. Some mutations that improve performance in bacteria may prove ineffective in mammalian systems (Tian et al, 2009), and binding affinities of proteins to small molecules can differ substantially between bacterial and mammalian settings (Marvin et al, 2013). Multiple methods exist to screen for improved protein function directly in mammalian cells (Hendel & Shoulders, 2021), but still involve time-consuming preparation steps in bacteria. Fluorescence-activated cell sorting (FACS) is commonly used to identify and isolate cells expressing improved mutations (Di Roberto et al, 2020; Iwamoto et al, 2010). If deep sequencing of FACS-enriched cell mixtures is performed to identify a pool of favorable mutations, this method is termed deep mutational scanning (DMS), where depth refers to the level of sequencing (Maes et al, 2023). With FACS, multiple rounds of enrichment and/or large secondary screens (Babakhanova et al, 2022; Limsakul et al, 2018; Mason et al, 2018), or repeated experiments in the case of DMS (Maes et al, 2023), are often required to identify improvements with high confidence. For the most reliable functional selections, mammalian cells expressing mutant libraries can be seeded as single cells in multiwell plates and then grown and assayed as populations, achieving medium throughput ($10^4$ per day) (Sullivan et al, 2017; Zhang et al, 2023). However, the above

[1]Department of Bioengineering, Stanford University, Stanford, CA, USA. [2]Department of Chemical Engineering, Stanford University, Stanford, CA, USA. [3]Department of Neurobiology, Stanford University, Stanford, CA, USA. [4]Institute for Translational Brain Research, State Key Laboratory of Medical Neurobiology, MOE Frontiers Center for Brain Science, Fudan University, Shanghai, China. [5]Promega Corporation, San Luis Obispo, CA, USA. [6]Promega Corporation, Madison, WI, USA. [7]Department of Nuclear Medicine, Zhongshan Hospital, Shanghai, China. [8]Department of Pediatrics, Stanford University, Stanford, CA, USA. [9]Department of Chemical and Systems Biology, Stanford University, Stanford, CA, USA. [10]Present address: Department of Antibody Engineering, Genentech Inc., South San Francisco, CA, USA. [11]These authors contributed equally: Yan Wu, Pengli Wang. [✉]E-mail: mzlin@stanford.edu

methods still utilize transduction of lentiviral libraries or recombinase-mediated integration of plasmid libraries (Matreyek et al, 2017), which require weeks of prior effort to construct in bacteria and purify. Continual evolution of proteins by in cellulo mutation of viruses and the selective propagation of beneficial mutants can be performed to avoid plasmid library construction, but this strategy is limited to protein functions that can be linked to the transcription of viral replication genes, which itself requires weeks of cloning and testing of candidate constructs to achieve (Berman et al, 2018; English et al, 2019). If libraries are constructed and screened in bacteria or yeast, validation of performance in mammalian cells becomes the time-consuming step, involving isolation of expression plasmids, plasmid preparation, transfection, and reassessment of protein function.

Another challenge in protein engineering is that random mutagenesis by error-prone polymerase chain reaction (PCR), the standard method of generating diversity for DMS and other selection schemes, lacks mutagenesis depth in practice. Random mutagenesis can identify sites where mutations improve protein function without any a priori knowledge, but is incomplete in coverage at each codon. In one study, a mutagenesis rate of 0.5% maintained functionality in 6.7% of sequences constructed by error-prone PCR (Daugherty et al, 2000). At this rate, the frequency of any mutated codon receiving a second mutation while the protein remains functional is $0.005 \times 0.067$, or ~1 in 3000. A mutagenesis rate of 3% maintained functionality in 0.17% of sequences, yielding a recoverable second-mutation rate per codon of $0.03 \times 0.0017$, or ~1 in 20,000. Thus, increasing mutagenesis rates actually makes the recovery of multiple mutations per codon even rarer.

If library construction and screening could be performed entirely in mammalian cells without a microbial step, then assessing protein variants for mammalian performance could be greatly accelerated. Here, we report the development of Microbe-Independent Deep Assembly and Screening (MIDAS) for protein variant characterization in mammalian cells. MIDAS integrates three steps in rapid succession: multiwell PCR to construct libraries of all possible variants at targeted sites, direct transfection of linear PCR products into mammalian cells, and cell-based phenotypic screening. We demonstrate the ability of MIDAS to improve the performance of an acetylcholine neurotransmitter bioluminescence indicator (ACh-NeuBI) 29-fold by rapidly performing multiple engineering tasks: combinatorial linker length screening, linker composition optimization, protein folding tuning, assessment of computationally proposed mutations, and multi-location combinatorial saturation mutagenesis. Improvements identified by MIDAS in mammalian cells in vitro corresponded to improved contrast in detecting acetylcholine (ACh) noninvasively in vivo. In addition, we adapted the single-template MIDAS method to rapidly improve enzyme function by deep mutagenesis of all active-site positions, generating NanoLuc luciferase variants with improved light production from three different substrates.

## Results

### General strategies for microbe-independent deep assembly and screening (MIDAS)

To bypass the labor-intensive and time-consuming steps of plasmid cloning in bacteria, we hypothesized that we could accelerate the

expression and characterization of protein variants in mammalian cells by constructing entire genes with the desired modifications as linear PCR products in multiwell blocks and directly transfecting them into mammalian cells in matching multiwell plates. Performance of variants could then be directly assessed in their desired cell type and subcellular location, and improved variants identified (Fig. 1A). This procedure would save time and costs compared to traditional cloning methods by bypassing the time-consuming and expensive cloning steps of ligation or recombineering, bacterial transformation, bacterial growth, plasmid purification, and sequence verification. We refer to this general approach as microbe-independent deep assembly and screening (MIDAS).

Previously, we had used PCR gene assembly and transfection to improve ASAP-family voltage sensors (Evans et al, 2023), but implementation was limited to substitutions at one location in a chimeric protein. Specifically, in the previous protocol, two primary PCRs were used to synthesize left and right gene fragments templated from different plasmids, with the left one containing a promoter and an upstream portion of the coding sequence, and the right one encoding the remaining portion of the coding sequence and the polyadenylation signal. The forward primer of the right-side PCR (F2) both overlapped with the left PCR amplicon and introduced a desired mutation, and 384-well arrays of right-side PCRs could be performed with arrays of F2 primers to generate hundreds of mutations in the protein in parallel. An array of secondary PCRs is then performed to amplify the complete gene by overlap extension, using gel-purified left PCR product (which contains no mutations) and a small amount of the completed right-side PCRs without purification as templates. Each PCR product encodes a defined variant and is transfected into a separate well, allowing variant identity to be tracked by well position, maintaining association between sequence and function throughout the screen. We now refer to this method as polytemplated monofocal MIDAS, or MIDAS-PM (Fig. 1B).

We aimed to adapt the concept of PCR gene synthesis and transfection so that it can be broadly applied to a wide array of protein engineering and characterization tasks. First, we sought to generate and test combinations of variations at multiple locations in a protein. We postulated that three primary overlapping PCRs can be performed with primers introducing variations near the overlap locations, and these products can be assembled into a complete gene by a secondary PCR reaction. In this envisioned protocol, the left and right primary PCRs would use different plasmid templates. We refer to this method as polytemplated polyfocal MIDAS, or MIDAS-PP (Fig. 1C). Incidentally, MIDAS-PM and MIDAS-PP can also be used to construct and express a series of protein chimeras. Specifically, coding sequences for different protein functional domains could be amplified in a set of primary PCRs and then fused to the coding sequence of the protein of interest in the secondary PCRs.

Second, we desired to generate and test protein variants beginning from a single plasmid template, which has not been demonstrated previously. The above protocols require the protein's coding sequence to be present in multiple expression plasmids with different backbone sequences. Small amounts of the mutagenic right-side primary PCRs are used to supply the right-side mutated amplicon as template in the secondary PCR, and we intend that they will be added without purification to avoid time-consuming gel electrophoresis and elution of hundreds of PCRs, but as a result,

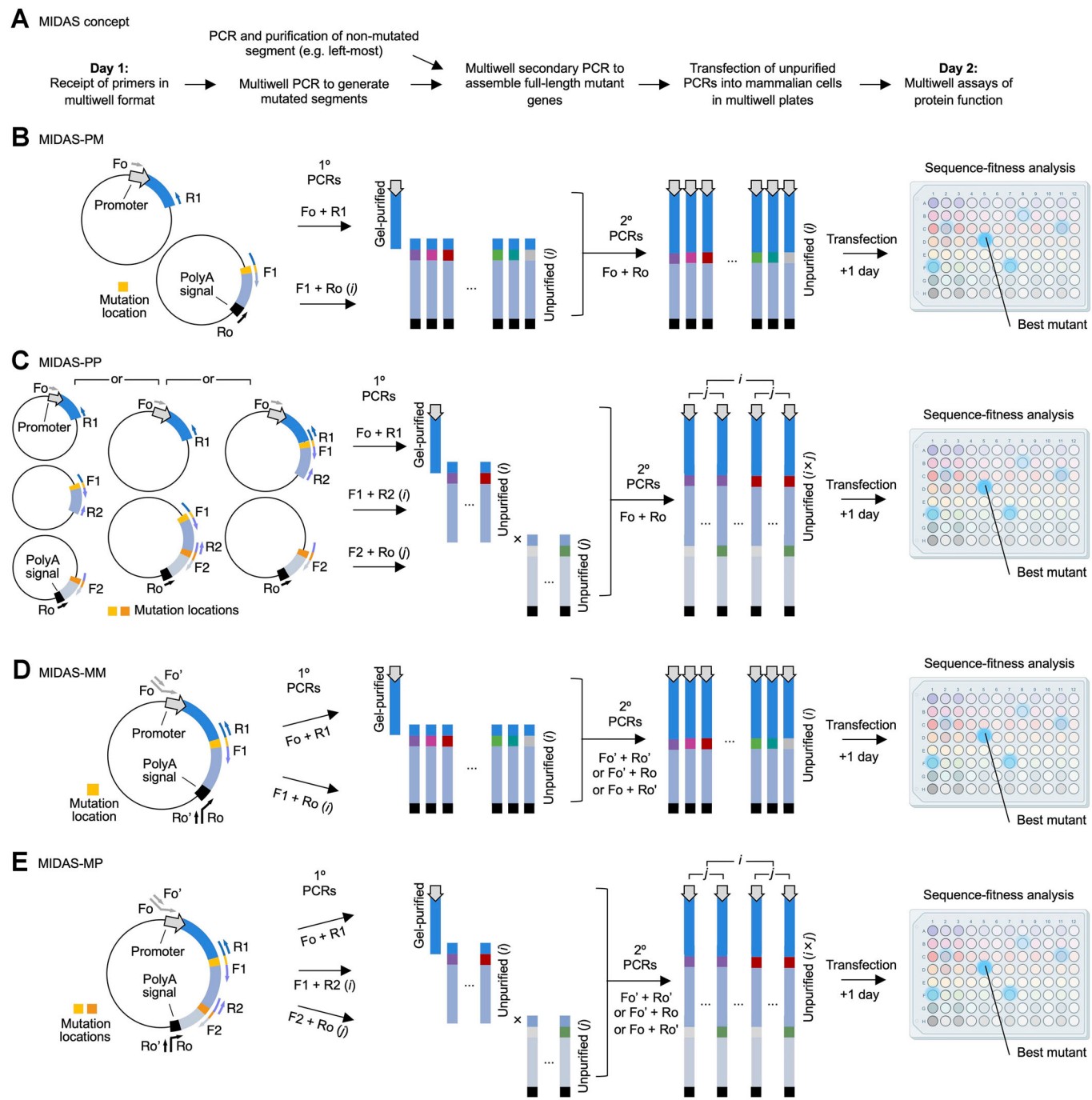

**Figure 1.  Theme and variations on microbe-independent deep assembly and screening (MIDAS).**

(**A**) General MIDAS concept allows testing of protein variant function in high-quality multiwell assays the day after primer receipt without intervening cloning steps. (**B**) Polytemplated monofocal MIDAS (MIDAS-PM) uses multiple plasmids as inputs and outputs arrays of mutants at one location in the protein. (**C**) Polytemplated polyfocal MIDAS (MIDAS-PP) uses multiple plasmids as inputs and outputs arrays of mutants at multiple locations in the protein. (**D**) Monotemplated monofocal MIDAS (MIDAS-MM) uses one plasmid as input and output arrays of mutants at one location in the protein. (**E**) Monotemplated polyfocal MIDAS (MIDAS-MP) uses one plasmid as input and output arrays of mutants at multiple locations in the protein.

the secondary PCRs also will contain residual amounts of the plasmid template from the primary PCRs. When different template plasmids are used for the left and right PCRs, primers for the secondary PCR can be designed to avoid amplifying the original plasmid templates. However, requiring subcloning of mutagenesis

targets into multiple plasmids as a precondition for MIDAS would run contrary to the goal of avoiding microbial steps to the greatest possible extent, and would be especially cumbersome for researchers improving a large number of proteins in parallel. To allow generation of only desired mutations by PCR assembly when the

same template plasmid is used for both the left- and right-most PCRs, we hypothesize that we can add unique DNA sequences to the 5' ends of the outermost forward and/or reverse primers in the primary PCRs. Primers that hybridize only to these specific sequences can then be used for the secondary PCRs (Fig. 1D). We termed this modified protocol as monotemplated monofocal MIDAS, or MIDAS-MM. By increasing the number of primary PCR segments, monotemplated polyfocal MIDAS (MIDAS-MP) is also possible (Fig. 1E).

When performing polyfocal MIDAS-PP or MIDAS-MP, mutations can be introduced at only one target per primary PCR amplicon and the primary PCR products mixed combinatorially to serve as templates for secondary PCRs to generate all possible permutations (Fig. 1C, E). Alternatively, mutations can be introduced combinatorially at both ends of one primary PCR amplicon; this necessitates setting up and performing more primary PCRs but allows one-to-one transfer of primary PCR material to serve as templates in secondary PCRs for simple completion of the full-length genes (Fig. EV1). The latter approach requires more parallelization earlier but reduces complexity later, and thus may be more suitable for robotic workflows.

## Design of a prototype ACh bioluminescent indicator

Examples of useful synthetic proteins that require comprehensive engineering at multiple elements are optical biosensors of biochemical activity, such as fluorescent and bioluminescent biosensors. We chose to create a bioluminescent small-molecule indicator based on a periplasmic binding protein (PBP) as a test case for synthetic protein optimization by MIDAS. Bacterial PBPs are bi-lobed proteins that bind to a variety of small molecules and have been used as a template for fluorescent indicators of glutamate, acetylcholine, serotonin, and GABA (Marvin et al, 2013; Borden et al, 2020; Marvin et al, 2019; Unger et al, 2020). In these fluorescent indicators, a circularly permuted GFP is inserted between the N-terminal (NT) and C-terminal (CT) lobes. Conformational changes in the PBP upon ligand binding are transduced to GFP side chain conformational changes to induce chromophore deprotonation. So far, however, despite the general utility of bioluminescence for noninvasive imaging in the body (Oh et al, 2019; Wu et al, 2023; Wu et al, 2025), there is no bioluminescent indicator for small-molecule analytes based on PBPs.

Given the importance of ACh in physiology and its dysregulation in disease, we chose to create an ACh bioluminescent indicator. In vertebrates, ACh serves as the primary neurotransmitter at neuromuscular junctions and in the parasympathetic and enteric nervous systems (Dunant and Gisiger, 2017). In the mammalian central nervous system, ACh functions as a neuromodulator involved in arousal, attention, learning, and memory (Hasselmo, 2006; Himmelheber et al, 2000). Cholinergic neurons, originating from the nucleus basalis, project throughout the cortex, with the highest innervation densities in the hippocampus. Reduced cholinergic activity occurs early in several neurodegenerative disorders, including Alzheimer's disease, Lewy body dementia, and Parkinson's disease (Bales et al, 2006; Bohnen et al, 2015; Kanel et al, 2021; Mesulam, 2004). In Parkinson's disease, dysautonomia due to degeneration of cholinergic parasympathetic neurons in the vagus nerve is also commonly observed (Bohnen et al, 2022). While

fluorescent indicators for ACh have been developed (Borden et al, 2020; Jing et al, 2020), their applications in vivo necessitate the implantation of optical elements such as fibers, windows, or lenses. This is feasible in the brain due to the essentially non-motile nature of the brain and surrounding skull, but implantation of optical elements is less compatible with visualizing tissues or organs in other parts of the body. In small animal models, a bioluminescent indicator would allow noninvasive, real-time monitoring of ACh release in locations where optical elements cannot be fixed while using affordable, readily available equipment.

We hypothesized that we could design a bioluminescent indicator of ACh using NanoLuc and a mutant OpuBC domain from *Thermoanaerobacter sp. X513* was previously engineered for ACh specificity over choline (Borden et al, 2020). A fluorescent sensor of ACh had been constructed by inserting circularly permuted GFP into OpuBC after position 106. We screened insertions at the same location of NanoLuc domains circularly permutated at position 65, 132, or 155, with the last one using either original NanoLuc or SmBiT and LgBiT sequences (Dixon et al, 2016) (Fig. EV2A). The variant with inserted SmBiT and LgBiT fragments showed the largest luminescence induction on HEK293 cells treated with 100 μM ACh treatment (Fig. EV2B). This result was not surprising, as assembly of LgBiT and SmBiT can be robustly modulated by intramolecular conformational changes (Wu et al, 2023; Wu et al, 2025), but the degree of luminescence change was small at only 22%. We designated this construct as Acetylcholine Neurotransmitter Bioluminescent Indicator 0.1 (ACh-NeuBI0.1) and selected it for further optimization (Fig. 2A).

## Linker optimization in mammalian cells by MIDAS

We first sought to simultaneously optimize the lengths of the linkers between OpuBC (30–106) and SmBiT (linker 1) and between LgBiT and OpuBC (107–305) (linker 2) using MIDAS-MP. We first tested if PCR products encoding ACh-NeuBI0.1 transfected into mammalian cells result in protein expression. We performed primary and secondary PCRs to amplify a gene encoding surface-anchored ACh-NeuBI0.1 fused to the PDGFR transmembrane domain, including promoter and polyadenylation signal, and transfected a subsample of the unpurified PCR into HEK293 cells. We then compared ACh-induced responses of the transfected PCR product with the transfected plasmid containing ACh-NeuBI0.1 and the same regulatory elements. Responses were similar between PCR amplicon-transfected and plasmid-transfected cells (Fig. 2B).

To improve ACh-NeuBI0.1, we first optimized the lengths of linker 1, between the OpuBC N-terminal fragment (amino acids 30 to 106) and SmBiT, and linker 2 between LgBiT and the OpuBC C-terminal fragment (amino acids 107 to 305), screening linkers of zero to five glycine residues on each end (Fig. 2A). Using the MIDAS-MP protocol of three primary PCR products for assembly in the secondary PCR reaction, we constructed all 36 possible combinations by PCR and expressed them in HEK293A cells without cloning. Among these combinations, whose responses exhibited more than a fourfold difference from weakest to strongest, one glycine at linker 1 and no amino acids at linker 2 were optimal, producing a ~50% increase in luminescence upon treatment with 100 μM ACh (Fig. 2C, left). Performance of this variant produced by traditional plasmid cloning and transfection

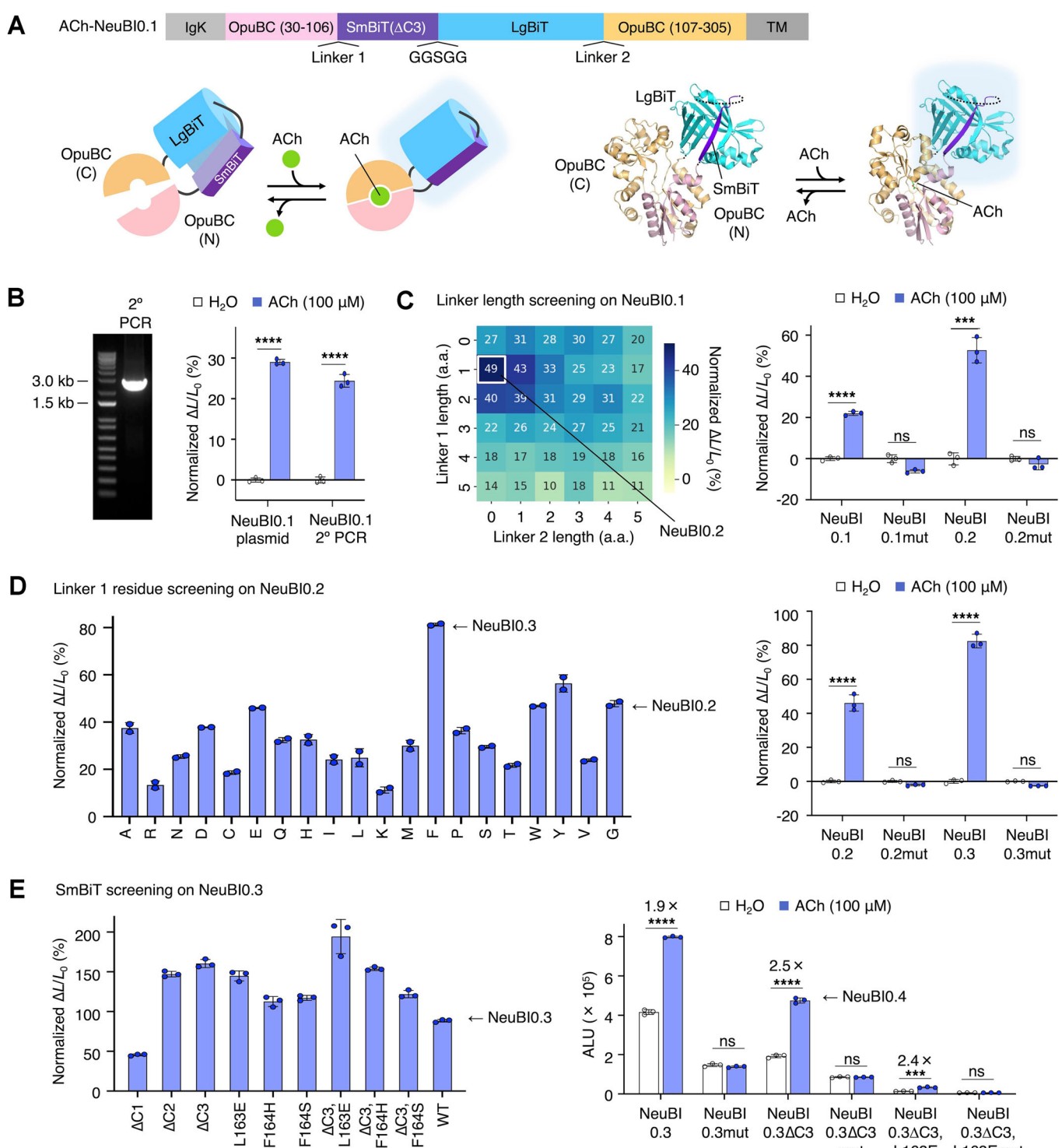

was similar to the transfected PCR product, validating the screening procedure, and this variant was designated ACh-NeuBI0.2. For control experiments, we introduced a mutation (Y140A) at the ACh-binding site of OpuBC into ACh-NeuBI0.1 and ACh-NeuBI0.2 that abolishes ACh binding (Borden et al, 2020). As expected, the resulting ACh-NeuBI0.1mut and ACh-NeuBI0.2mut showed no brightness change upon ACh treatment (Fig. 2C, right),

confirming that ACh-NeuBI0.1 and ACh-NeuBI0.2 responses were triggered by ACh binding to the known interaction site.

Next, to improve the performance of ACh-NeuBI0.2, we used MIDAS-MM to test all 20 amino acids at the single residue of linker 1. Phe was the most effective, improving responsiveness to ~80% (Fig. 2D, left). We designated this variant ACh-NeuBI0.3 and verified it by plasmid cloning and transfection (Fig. 2D, right).

**Figure 2.  Improvement of ACh-NeuBI using MIDAS-MP.**

(A) ACh-NeuBI0.1 architecture. Top, polypeptide map of ACh-NeuBI0.1 elements. Bottom left, cartoon of major structural elements. Bottom right, atomic model rendered from PDB coordinates 6URU (OpuBC) and 5IBO (NanoLuc). ACh acetylcholine. (B) Validation of PCR transfection. Left, secondary PCR yields a single 2.6-kb product as expected. Right, responses of ACh-NeuBI0.1 expressed in HEK293A cells by plasmid transfection or those transfected with PCR products to ACh (100 μM). Left: $p = 4.33 \times 10^{-7}$, Right: $p = 1.66 \times 10^{-5}$. (C) Linker length screening by MIDAS-MP. Left, heatmap showing the responses of the screened variants to ACh (100 μM). Right, responses of plasmid-transfected ACh-NeuBI0.1, ACh-NeuBI0.1mut, ACh-NeuBI0.2, and ACh-NeuBI0.2mut to ACh (100 μM) in HEK293A cells. Left: $p = 1.07 \times 10^{-5}$, Right: $p = 1.78 \times 10^{-4}$. (D) Linker 1 residue screening. Left, responses of PCR product-transfected ACh-NeuBI0.2 mutants with different amino acids on linker 1 to ACh (100 μM). Right, responses of plasmid-transfected ACh-NeuBI0.2, ACh-NeuBI0.2mut, ACh-NeuBI0.3, and ACh-NeuBI0.3mut to ACh (100 μM) in HEK293A cells. Left: $p = 8.09 \times 10^{-5}$, Right: $p = 4.36 \times 10^{-6}$. (E) SmBiT screening. Left, responses of PCR product-transfected ACh-NeuBI0.3 mutants with different SmBiT sequences to ACh (100 μM) in HEK293A cells. Right, responses of selected plasmid-expressed ACh-NeuBI variants. ALU, arbitrary luminescence units. Left: $p = 9.63 \times 10^{-9}$, Middle: $p = 5.82 \times 10^{-6}$, Right: $p = 2.29 \times 10^{-4}$ (B–E) ns not significant; *$p < 0.05$; **$p < 0.01$; ***$p < 0.001$; ****$p < 0.0001$, by unpaired two-tailed Student's $t$-test compared to the $H_2O$ control. Data were presented as mean ± SD. Three technical replicates. Source data are available online for this figure.

## Optimizing domain assembly in mammalian cells by MIDAS and red-shifting NeuBI

Subsequently, within ACh-NeuBI0.3, we used MIDAS-MM to tune the energetics of NanoLuc domain assembly to increase its dependence on ACh. Specifically, we screened SmBiT variants with varied affinities to LgBiT (Wu et al, 2023). Notably, both SmBiT (ΔC3), a variant with a three-residue truncation at the C-terminus, and SmBiT (ΔC3, L163E) exhibited substantial enhancement in their responsiveness (Fig. 2E, left). Upon revalidation by plasmid transfection, both showed a ~2.5-fold increase upon 100 μM ACh exposure, while their non-binding mutants showed no significant signal change (Fig. 2E, right). However, the version with SmBiT (ΔC3, L163E) was extremely dim. Due to its superior brightness, the variant with SmBiT (ΔC3) was designated ACh-NeuBI0.4 and selected for further development.

The emission peak of NanoLuc at ~450 nm is not ideal for tissue penetration due to significant absorption in biological tissues of light at wavelengths shorter than 600 nm (Upputuri and Pramanik, 2019). To improve the collection of NeuBI emissions through tissue, we investigated shifting a fraction of emission to wavelengths longer than 600 nm via resonance energy transfer (RET) (Fig. EV3A). We tested fusion to NeuBI or CyOFP1, which features a cyan excitation peak within NanoLuc's emission band, as well as mScarlet-I and mScarlet3 (Fig. EV3B). All three proteins exhibit peak emission below 600 nm and a majority of emission below 620 nm, and are thus most appropriately classified as orange fluorescent proteins (Chu et al, 2014). Among various fusions tested on HEK293A cells, the protein with mScarlet-I at the N-terminus of ACh-NeuBI0.4 displayed high RET (Fig. EV3C) and an even larger induction of bioluminescence by ACh than NeuBI0.4 (Fig. EV3D), and was designated ACh-NeuBI0.5 (Table EV1). With increasing ACh concentration, ACh-NeuBI0.5 expressed on HEK293A cells (Fig. EV4A) both produced more photons (Fig. EV4B) and exhibited higher RET efficiency (Fig. EV4C). As a result, emission above 610 nm responds more strongly to ACh than overall emission (Fig. EV4D), with 100 μM ACh eliciting a ~6-fold signal increase. The high responsiveness of ACh-NeuBI0.5 in orange-red wavelengths, which are favorable for signal detection in tissue, makes it well suited for in vivo applications.

## Testing computational suggestions for affinity improvement by MIDAS

We tested the affinity and specificity of ACh-NeuBI0.5 for ACh. ACh-NeuBI0.5 expressed on HEK293A cells bound to ACh with an apparent $K_d$ of 898 μM. Affinity for choline was much weaker with $K_d = 11$ mM (Fig. EV4E). ACh-NeuBI0.5 showed no detectable response to glutamate, GABA, serotonin, dopamine, glycine, and norepinephrine, suggesting specificity for ACh (Fig. EV4E). ACh is estimated to reach concentrations as high as 1–10 mM at neuromuscular junctions and release sites of striatal cholinergic interneurons, but ACh-NeuBI0.5's $K_d$ of 898 μM is still far above the basal ACh concentrations in the brain (~500 nM) (Disney and Higley, 2020). We thus sought to further engineer ACh-NeuBI0.5 for improved affinity for ACh.

To enhance the affinity of ACh-NeuBI0.5, we used computational protein structural predictions to identify sites where mutations are predicted to improve protein folding in the ACh-bound state, and screened these predictions by MIDAS. We selected 16 sites in direct contact with ACh, in the second shell, at the interface between OpuBC and NanoBiT, and at the linker region (Fig. 3A). We then used MIDAS-MM to perform enumerated saturation mutagenesis at each site, testing four concentrations of ACh in parallel in the primary screen (Figs. 3B and EV5A). Mutations producing the largest responses to ACh were clustered at three second-shell locations, Gly-550, Leu-560, and Ala-610 (Figs. 3B and EV5A). Specifically, G550Q, G550R, L560C, L560E, A610D, and A610E demonstrated up to fivefold enhancements in affinity compared to ACh-NeuBI0.5 (Figs. 3C and EV5B).

## Deep combinatorial mutagenesis for further affinity and response tuning by MIDAS

We then used MIDAS-MP to improve ACh-NeuBI0.5 further via polyfocal combinatorial saturation mutagenesis. As only Gln and Arg were beneficial at Gly-550, and this site was distant from the other two hotspots at positions 560 and 610, we decided to fix position 550 as either Gln or Arg in the next round. We were concerned that the positively charged Arg may create electrostatic repulsion for ACh in some sectors of sequence and conformation space, so we selected to fix position 550 to Gln. Considering the potential for interactions between residues at positions Leu-560 and Ala-610 due to their spatial proximity, we then conducted enumerated saturated combinatorial mutagenesis at these positions. As occurred with the above combinatorial linker length screening, these two positions are located in different regions of the primary sequence. Thus, three overlapping amplicons were generated in the primary PCRs with the downstream two amplicons encoding variations in their forward primers, and these amplicons were assembled in the secondary PCRs.

Using this procedure, we assessed all 400 possible combinations of side chains at positions 560 and 610 (Fig. 3D). Several mutation combinations, designated QAD, QAE, QCD, QCE, QED, and QEE according to the amino acids at positions 550, 560, and 610, exhibited up to twofold and threefold improvements over ACh-NeuBI0.5 in response to 10 and 100 μM ACh, respectively (Figs. 3E and EV6A), improving in responsivity over G550Q alone. Variant QAE exhibited the highest affinity ($K_d = 188$ μM) and highest brightness at 10–100 μM ACh, while QED exhibited decreased brightness in the absence of ACh (Figs. 3E right and EV6B) and thereby the largest contrast between 0 and 10 or 100 μM ACh (Fig. 3E left). Variant QED exhibited a 640% luminescence increase to 100 μM ACh, representing 29-fold higher responsivity compared to the pre-MIDAS reporter ACh-NeuBI0.1. We designated ACh-NeuBI0.5 QAE as ACh-NeuBI1b for its high brightness, and ACh-NeuBI0.5 QED as ACh-NeuBI1c for its high contrast. Overall, the development of ACh-NeuBI1b and ACh-NeuBI1c involved six optimization steps, five of which were performed by MIDAS (Fig. EV7).

## Analysis of potential mutational effects by AlphaFold3

We sought to understand the physical basis for the stepwise improvement in affinity from ACh-NeuBI0.5 to ACh-NeuBI.05 G550Q, ACh-NeuBI1b (QAE), and ACh-NeuBI1c (QED). We used the AlphaFold3 algorithm (Abramson et al, 2024), accessed via the public Chai-1 server (Discovery et al, 2024), to predict the structure of each variant bound to ACh. Chai-1 accurately recapitulated the known OpuBC-ACh cocrystal structure (PDB: 6V1R) and predicted no significant changes in the ACh binding pocket for an OpuBC domain with the ACh-NeuBI0.5 sequence (Fig. 3F). In ACh-NeuBI0.5 G550Q, the Gln side chain at position 550 was predicted to engage in hydrogen bonds with Tyr-512 and Tyr-615 which are situated on the two lobes on opposite sides of the ACh-binding pocket. Thus, Gln-550 may help stabilize the closed conformation of the OpuBC domain around the bound ACh.

The effects of mutations at 560 and 610 are less clear from these predictions, as the conformations of side chains in the ACh-binding pocket are not predicted to change. As QAE in ACh-NeuBI1b improves affinity over G550Q alone, L560A and A610E may subtly stabilize the closed conformation of the OpuBC domain. These side chains face solvent, so the mutations may optimize the water shell in their vicinity in the closed state. In contrast, the benefit of QED in ACh-NeuBI1b responsivity derives primarily from lowering baseline bioluminescence rather than improving affinity relative to G550Q alone, so the combination of L560E and A610D apparently destabilizes SmBiT-LgBiT complementation in the absence of ACh. As these side chains are located close to SmBiT in the AlphaFold3-predicted structure, they may exhibit a gate-keeping effect against spontaneous SmBiT-LgBiT complementation.

Interestingly, except for A610E, based on the parental codons we used, each individual mutation in ACh-NeuBI1b and ACh-NeuBI1c required two nucleotide changes, including G550Q, L560A, L560E, and A610D. One mutated base in A610D and one in L560A were in degenerate positions, allowing two nucleotide choices for the same amino acid. Using the previously measured estimates of 93.3% nonfunctional genes at a mutagenesis rate of 0.5% per nucleotide[15], the individual beneficial amino acid

alterations would occur in randomly mutated libraries at a nominal frequency per clone of $(0.005 \times 0.067)^2 \times 0.25 \times 0.5$ or 1 in $7.1 \times 10^7$ clones for A610D, $(0.005 \times 0.067)^2 \times 0.25 \times 0.25$ or 1 in $1.4 \times 10^8$ clones for G550Q, L560A, and L560E . These mutations were thus unlikely to have been identified by error-prone PCR within reasonably sized multiwell-based screens of $<10^5$ samples, and thus illustrate again how targeted mutagenesis explores a sequence space mostly inaccessible to error-prone PCR.

## Responsivity, specificity, and reversibility of optimized ACh-NeuBIs

We extensively tested the performance of ACh-NeuBI1b and ACh-NeuBI1c in cells and in vitro. Both proteins expressed well at the plasma membrane (Fig. EV8A) and maintained selectivity for ACh over choline (Fig. EV8B). Additionally, their RET efficiencies increased with ACh concentrations (Fig. EV8C), resulting in higher responses at wavelengths above 610 nm (Fig. EV8D). This feature makes them especially well-suited for in vivo applications.

We next characterized the kinetics of ACh-NeuBI0.5, ACh-NeuBI1b, and ACh-NeuBI1c on HEK293 cells (Fig. EV8E). After ACh addition, all three indicators reached maximal signal within 60 s. After ACh washout, ACh-NeuBI1b and ACh-NeuBI1c exhibited slightly slower reversibility than ACh-NeuBI0.5 (41 s for ACh0NeuBI1b and 47 s for ACh-NeuBI1c vs. 26 s for ACh-NeuBI0.5), consistent with their higher affinity.

We next characterized the affinity of ACh-NeuBI1b and ACh-NeuBI1c in free soluble form. Purified untethered ACh-NeuBI1b and ACh-NeuBI1c proteins were enhanced in terms of their ACh responsivity and selectivity over choline, but surprisingly did not exhibit higher affinity to ACh compared to ACh-NeuBI0.5 in vitro (Figs. EV8F and EV4G). This suggests that our ACh-NeuBI improvements identified by MIDAS with mammalian cells may not have been identified if screened as bacterial periplasmic proteins.

To explore ACh-NeuBI applicability in more physiologically relevant settings, we extended our analysis to primary cortical neurons. Remarkably, ACh-NeuBIs displayed excellent membrane targeting across cell bodies, dendrites, and spines (Fig. EV9A). ACh-NeuBI0.5 exhibited up to a 60-fold increase in luminescence in an ACh concentration-dependent manner (Fig. EV9B), while ACh-NeuBI1b and ACh-NeuBI1c demonstrated even greater responses, achieving up to 200-fold increases (Fig. EV9C,D). Notably, their ACh responsivity exceeded that observed in HEK293A cells (Fig. EV9B–D). Moreover, all ACh-NeuBIs displayed high selectivity for ACh over choline in primary cortical neurons (Fig. EV9B–D). These results suggest that ACh-NeuBIs are well-suited for studying cholinergic transmission under physiologically relevant conditions.

## MIDAS-identified improvements in mammalian cell culture extend to in vivo performance

To investigate the functionality of ACh-NeuBIs in vivo, we expressed ACh-NeuBI0.5, ACh-NeuBI1b, and ACh-NeuBI1c in the mouse liver by hydrodynamic transfection. After 24 h, mice were intraperitoneally injected with FFz and imaged for ~3 min until bioluminescent signals reached a plateau. Once the plateau was reached, saline or ACh at various concentrations was administered and imaging was resumed immediately (Fig. 4A). In

ACh-NeuBI0.5-expressing mice, ACh administered at 0.1, 1, and 10 mg/kg resulted in significant signal increases of 3.5-, 7.5-, and 30-fold compared to saline, respectively (Fig. 4B,C). Compared to ACh-NeuBI0.5, ACh-NeuBI1b and ACh-NeuBI1c demonstrated further improvements at all tested ACh amounts. Specifically, ACh-NeuBI1b and ACh-NeuBI1c showed signal increases of 10.6- and 10.2-fold at 0.1 mg/kg, 38- and 35-fold at 1 mg/kg, and 103- and 173-fold at 10 mg/kg (Fig. 4B,C). Thus, ACh-NeuBI1b and ACh-NeuBI1c detect ACh with larger signal changes than ACh-NeuBI0.5 across a wide range of ACh concentrations in vivo, confirming earlier in vitro results.

To rule out the possibility of ACh influencing substrate delivery or other non-specific factors unrelated to ACh binding to ACh-NeuBIs, we also performed hydrodynamic transfection of a mScarlet-I-NanoLuc plasmid, which is unregulated by ACh, into nude mice. Subsequent measurements of responses to ACh and saline revealed no significant differences between the saline and ACh treatment groups (Fig. EV10). Thus, the observed responses of ACh-NeuBI reporters were specifically triggered by ACh binding. Taken together, these results demonstrate the utility of MIDAS in discovering protein optimizations that are useful in vivo.

## Improving enzyme catalysis by MIDAS-MM

If secondary PCR primers are designed to reamplify the same gene structure as the plasmid template in the mutagenic primary PCR, the resulting secondary PCR products may include non-mutated parental sequences. To test this potential problem, we performed overlap PCR to introduce mutations into the luciferase NanoLuc at one site, with secondary PCR primers recognizing the template plasmid used in the mutagenic primary PCR, then performed sequencing on the PCR products. Indeed, we observed a mixture of wild-type nucleotides and the desired mutant nucleotides at the targeted site (Fig. EV11A,B), indicating that some template plasmid was reamplified in the secondary PCR. In contrast, when we used the modified MIDAS-MM protocol with secondary PCR primers specific for the primary PCR amplicons, sequencing of the full-length secondary PCR products showed that the desired mutations were produced without contamination by parental sequences (Fig. EV11C).

A major use of protein engineering is to enhance enzyme function on particular substrates of interest. NanoLuc luciferase itself is a widely used enzyme in biotechnology, serving as a translational or transcriptional reporter for various in vitro assays (England et al, 2016), a fragment complementation-based reporter of protein–protein interactions (Dixon et al, 2016), a marker of cell or pathogen localization and abundance in vivo (Liu et al, 2021), and as the enzymatic output for live-cell indicators of calcium (Oh et al, 2019), protease activity (Westberg et al, 2024), kinase activity (Wu et al, 2023; Wu et al, 2025), and, as demonstrated by ACh-NeuBI, neurotransmitters. While NanoLuc was originally optimized for the substrate furimazine (Fz), we recently developed FFz and CFz9 as substrates with increased aqueous solubility, allowing higher amounts to be delivered in vivo and improving bioluminescence sensitivity in mice (Gao et al, 2025; Su et al, 2020; Su et al, 2023). FFz and CFz9 differ from FFz by the addition of substituent groups and thus might be expected to prefer a different catalytic pocket surface shape than Fz. However, NanoLuc mutants with improved activity toward CFz and FFz have not yet been described. NanoLuc thus serves as an example of an enzyme whose further optimization toward specific substrates would be of practical benefit.

Based on the NanoLuc crystal structure, we identified 25 residues that form the substrate-binding pocket or were previously proposed to influence catalytic activity (Fig. 5B) and performed MIDAS-MM to generate and express 500 NanoLuc variants representing all 20 possible amino acids at each of these residues. To correct NanoLuc photon production for the amount of transfected PCR product, we placed NanoLuc downstream of the substrate-orthogonal firefly luciferase (FLuc) and the poliovirus P2A ribosome skipping sequence, and used FLuc light production from D-luciferin substrate as a normalization signal (Fig. 5A). We then identified variants with increased NanoLuc/FLuc bioluminescence from Fz, FFz, or CFz9. A D108N mutant exhibited improved photon production from all three substrates, while a D108S mutant exhibited improved photon production from only FFz (Fig. 5C). To verify that MIDAS-MM results were reliable, we further tested these variants by cloning them into plasmids and transfecting them into HEK293A cells. We confirmed D108N generated brighter luminescence than the wild-type NanoLuc with Fz, FFz, or CFz9, while D108S was brighter only with FFz (Fig. 5D). These results demonstrated that the MIDAS-MM strategy can indeed be used for rapid enzyme improvement.

Interestingly, the D108S mutant would have been unlikely to be found without truly deep mutagenesis. This mutation required two adjacent nucleotide changes (GAC to AGC), and an examination of the codon table shows that all paths from Asp to Ser require GA to AG or GA to TC double mutations. Using the previously measured functional recovery rates (Daugherty et al, 2000), the frequency of this double mutation appearing in an error-prone PCR mutagenesis screen with a 0.5% mutagenesis rate is $(0.005 \times 0.067)^2 \times 0.25 \times 0.25 \times 2$, or 1 in $7.1 \times 10^7$ clones. Typically, libraries are oversampled by at least tenfold to overcome the Poisson nature of sampling and any unevenness in mutation generation. This results in a screening requirement of $>10^8$ clones, more than can be reasonably performed using multiwell assays, which are limited to $<10^4$ per day per machine.

While a large FACS-based screen could theoretically screen $10^8$ cells in a week, it is not possible to screen for NanoLuc catalysis by FACS. The relatively low photon production rates of NanoLuc ($\sim10^2$ photons/molecule/s) compared to fluorescent proteins ($>10^8$ photons/molecule/s), as well as the need to normalize for expression differences with FLuc, are incompatible with FACS, in which optical measurements of cells are limited to a $\sim$10-μs transit time. In this way, NanoLuc is representative of the vast majority of enzymes which can be assayed by multiwell analytical chemistry methods such as absorbance spectroscopy or chromatography, but for which a FACS-compatible readout does not exist. Even if FACS were possible for an enzyme of interest, the cell population data obtained in multiwell plates is much less noisy than FACS data. Our results thus also demonstrate how the multiwell format of MIDAS enables gold-standard enzymatic performance assays to be used for primary screening.

## Leveraging massively parallel characterization in MIDAS-MM for rapid sequence-fitness analyses

The ability of MIDAS-MM to reliably generate and characterize each possible mutation at sites of interest in a parallel manner enables not only rapid screening for improved variants, but also allows rapid

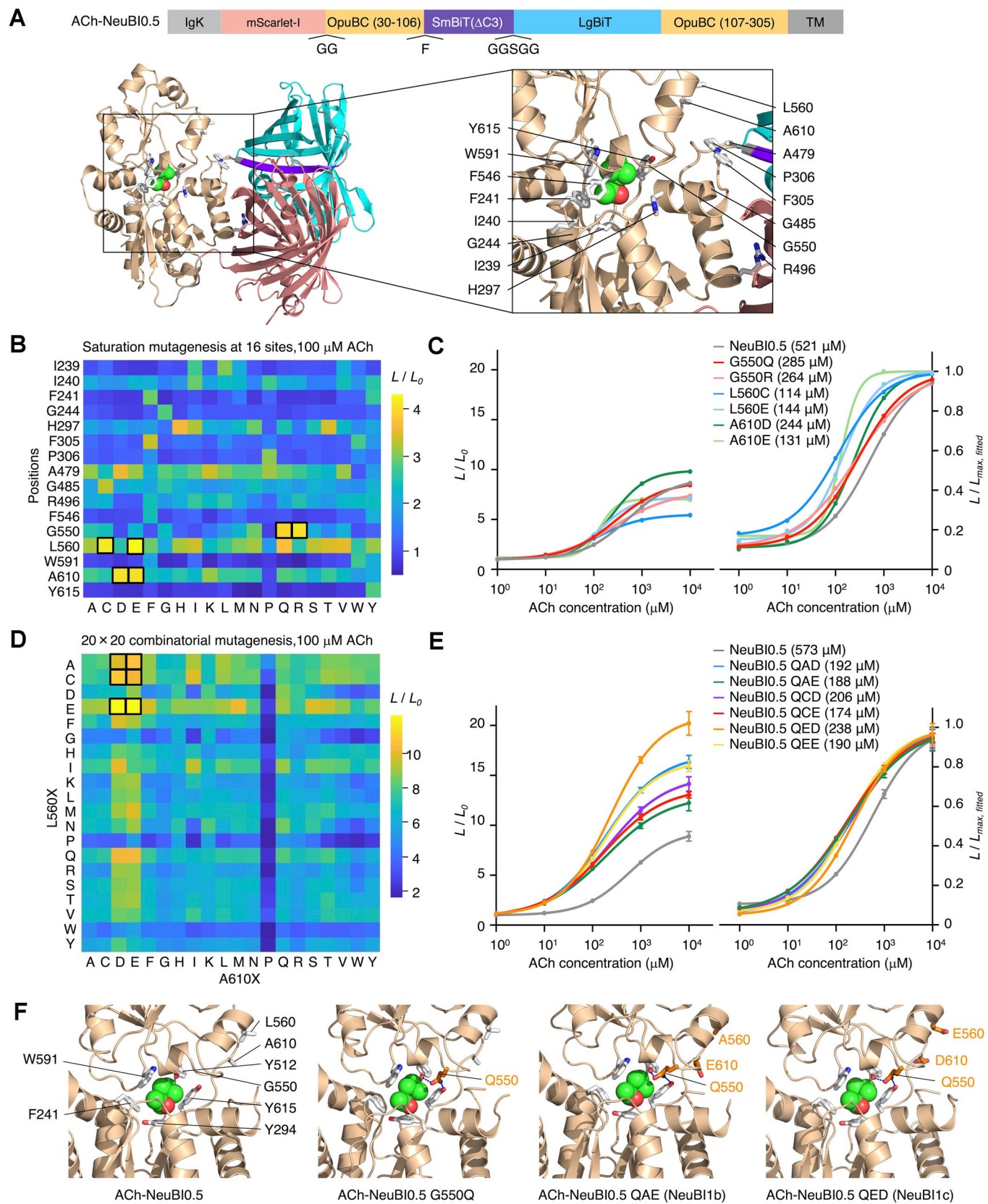

**Figure legends for panels A–F (ACh-NeuBI0.5 construct diagram, structural views, saturation mutagenesis and combinatorial mutagenesis heatmaps, dose-response curves).**

**Figure 3. Deep multi-location combinatorial mutagenesis with MIDAS.**

(A) ACh-NeuBI0.5 structure predicted by Chai-1. Wheat, OpuBC; pink, mScarlet-I; cyan, LgBiT; purple, SmBiT; gray, linkers; green spheres, ACh. Side chains of the 16 residues selected for deep mutagenesis are shown as white sticks. (B) Responses of 16 × 20 mutants created and expressed by MIDAS, tested at 100 μM ACh in HEK293A cells. Boxes, top-performing mutants selected for further characterization. (C) Dose-response curves of plasmids encoding top mutants, normalized to the $H_2O$ control (left) or to fitted maximum (right). Curves were fitted using the "nonlinear regression (log(inhibitor) vs. response – variable slope)" model in Prism. (D) Responses of 20 × 20 combinatorial mutations at positions 560 and 610 were constructed and tested by MIDAS with 100 μM ACh. Boxed cells, top-performing mutants selected for further characterization. (E) Dose-response curves of plasmids encoding top mutants, normalized to the $H_2O$ control (left) or to fitted maximum (right). Curves were fitted using the "nonlinear regression (log(inhibitor) vs. response – variable slope)" model in Prism. (F) Structures of ACh-NeuBI0.5, ACh-NeuBI0.5 G550Q, ACh-NeuBI1b, and ACh-NeuBI1c predicted by Chai-1. Wheat, OpuBC; green spheres, ACh. Side chains of residues in direct contact of ACh are shown as white sticks. Side chains of mutated residues are shown as orange sticks. (C, E) Data were presented as mean ± SD. Three technical replicates. Source data are available online for this figure.

comprehensive sequence-fitness analysis. One characteristic of interest is the tolerance of enzymatic function for substitutions at specific sites. Knowledge of tolerance at each residue is valuable for protein engineering, as tolerant positions may serve as sites for diversification in efforts to improve catalysis or alter substrate specificity when combined with other mutations. In contrast, future engineering efforts can avoid mutating intolerant sites.

For NanoLuc, we used the measured performance of all amino acid substitutions at active-site residues to identify sites where mutations were especially well or poorly tolerated. As a simple measure of tolerance, we summed the enzymatic activity of all 20 side chains normalized to wild-type activity. (Fig. 5C,E) The most tolerant positions across all substrates were Tyr-109, Ile-137, and Trp-161. Among these, Trp-161 faces the outside, and Ile-137 is distant from the substrate (Fig. 5E), so their tolerance may be expected. The most intolerant positions were Pro-40, Ile-56, Tyr-114, and Arg-162. Among these, Pro-40 is expected to play a crucial structural role in fixing backbone φ and ψ angles, Tyr-114 and Ile-56 face the substrate, and Arg-162 is proposed to carry out a key catalytic step in proton donation to the substrate (Nemergut et al, 2023). However, the relative tolerance of Tyr-109 was less predictable, as its side chain also directly contacts the substrate (Fig. 5E).

As the D108S mutant shows an intriguing preference for FFz over Fz or CFz9, we decided to systematically analyze specificity determinants in the active site by examining the influence of each amino acid change on the enzyme's substrate specificity in the NanoLuc sequence-fitness data array. Differences between variants in preference for Fz, FFz, or CFz9 substrates could be deduced from differences in performance on each substrate relative to parental protein (Fig. 6A). For each substrate, we identified two mutations that favored that substrate over each of the other two in their separate assays. These were L18M and G160Q for Fz, V38E and D108S for FFz, and V58N and L92Q for CFz9. We then examined their locations in the substrate-binding pocket relative to the locations of atomic differences between the substrates (Fig. 6B). G160Q can be rationalized as being compatible with the smaller benzyl group of Fz compared to the larger fluorobenzyl group of FFz or CFz9, but other specificity determinants are situated far from sites of structural differences between substrates (Fig. 6B). These results demonstrate how MIDAS can be used to rapidly characterize importance of specific residues to protein function, and how the specificity determinants often cannot be predicted without empirical experimentation.

## Discussion

In this study, we describe microbe-independent deep assembly and screening (MIDAS) as a set of rapid, cost-effective, and versatile methods for engineering improved proteins and performing sequence-fitness analyses in mammalian cells. MIDAS enables the deterministic construction and phenotypic screening of gene libraries without relying on microbial systems, thus substantially accelerating protein optimization. As demonstrations, we used MIDAS to extensively optimize ACh-NeuBI, a bioluminescent indicator for ACh, and to improve the activity of NanoLuc against the substrate FFz. Through five rounds of MIDAS, we achieved a 29-fold improvement in responsivity from 22 to 640% in ACh-NeuBI. Through one round of parallel 25-site 20 amino acid mutagenesis and characterization by MIDAS, we identified NanoLuc mutants with 30% higher photon production from FFz and CFz substrates, characterized substrate-binding sites for substitution tolerance, and discovered variants with altered specificity. This work highlights MIDAS as a powerful and generalizable set of methods for engineering improvements and performing sequence-fitness analysis of proteins in mammalian systems, particularly when native cellular environments are critical for performance evaluation.

In previous work, we showed that PCR assembly of gene chimeras can be used to generate and screen mutations at one location for improved function[17]. The current study extends these results to additional mutational types and patterns. Specifically, we generalize the MIDAS method to multiple types of protein modifications not previously performed by PCR transfection: (1) deep screening of multiple sites throughout a functional surface of the protein as demonstrated by improving NanoLuc catalytic activity, (2) combinatorial linker length optimization, (3) testing of computational predictions for higher-affinity ligand binding; and (4) deep combinatorial mutagenesis of multiple sites located in discontinuous portions of the primary sequence. In addition, we introduce the modified MIDAS-MM protocol, where the use of nested primers avoids re-amplification of parental plasmid in primary PCR reactions, allowing microbe-independent deep assembly and screening to be immediately applied to optimizing a protein of interest without requiring it to be first subcloned into multiple plasmids with different priming sites.

We also demonstrate in this study that MIDAS can be used for rapid and highly parallel sequence-fitness analysis of proteins. Previously, when sequence-fitness analysis was desired, researchers would either perform deep sequencing of variants generated by shallow random mutagenesis and selected in microbial hosts (i.e., DMS), or less commonly individually construct and collect mutants to build a truly deep variant library at sites of interest. The limitations of these methods, where either mutagenic depth or speed were sacrificed, severely limited the ability of researchers to perform detailed sequence-fitness analysis of proteins of interest. In MIDAS, hundreds of variants are rapidly created in parallel using a

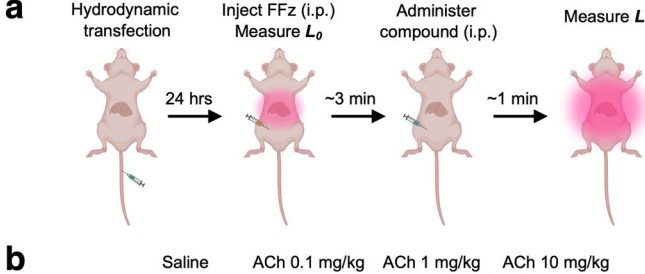

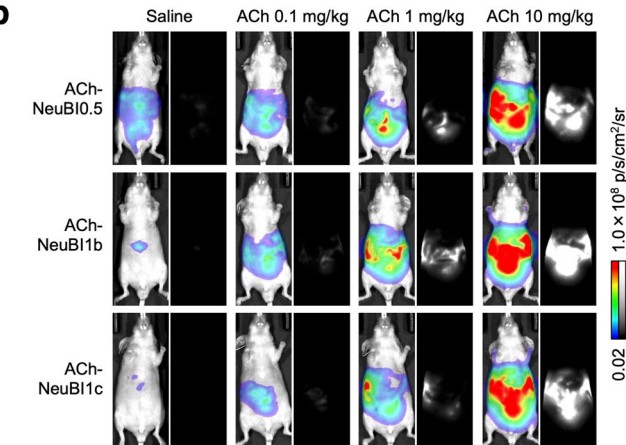

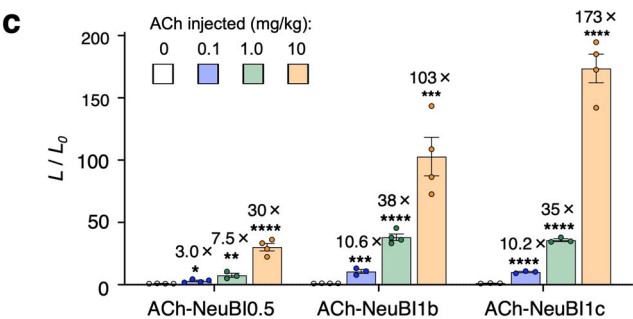

**Figure 4. In vivo validation of ACh-NeuBIs.**

(A) Experimental scheme for testing NeuBI in the mouse liver. NeuBI plasmids are hydrodynamically transfected into nude mice. After 24 h, FFz is injected intraperitoneally (i.p.) to measure baseline luminescence. Once a plateau is reached, saline or ACh is injected i.p., and imaging is continued. (B) Representative bioluminescence images acquired after the treatment of saline or ACh (0.1 mg/kg, 1 mg/kg, or 10 mg/kg) in NeuBI0.5-, NeuBI1b-, or NeuBI1c-expressing mice. (C) Fold of signal increase in response to ACh (0.1, 1, or 10 mg/kg), normalized to the saline control. ACh-NeuBI0.5: 0.1 mg/kg: $p = 0.014$, 1 mg/kg: $p = 0.004$, 10 mg/kg: $p = 6.64 \times 10^{-5}$, ACh-NeuBI1b: 0.1 mg/kg: $p = 7.76 \times 10^{-4}$, 1 mg/kg: $p = 7.62 \times 10^{-6}$, 10 mg/kg: $p = 5.84 \times 10^{-4}$, ACh-NeuBI1c: 0.1 mg/kg: $p = 6.21 \times 10^{-5}$, 1 mg/kg: $p = 6.47 \times 10^{-6}$, 10 mg/kg: $p = 5.33 \times 10^{-5}$ ns not significant; $*p < 0.05$; $**p < 0.01$; $***p < 0.001$; $****p < 0.0001$, by unpaired two-tailed Student's $t$-test compared to the saline control. Data were presented as mean ± SEM. One biological replicate and three-four technical replicates. Source data are available online for this figure.

single PCR machine, and then expressed in mammalian cells in multiwell plates with one variant per well for high-quality protein assays the following day. Indeed, by enabling both engineering and sequence-fitness analysis in one procedure, as demonstrated with NanoLuc, MIDAS blurs the boundaries between previously separate efforts of screening and analysis, enabling a new type of unified protein engineering workflow.

We further consider the degree to which MIDAS accelerates protein engineering or mutational analysis. Traditionally, performing these procedures in mammalian cells involves plasmid cloning. For example, to perform a deep sequence-function analysis of esterase function, more than 1000 variants were manually cloned, purified, and transfected into mammalian cells (Markin et al, 2021). Similarly, more than 800 plasmids were cloned, purified, and transfected into mammalian cells to develop jGCaMP8 calcium indicators (Zhang et al, 2023). Cloning involves, at a minimum, PCR, vector digestion, transformation, colony picking, microbial culture, plasmid purification, and sequence verification. In contrast, MIDAS substitutes the entire cloning procedure with simply two PCR steps. In our experience, generating 384 variants via MIDAS requires only ~4 h of hands-on time and $2000 in reagent costs. In contrast, an experienced researcher working in parallel on 24 samples at a time to clone the same number of variants would require 192 h of time and ~$20,000 in reagent costs. Thus, MIDAS is ~48 times faster and ~10 times less expensive than cloning-based approaches, providing at least an order of magnitude improvement in time and expense for high-quality variant screening or characterization. Random mutagenesis and ultra-high-throughput pooled screening approaches are still useful for identifying functional sites in proteins in a relatively unbiased manner, if an assay compatible with pooled screening exists. However, MIDAS could be used to mutate every residue to every possible side chain in smaller proteins, and is especially useful for exploring specific sequence regions quickly, accurately, and cost-effectively.

The deterministic construction of mutants at sites of interest, as performed in MIDAS, also offers a solution to the inefficiency of error-prone PCR in generating functional mutants with more than one nucleotide change. Seven of the nine individual beneficial amino acid changes discovered in this study required two or three nucleotide mutations. Based on published measurements, functional clones with two mutations generated by whole-gene error-prone PCR would occur once in $>7.1 \times 10^7$ clones, which can only be screened by noisy single-cell fluorescence-based methods such as FACS. If beneficial sites for mutation can be identified, then deterministic mutagenesis allows all possibilities to be generated and assayed in 20 wells for one site or 400 wells for two sites, allowing for multiwell-based measurements of any type and with less noise than FACS. Once sites for targeted mutagenesis are identified, MIDAS then tests them more quickly and efficiently than cloning with degenerate oligonucleotides. In contrast, using degenerate oligonucleotides, mutant libraries of all possible amino acid combinations at P positions (e.g., encoded as NNS or NNK) can be expressed and screened in $32^P \times 10$ cells, where an oversampling factor of ten overcomes Poisson noise to provide a 99% probability that all possible combinations are present[42]. Screening of targeted degenerate codon-based libraries in mammalian cells thus requires screening only 320 samples for one site, but becomes cumbersome to scale up beyond that, with just two sites requiring the screening of >10,000 library samples, a 25-fold larger number than required by MIDAS. Thus, by allowing deterministic construction by PCR and transfection directly in mammalian cells, MIDAS restricts the required sample size to the exact number of combinations while also avoiding the need to construct, grow, and purify each combination as a bacterial plasmid.

Looking forward, MIDAS should be useful for rapidly generating high-quality datasets for training machine-learning models on

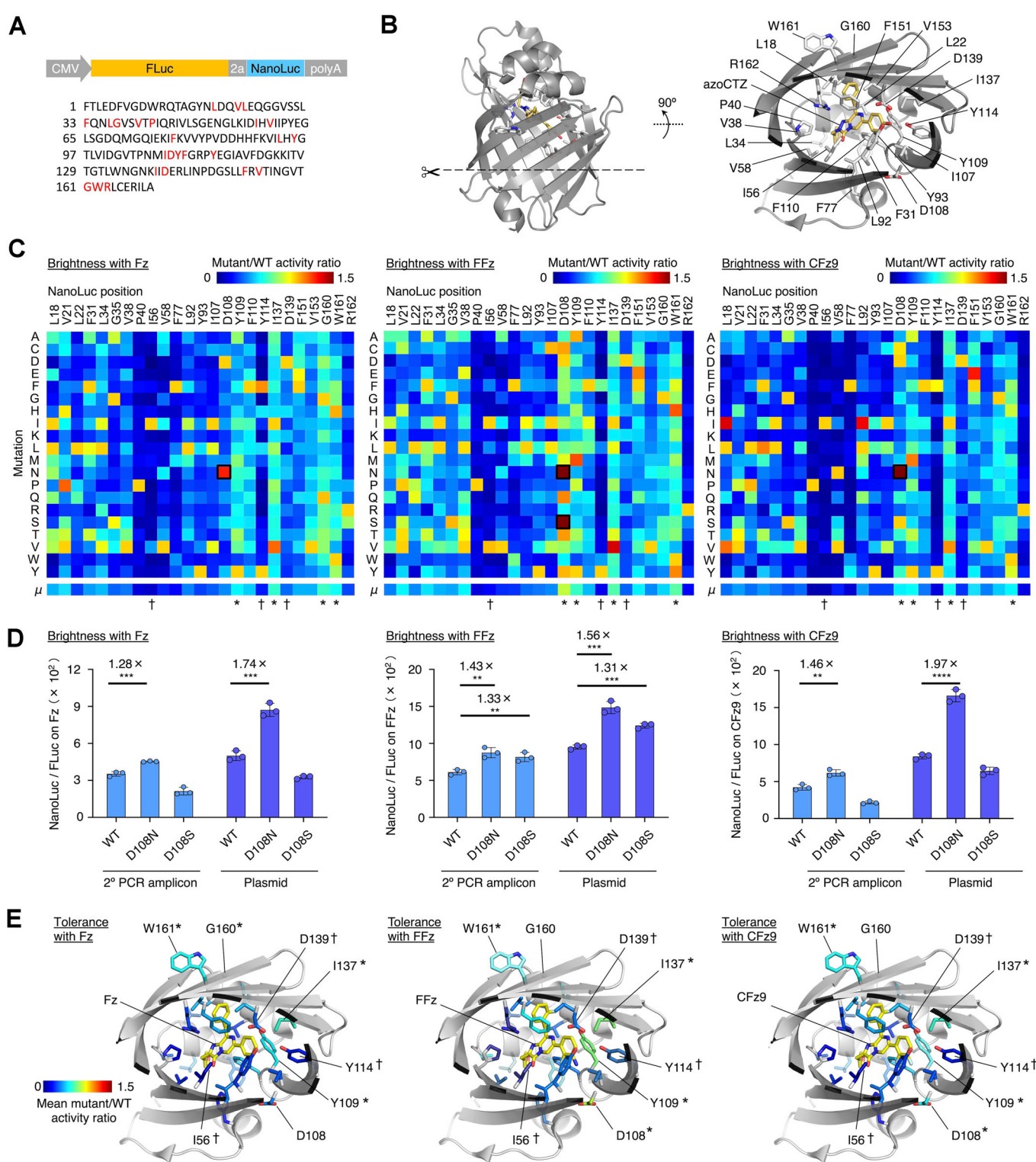

**A**

```
CMV  FLuc  2a  NanoLuc  polyA
```

```
  1  FTLEDFVGDWRQTAGYNLDQVLEQGGVSSL
 33  FQNLGVSVTPIQRIVLSGENGLKIDIHVIIPYEG
 65  LSGDQMGQIEKIFKVVYPVDDHHFKVILHYG
 97  TLVIDGVTPNMIDYFGRPYEGIAVFDGKKITV
129  TGTLWNGNKIIDERLINPDGSLLFRVTINGVT
161  GWRLCERILA
```

**B**

**C**

Brightness with Fz — Mutant/WT activity ratio 0 – 1.5

Brightness with FFz — Mutant/WT activity ratio 0 – 1.5

Brightness with CFz9 — Mutant/WT activity ratio 0 – 1.5

**D**

Brightness with Fz

Brightness with FFz

Brightness with CFz9

**E**

Tolerance with Fz

Tolerance with FFz

Tolerance with CFz9

Mean mutant/WT activity ratio 0 – 1.5

---

any type of protein sequence-fitness relationships. A major challenge in applying machine learning to predict the effects of mutations on dynamic protein functions, such as enzyme catalysis or conformational changes, is the limited availability of high-quality experimental datasets, particularly for functional optimization in mammalian systems (Markus et al, 2023). MIDAS' ability to

rapidly generate and characterize protein mutants directly in mammalian cells in high-quality enzymatic assays should be useful for training machine-learning algorithms for rational protein engineering. For example, the molecular changes in our final round of optimization do not contact ACh and are not predicted to alter protein folding, but may instead affect weak interactions

**Figure 5.  MIDAS-MM concept and application to rapid deep enzyme characterization and improvement.**

(A) Organization of the construct for assessing NanoLuc variant activity relative to FLuc. (B) Sequence and structure of NanoLuc luciferase bound to substrate analog 3-methoxy-furimazine (PDB:7SNT). The side chains of the 25 residues selected for saturation mutagenesis are shown as white sticks; the 23 that are not Gly or Ala are labeled. (C) Relative catalytic activity of all 20 NanoLuc variants at each of the 25 residues on 5 µM of Fz, FFz, or CFz9 after generation and expression by MIDAS-MM. Mean normalized score below the maps indicates the mutational tolerance at each active-site position. Asterisks, sites showing tolerance, selected for later analysis. Daggers, sites showing intolerance selected for later analysis. (D) Relative catalytic activity of all 20 NanoLuc variants expressed by MIDAS-MM or after plasmid cloning and transfection. left: WT vs D108N in amplicon: $p = 0.0008$, in plasmid: $p = 0.0006$; middle: WT vs D108N in amplicon: $p = 0.0042$, in plasmid: $p = 0.0004$, WT vs D108S in amplicon: $p = 0.0079$, in plasmid: $p = 0.0003$; left: WT vs D108N in amplicon: $p = 0.0042$, in plasmid: $p = 0.0001$, *$p < 0.05$; **$p < 0.01$; ***$p < 0.001$; ****$p < 0.0001$, by unpaired two-tailed Student's $t$-test compared to the saline control. Error bars, SDs. Three technical replicates. (E) Models of NanoLuc with tolerance scores at each active-site residue, coupled with Fz (left), FFz (middle), and CFz9 (right), respectively. Selected tolerant and intolerant sites are indicated. Source data are available online for this figure.

between domains or between side chains and solvent. Current computational algorithms based on machine learning from known structural, biochemical, or genetic data on protein–protein interactions do not predict interactions of synthetic proteins well (Gainza et al, 2025), so these types of improvements can currently only be obtained through empirical screening. MIDAS greatly facilitates such screens, which at the same time can be used to provide data for the training of future computational models of protein function.

While the ACh-NeuBIs developed in this study served to demonstrate the versatility and speed of MIDAS, they constitute an important methodological advance in themselves as well. Unlike fluorescent indicators, bioluminescent indicators do not require the insertion of optical elements such as windows or fibers to deliver excitation light, and thus can report biological events without damaging tissue and from locations undergoing large movements, such as muscles or abdominal organs. Our ACh-NeuBIs are the first examples of bioluminescent neurotransmitter indicators with demonstrated functionality in vivo. In addition, compared to the fluorescent ACh indicators GRAB-ACh3.0 and iAChSnFR (Borden et al, 2020; Jing et al, 2020), ACh-NeuBI1b and ACh-NeuBI1c demonstrate larger input and output dynamic ranges. Specifically, GRAB-ACh3.0 and iAChSnFR expressed on mammalian cells sense ACh concentrations up to 100 µM, and respond with maximum increases of threefold and eightfold, respectively. ACh-NeuBI1b and ACh-NeuBI1c respond to 100 µM ACh with fivefold and sevenfold brightness increases, respectively, then respond further to higher concentrations to reach 15-fold and 20-fold maximum changes. Thus, the higher $K_d$ of ACh-NeuBI1b/c extends their input dynamic range while their larger overall responsivity maintains sizable responses at intermediate inputs. This high input dynamic range may be useful where ACh concentrations exceed 100 µM, such as neuromuscular junctions and cholinergic interneuron release sites in the striatum (Matityahu et al, 2023; Nosaka and Wickens, 2022; Smart and McCammon, 1998). The higher $K_d$ also reduces competition with endogenous ACh receptors and thereby is less likely to perturb ACh responses. This may be especially useful where responses are mediated by low-affinity ACh receptors; for example, ACh activates skeletal muscles via α7 nAChR, which binds to ACh with $K_d$ ~180 µM (Letsinger et al, 2022).

In summary, the set of MIDAS protocols described here provides a generalizable approach for rapid protein optimization and sequence-fitness analysis directly in mammalian cells, bypassing time- and resource-intensive microbial cloning steps. As examples, we used MIDAS to extensively optimize ACh-NeuBI, a genetically encoded bioluminescent indicator for noninvasive imaging of acetylcholine in vivo, and to improve the catalytic performance of NanoLuc luciferase on two recently developed substrates. With ACh-NeuBI, we demonstrated that MIDAS could be broadly applied to rapidly test multiple types of mutations: linker length, linker composition, protein fold stability, computationally suggested ligand affinity changes, and distantly located mutation combinations. The final optimized variants, ACh-NeuBI1b and ACh-NeuBI1c, enabled robust detection of ACh dynamics in primary neurons and living mice, highlighting the utility of MIDAS in creating highly functional novel proteins for mammalian systems. With NanoLuc, we demonstrated that MIDAS can be used to simultaneously engineer improved catalytic performance and characterize mutation-tolerant and intolerant sites in the substrate-binding site. Notably, the MIDAS workflow can be directly coupled to the wide variety of existing high-quality multiwell-based assays of protein function, including fluorescence, bioluminescence, absorbance, or chromatography assays. MIDAS can thus be readily extended to a wide variety of protein families and activities, providing a powerful and efficient platform for protein engineering and sequence-fitness analysis in mammalian cells.

# Methods

**Reagents and tools table**

| Reagent/resource | Reference or source | Identifier or catalog number |
| --- | --- | --- |
| **Experimental models** | | |
| HEK293A cells | Invitrogen | R70507 |
| Primary rat cortical neurons | Prepared from E18 Sprague Dawley rats | |
| Sprague Dawley (wild-type) rat | Charles River | Sprague Dawley |
| Nude mouse | Jackson Laboratories | J:NU no. 007850 |
| **Recombinant DNA** | | |
| pAAV.CAG.iAChSnFR | Addgene | 137955 |
| pF4Ag.NanoLuc (ATG-42) | Addgene | 137777 |
| Lck-mScarlet-I | Addgene | 98821 |
| **Oligonucleotides and other sequence-based reagents** | | |
| MIDAS-CMV-F | This study | ggccagatatacgcgttgacattg |
| MIDAS-BGH-R | This study | ctttccgcctcagaagccatagag |
| NanoLuc-D108-R | This study | gatcatgttcggcgtaacc |

| Reagent/resource | Reference or source | Identifier or catalog number |
| --- | --- | --- |
| NanoLuc-D108A-F | This study | ggttacgccgaacatgatc**GCC**tatttcggacggccgtatg |
| NanoLuc-D108C-F | This study | ggttacgccgaacatgatc**TGC**tatttcggacggccgtatg |
| NanoLuc-D108D-F | This study | ggttacgccgaacatgatc**GAC**tatttcggacggccgtatg |
| NanoLuc-D108E-F | This study | ggttacgccgaacatgatc**GAG**tatttcggacggccgtatg |
| NanoLuc-D108F-F | This study | ggttacgccgaacatgatc**TTC**tatttcggacggccgtatg |
| NanoLuc-D108G-F | This study | ggttacgccgaacatgatc**GGG**tatttcggacggccgtatg |
| NanoLuc-D108H-F | This study | ggttacgccgaacatgatc**CAC**tatttcggacggccgtatg |
| NanoLuc-D108I-F | This study | ggttacgccgaacatgatc**ATC**tatttcggacggccgtatg |
| NanoLuc-D108K-F | This study | ggttacgccgaacatgatc**AAG**tatttcggacggccgtatg |
| NanoLuc-D108L-F | This study | ggttacgccgaacatgatc**CTG**tatttcggacggccgtatg |
| NanoLuc-D108M-F | This study | ggttacgccgaacatgatc**ATG**tatttcggacggccgtatg |
| NanoLuc-D108N-F | This study | ggttacgccgaacatgatc**AAC**tatttcggacggccgtatg |
| NanoLuc-D108P-F | This study | ggttacgccgaacatgatc**CCC**tatttcggacggccgtatg |
| NanoLuc-D108Q-F | This study | ggttacgccgaacatgatc**CAG**tatttcggacggccgtatg |
| NanoLuc-D108R-F | This study | ggttacgccgaacatgatc**AGA**tatttcggacggccgtatg |
| NanoLuc-D108S-F | This study | ggttacgccgaacatgatc**TCC**tatttcggacggccgtatg |
| NanoLuc-D108T-F | This study | ggttacgccgaacatgatc**ACC**tatttcggacggccgtatg |
| NanoLuc-D108V-F | This study | ggttacgccgaacatgatc**GTG**tatttcggacggccgtatg |
| NanoLuc-D108W-F | This study | ggttacgccgaacatgatc**TGG**atttcggacggccgtatg |
| NanoLuc-D108Y-F | This study | ggttacgccgaacatgatc**TAC**tatttcggacggccgtatg |
| ACh NeuBI-L560-R | This study | tttttgcaggcctgggtaacc |
| ACh NeuBI-L560A-F | This study | ggttacccaggcctgcaaaaa**GCC**tacaatttcaaattcaagcacaccaaaagc |
| ACh NeuBI-L560C-F | This study | ggttacccaggcctgcaaaaa**TGC**tacaatttcaaattcaagcacaccaaaagc |
| ACh NeuBI-L560D-F | This study | ggttacccaggcctgcaaaaa**GAC**tacaatttcaaattcaagcacaccaaaagc |
| ACh NeuBI-L560E-F | This study | ggttacccaggcctgcaaaaa**GAG**tacaatttcaaattcaagcacaccaaaagc |
| ACh NeuBI-L560F-F | This study | ggttacccaggcctgcaaaaa**TTC**tacaatttcaaattcaagcacaccaaaagc |
| ACh NeuBI-L560G-F | This study | ggttacccaggcctgcaaaaa**GGG**tacaatttcaaattcaagcacaccaaaagc |
| ACh NeuBI-L560H-F | This study | ggttacccaggcctgcaaaaa**CAC**tacaatttcaaattcaagcacaccaaaagc |
| ACh NeuBI-L560I-F | This study | ggttacccaggcctgcaaaaa**ATC**tacaatttcaaattcaagcacaccaaaagc |
| ACh NeuBI-L560K-F | This study | ggttacccaggcctgcaaaaa**AAG**tacaatttcaaattcaagcacaccaaaagc |
| ACh NeuBI-L560L-F | This study | ggttacccaggcctgcaaaaa**CTG**tacaatttcaaattcaagcacaccaaaagc |
| ACh NeuBI-L560M-F | This study | ggttacccaggcctgcaaaaa**ATG**tacaatttcaaattcaagcacaccaaaagc |
| ACh NeuBI-L560N-F | This study | ggttacccaggcctgcaaaaa**AAC**tacaatttcaaattcaagcacaccaaaagc |
| ACh NeuBI-L560P-F | This study | ggttacccaggcctgcaaaaa**CCC**tacaatttcaaattcaagcacaccaaaagc |
| ACh NeuBI-L560Q-F | This study | ggttacccaggcctgcaaaaa**CAG**tacaatttcaaattcaagcacaccaaaagc |
| ACh NeuBI-L560R-F | This study | ggttacccaggcctgcaaaaa**AGA**tacaatttcaaattcaagcacaccaaaagc |
| ACh NeuBI-L560S-F | This study | ggttacccaggcctgcaaaaa**TCC**tacaatttcaaattcaagcacaccaaaagc |
| ACh NeuBI-L560T-F | This study | ggttacccaggcctgcaaaaa**ACC**tacaatttcaaattcaagcacaccaaaagc |
| ACh NeuBI-L560V-F | This study | ggttacccaggcctgcaaaaa**GTG**tacaatttcaaattcaagcacaccaaaagc |
| ACh NeuBI-L560W-F | This study | ggttacccaggcctgcaaaaa**TGG**tacaatttcaaattcaagcacaccaaaagc |
| ACh NeuBI-L560Y-F | This study | ggttacccaggcctgcaaaaa**TAC**tacaatttcaaattcaagcacaccaaaagc |
| before linker1_R | This study | cgggaaccccagaatatccacaag |
| linker1_0aa_F | This study | cttgtggatattctggggttcccg |
| linker1_1aa_F | This study | cttgtggatattctggggttcccgGGTGTGACCGGCTACCGGCTG |
| linker1_2aa_F | This study | cttgtggatattctggggttcccgGGTggaGTGACCGGCTACCGGCTG |
| linker1_3aa_F | This study | cttgtggatattctggggttcccgGGTggagggGTGACCGGCTACCGGCTG |
| linker1_4aa_F | This study | cttgtggatattctggggttcccgGGTggagggGGAGTGACCGGCTACCGGCTG |
| linker1_5aa_F | This study | cttgtggatattctggggttcccgGGTggagggGGAggtGTGACCGGCTACCGGCTG |
| before linker2_R | This study | GCTGTTGATGGTTACTCGGAACAG |
| linker2_0aa_F | This study | CTGTTCCGAGTAACCATCAACAGCgcgactactgatccagaaggtgc |
| linker2_1aa_F | This study | CTGTTCCGAGTAACCATCAACAGCggagcgactactgatccagaaggtgc |
| linker2_2aa_F | This study | CTGTTCCGAGTAACCATCAACAGCggaGGCgcgactactgatccagaaggtgc |
| linker2_3aa_F | This study | CTGTTCCGAGTAACCATCAACAGCggaGGCggagcgactactgatccagaaggtgc |
| linker2_4aa_F | This study | CTGTTCCGAGTAACCATCAACAGCggaGGCggaggtgcgactactgatccagaaggtgc |
| linker2_5aa_F | This study | CTGTTCCGAGTAACCATCAACAGCggaGGCggaggtggagcgactactgatccagaaggtgc |
| linker1_(1)A_F | This study | cttgtggatattctggggttcccgGCTGTGACCGGCTACCGGCTG |
| linker1_(2)R_F | This study | cttgtggatattctggggttcccgCGTGTGACCGGCTACCGGCTG |
| linker1_(3)N_F | This study | cttgtggatattctggggttcccgAACGTGACCGGCTACCGGCTG |
| linker1_(4)D_F | This study | cttgtggatattctggggttcccgGACGTGACCGGCTACCGGCTG |
| linker1_(5)C_F | This study | cttgtggatattctggggttcccgTGCGTGACCGGCTACCGGCTG |
| linker1_(6)E_F | This study | cttgtggatattctggggttcccgGAAGTGACCGGCTACCGGCTG |
| linker1_(7)Q_F | This study | cttgtggatattctggggttcccgCAAGTGACCGGCTACCGGCTG |
| linker1_(8)H_F | This study | cttgtggatattctggggttcccgCATGTGACCGGCTACCGGCTG |
| linker1_(9)I_F | This study | cttgtggatattctggggttcccgATCGTGACCGGCTACCGGCTG |

| Reagent/resource | Reference or source | Identifier or catalog number |
|---|---|---|
| linker1_(10)L_F | This study | cttgtggatattctgggggttcccgCTCGTGACCGGCTACCGGCTG |
| linker1_(11)K_F | This study | cttgtggatattctgggggttcccgAAAGTGACCGGCTACCGGCTG |
| linker1_(12)M_F | This study | cttgtggatattctgggggttcccgATGGTGACCGGCTACCGGCTG |
| linker1_(13)F_F | This study | cttgtggatattctgggggttcccgTTCGTGACCGGCTACCGGCTG |
| linker1_(14)P_F | This study | cttgtggatattctgggggttcccgCCTGTGACCGGCTACCGGCTG |
| linker1_(15)S_F | This study | cttgtggatattctgggggttcccgTCCGTGACCGGCTACCGGCTG |
| linker1_(16)T_F | This study | cttgtggatattctgggggttcccgACCGTGACCGGCTACCGGCTG |
| linker1_(17)W_F | This study | cttgtggatattctgggggttcccgTGGGTGACCGGCTACCGGCTG |
| linker1_(18)Y_F | This study | cttgtggatattctgggggttcccgTACGTGACCGGCTACCGGCTG |
| linker1_(19)V_F | This study | cttgtggatattctgggggttcccgGTAGTGACCGGCTACCGGCTG |
| after SmBiT_F | This study | GGCGGAAGTGGAGGTAGCTTC |
| (1)SmBiT(-C1)_R | This study | GAAGCTACCTCCACTTCCGCCAATCTCCTCGAACAGCCGGTAG |
| (2)SmBiT(-C2)_R | This study | GAAGCTACCTCCACTTCCGCCCTCCTCGAACAGCCGGTAGC |
| (3)SmBiT(-C3)_R | This study | GAAGCTACCTCCACTTCCGCCCTCGAACAGCCGGTAGCCG |
| (4)SmBiT(L163E)_R | This study | GAAGCTACCTCCACTTCCGCCCAGAATCTCCTCGAATTCCCGGTAGCCGGTCACGAAc |
| (5)SmBiT(F164H)_R | This study | GAAGCTACCTCCACTTCCGCCCAGAATCTCCTCGTGCAGCCGGTAGCCGGTCAC |
| (6)SmBiT(F164S)_R | This study | GAAGCTACCTCCACTTCCGCCCAGAATCTCCTCGCTCAGCCGGTAGCCGGTCAC |
| (7)SmBiT(L163E,-C3)_R | This study | GAAGCTACCTCCACTTCCGCCCTCGAATTCCCGGTAGCCGGTCACGAAc |
| (8)SmBiT(F164H,-C3)_R | This study | GAAGCTACCTCCACTTCCGCCCTCGTGCAGCCGGTAGCCGGTCAC |
| (9)SmBiT(F164S,-C3)_R | This study | GAAGCTACCTCCACTTCCGCCCTCGCTCAGCCGGTAGCCGGTCAC |
| IgK_R | This study | cactgcctcgcccttgctcaccatgtcaccagtggaacctggaacc |
| mScarlet_F | This study | atggtgagcaagggcgagg |
| mScarlet_R | This study | TCCggtggagtggcggc |
| nOpuBC_F | This study | tccgagggccgccactccaccGGAGGTgggagatctgcgaacgacaccgtag |
| cOpuBC_R | This study | cactgcctcgcccttgctcaccatTCCTCCGCCgtcgacctgcagaattaaacctttctc |
| Myc_F | This study | tccgagggccgccactccaccGGAgaacaaaaactcatctcagaagaggatctg |
| SmBiT_R | This study | cactgcctcgcccttgctcaccatACCGCCAGACCCTCCCTCGAACAGCCGGTAGCCG |
| LgBiT_F | This study | tccgagggccgccactccaccGGAGGCGGAAGTGGAGGTAGCTTC |
| **Chemicals, enzymes and other reagents** | | |
| Acetylcholine chloride | Sigma-Aldrich | A2661 |

| Reagent/resource | Reference or source | Identifier or catalog number |
|---|---|---|
| Choline chloride | Sigma-Aldrich | C7527 |
| L-glutamate potassium monohydrate | Sigma-Aldrich | G1149 |
| Serotonin hydrochloride | Sigma-Aldrich | H9523 |
| Dopamine hydrochloride | Sigma-Aldrich | H8502 |
| Glycine | Sigma-Aldrich | G8790 |
| (-)-Norepinephrine | Selleck Chemicals | S9507 |
| GABA | Tocris | 0344 |
| Phusion Flash High-Fidelity PCR Master Mix | Thermo Scientific | F548L |
| **Software** | | |
| Prism 10 | GraphPad | |
| Office 2024 | Microsoft | |
| ImageJ | NIH | |
| BioRender | BioRender.com | |

## Molecular cloning

DNA primers for molecular cloning were synthesized by Integrated DNA Technologies. Optimized OpuBC domains were PCR amplified from pAAV.CAG.iAChSnFR, a gift from Loren Looger (Addgene 137955)[5]. NanoLuc was PCR amplified from pF4Ag.NanoLuc (ATG-42), a gift from Lance Encell (Addgene 137777). mScarlet-I was PCR amplified from Lck-mScarlet-I, a gift from Dorus Gadella (Addgene 98821). PCR amplification was conducted with 20 to 40 bp primers with ~20 bp overhangs and the Phusion Flash High-Fidelity PCR Master Mix (Thermo Scientific). Molecular cloning was carried out using the In-Fusion HD cloning kit (Takara Bio) with a plasmid vector linearized by restriction enzymes and 1–2 DNA inserts with ~20 bp overhangs. Sequences of constructed plasmids were verified using Sanger sequencing (Elim Biopharm) or whole-plasmid sequencing (Primordium Labs). The intermediate ACh-NeuBI constructs were assembled on the pc3 vector (Addgene) behind the CMV promoter. The best ACh-NeuBI indicator, ACh-NeuBI0.2, and ACh-NeuBI0.2mut were then cloned into the AAV packaging vector pAAV under the CMV promoter.

## Gene construction in MIDAS

In MIDAS, entire transcription units (genes) expressing variants of a protein of interest are synthesized in vitro by PCR and directly transfected into cells. Each transcription unit includes a promoter, the coding sequence of the protein of interest (with or without another protein sequence in a bicistronic arrangement), and a polyadenylation signal. One or more target regions (TRs) are first identified for mutagenesis in the coding sequence of the protein of interest. Each TR can be one codon or several nearby codons (within a ~12-base span) that encode one or several amino acid positions where deep mutagenesis is desired. These positions can be identified based on their position within a synthetic multidomain

protein architecture, e.g., within or close to an introduced linker, or based on structural information, e.g., substrate-facing residues inside a catalytic cleft, or based on prior results of random mutagenesis, e.g., hotspots where prior useful mutations were found after random mutagenesis.

Primer design for polytemplated monofocal microbe-independent deep assembly and screening (MIDAS-PM): A Fo primer is designed to bind to the promoter or a segment of DNA upstream of the promoter in a template plasmid for strand extension by polymerase in the forward or downstream direction, and a Ro primer is designed to bind to the polyadenylation signal or a segment of DNA downstream of the polyadenylation signal for strand extension by polymerase in the reverse or upstream direction. These primers represent the outer ends of the genes that will express the desired protein variants. For each desired variation at the TR, a forward F1 primer is designed to begin with 18–24-nucleotide (nt) sequence from the gene immediately upstream of the TR followed by mutation-encoding sequences within the TR, and to end with an 18–24-nt sequence that matches the template downstream of the TR. A non-mutagenic reverse R1 primer is designed so its 5' region has sufficient reverse-complementary to the 5' region of the F1 primers to hybridize with a melting temperature of 60–66 °C. Alternatively, for each desired variation at the TR, a reverse R1 primer is designed to begin with 18–24-nucleotide (nt) sequence from the gene immediately downstream of the TR followed by mutation-encoding sequences within the TR, and to end with an 18–24-nt sequence that matches the template upstream of the TR. A non-mutagenic forward F1 primer is designed so its 5' region has sufficient reverse-complementary to the 5' region of the R1 primers to hybridize with a melting temperature of 60–66°. These primers are used for two primary PCRs with plasmid templates, primed by Fo to R1 and by F1 to Ro, one of which encodes mutations at the TR and the other of which generates a non-mutant overlapping product. The primary PCR products are then extended in a secondary PCR reaction to recreate the full-length transcription unit. To generate a set of variations at the TR, a series of mutagenic (either F1 or R1) primers are designed for use in a set of mutagenic primary PCRs. Mutagenic primary PCR primers can be acquired in multiwell plates for matched transfer to multiwell PCR plates.

Primer design for polytemplated polyfocal microbe-independent deep assembly and screening (MIDAS-PP): If simultaneous mutagenesis at two TRs in different locations of the gene is desired, three pairs of primers (Fo and R1, F1 and R2, F2 and Ro) are then designed to amplify left, middle, and right PCR segments. The F1 or R1 primer contains a desired alteration at the first TR, and a set of F1 or R1 primers covers all desired variants. The F2 or R2 primer contains a desired alteration at the first TR, and a set of F2 or R2 primers covers all desired variants. For the F1 and R1 pairings, and for the F2 and R2 pairings, other design requirements are the same as for single-region mutagenesis above. Left, middle, and right primary PCRs are then executed using Fo + R1, F1 + R2, and F2 + Ro primer pairs. More than two TRs can be simultaneously mutagenized by increasing the number of primary PCRs whose overlapping products will be annealed and extended in the secondary PCR to recreate the full-length transcription unit. For example, for $i$ TRs where all distances between TRs are too large to be spanned by a single oligonucleotide, then $j = i + 1$ PCRs are performed with $j$ primer pairs, Fo and R1 up to F$i$ and Ro. For each

Fi and Ri pairing, other design requirements are the same as for single-region mutagenesis above.

For both MIDAS-PM and MIDAS-PP, secondary PCRs are performed to generate complete transcription units in vitro as follows: Non-mutated primary PCR products can be gel-purified using a standard gel extraction protocol if desired, as these products are constant across all variants. The product sizes of mutagenic primary PCRs are verified by gel electrophoresis, then the reactions are diluted 50–100× for use as templates in secondary PCRs. The secondary PCRs are conducted using common non-mutagenic primary PCR fragments and diluted mutagenic primary PCRs as templates, and Fo and Ro as primers.

When the coding sequence of interest is only available in one plasmid, then all PCR reactions will use that plasmid as a template. In these cases, the outer Fo and Ro primers in the secondary PCRs will recognize the plasmid template remaining within the unpurified mutagenic primary PCRs. A modified primer design protocol is thus necessary for monotemplated monofocal or polyfocal microbe-independent deep assembly and screening (MIDAS-MM and MIDAS-MP). More precisely, when the plasmid used as template for any mutagenic PCR contains sequences recognized by both outer primers, Fo and Ro, either the Fo primer or the Ro primer, or both, should also bear a 5' extension with a DNA sequence "tag" that is not present in any of the templates for the mutagenic PCRs. In the case of NanoLuc improvement, for example, the sequence 5'-agttctgggggcagctctagagc-3' was added to the 5' end of the Fo primer. An Fo' primer is then designed that overlaps with the unique 5' tag of the Fo primer, and/or an Ro' primer is then designed that overlaps with the unique 5' tag of the Ro primer. To generate mutations at one or more TRs, the Fi and Ri primers are designed for MIDAS-PM and MIDAS-PP. Sequential PCRs are then performed as for MIDAS-PM and MIDAS-PP, except the secondary PCRs use as primers Fo' and Ro' (if both Fo and Ro contain extensions), or Fo' and Ro (if Fo contains an extension), or Fo or Ro' (if Ro contains an extension), disallowing amplification from the plasmid templates and forcing amplification of overlap-extended primary PCR products.

In this study, all PCRs used Phusion Flash High-Fidelity PCR Master Mix (Thermo Scientific), although other polymerases should be possible. All PCRs are performed in 96-well PCR blocks. After verifying the integrity of the secondary PCR products using gel electrophoresis, 1 μL of the PCR products are used with lipofection reagents to create liposomes for transfection of mammalian cells in one well of a 96-well plate.

## Cell culture and transfection

HEK293A cells were obtained from ATCC and used within 6 months of purchase or resuscitation from original stocks; they were confirmed to be mycoplasma-free by PCR-based testing. HEK293A cells (Thermo, R70507) were cultured at 37 °C with 5% $CO_2$ in Dulbecco's Modified Eagle's Medium (DMEM) with 10% fetal bovine serum (FBS), 2 mM L-glutamate, 100 U/mL penicillin, and 100 μg/mL streptomycin (Thermo). Cells were newly bought and tested as mycoplasma-free. Cells were seeded at a density of $4.5 \times 10^4/cm^2$ in white 96-well tissue-culture plates (Greiner Bio-One) and transfected 24 h later with 50 ng plasmids or 1 μL of completed PCR using Lipofectamine 3000 (Thermo) following the manufacturer's instructions. Cells were assayed 24–48 h post-

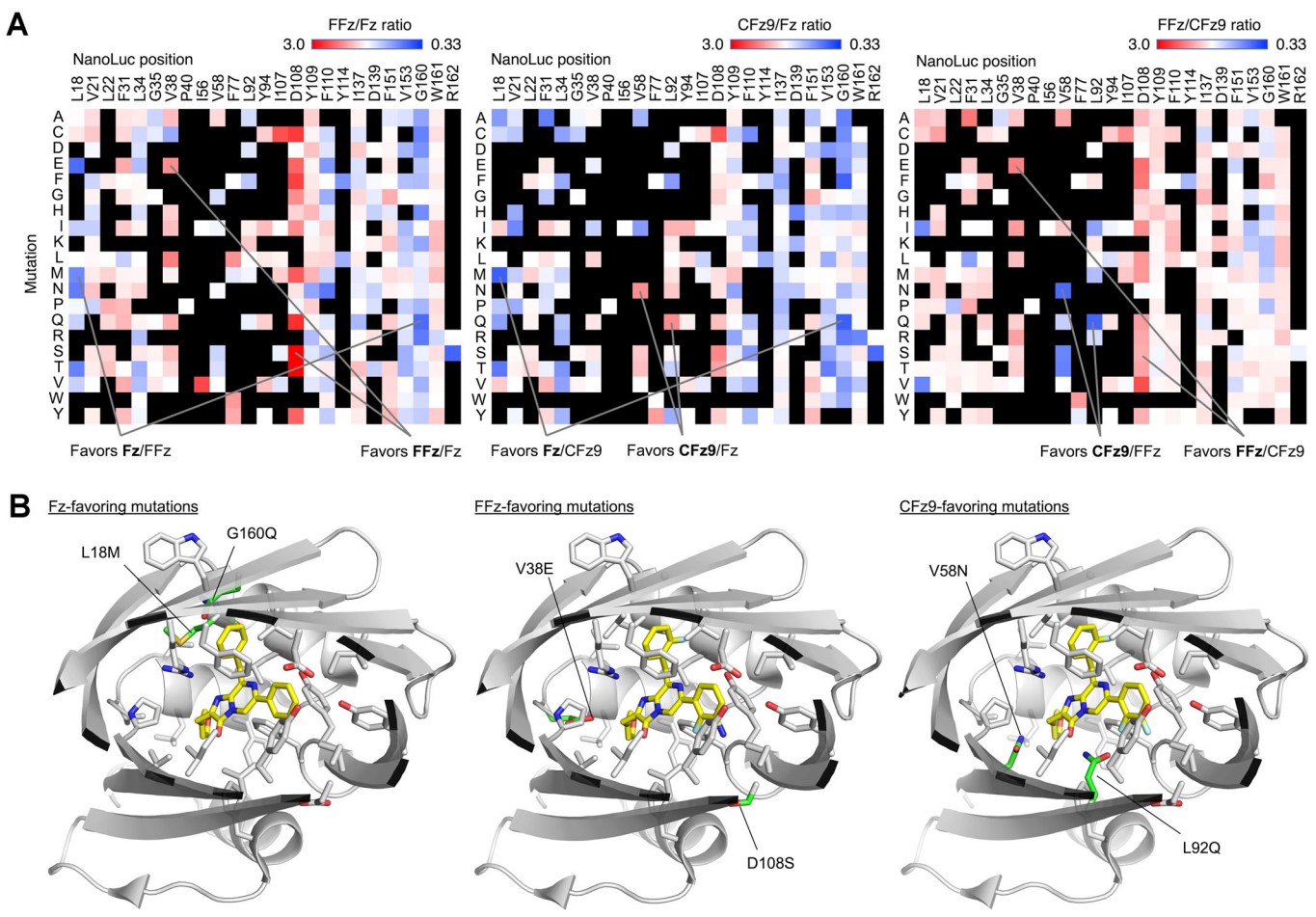

**Figure 6. Substrate specificity of NanoLuc variants.**

(**A**) Heatmaps showing substrate specificity at each active-site position for FFz, CFz9, and Fz, respectively. Black indicates variants where light production from at least one of the substrates was less than 25% of the non-mutated parent; these variants were censored to avoid propagation of error from the higher coefficients of variation with lower brightness values. Mutations that prefer one substrate over the other two are indicated. (**B**) Models of NanoLuc variants preferring Fz (left), FFz (middle), or CFz9 (right). Altered side chains are shown in green. Source data are available online for this figure.

transfection. All plasmids were amplified by XL10-Gold ultra-competent cells (Agilent, 200315) and then purified by Miniprep kit (Zymo) or Midiprep kit (Thermo).

## Primary neuronal culture and transfection

Cortical neurons were isolated from embryonic day 18 Sprague Dawley rat embryos by dissociation in RPMI medium containing 5 units/mL papain (Worthington Biochemical) and 0.005% DNase I and therefore did not require further authentication. Dissociated neurons were plated at a density of $5.3 \times 10^4$ cells/cm² in white TC-treated 96-well plates with clear bottom (Corning) pre-coated overnight with >300-kDa poly-D-lysine hydrobromide (Sigma-Aldrich). Cells were plated overnight in Neurobasal media with 10% FBS, 2 mM GlutaMAX, and B27 supplement (Thermo), then the media were replaced with Neurobasal with 1% FBS, 2 mM GlutaMAX, and B27 the next day. Half of the media was replaced every 3–4 days with fresh media without FBS. 5-Fluoro-20-deoxyuridine (Sigma-Aldrich) was typically added at a final concentration of 16 mM at 7–9 DIV to limit glia growth. Cortical

neurons were transfected at 9–11 DIV using a modified Lipofectamine 2000 (Thermo) transfection protocol in which media in each well of a 96-well plate was replaced for 60 min with 40 μL of DNA-lipid complexes (50 ng of anion exchange-purified ACh-NeuBI plasmids, and 0.1 μL of Lipofectamine 2000, in 40 μL of Neurobasal with 2 mM GlutaMAX per well). Neurons were assayed 48 h post-transfection.

## Cell-based bioluminescence assay of indicators

ACh-NeuBI-expressing cells were assayed in Opti-MEM (for HEK293A cells) or Neurobasal media with 2 mM GlutaMAX (for cultured neurons) using Nano-Glo live-cell assay system (Promega) per manufacturer's instructions. Luminescence was measured on a Varioskan LUX multimode microplate reader (Thermo). Following baseline stabilization and signal plateau, the assay was temporarily paused for the addition of ACh or other neurotransmitters at predetermined concentrations. Subsequently, the plate was returned to the microplate reader to resume signal monitoring. Data analysis was done using signals from the last measurement

prior to neurotransmitter addition, with the peak signals recorded post-addition.

## Confocal microscopy

HEK293A cells were plated in cell culture chamber slides (ibidi, eight-well) and transfected with ACh-NeuBI variants with Lipofectamine 2000 (Thermo Fisher Scientific) 24 h after plating according to the manufacturer's protocol. Twenty-four hours post-transfection, fluorescence was imaged with a confocal microscope (LSM 980, Carl Zeiss) equipped with a Plan-Apochromat 63×/1.4 Oil DIC M27 objective and a GaAsP-PMT detector. Live-cell imaging was performed in an incubation chamber maintained at 37 °C with 5% $CO_2$. mScarlet-I fluorescence was acquired with a 561 nm laser at 0.03 mW for excitation, and emission was collected from 565 to 758 nm.

## In vitro characterization of purified indicators

pET-28a(+)-ACh-NeuBI plasmids were transformed into *E. coli* BL21(DE3) cells. Bacterial cultures were grown at 37 °C to reach an OD600nm of 0.6–0.8. Protein expression was induced using isopropyl β-D-1-thiogalactopyranoside (IPTG) at 16 °C for 16 h. Cells were centrifuged and lysed using a lysis reagent (TieChui *E. coli* Lysis Buffer, ACE Biotech) on ice. The recombinant proteins were purified using Ni-NTA affinity chromatography and were eluted using PBS (Sangon Biotech, E607008) containing 300 mM imidazole. Finally, the eluted proteins were buffer-exchanged into PBS pH 7.4 using a HiTrap desalting column (GE Healthcare). Absorbance at 280 nm was used to determine purified protein concentration using the extinction coefficient calculated by ExPASy's ProtParam tool. Purified indicators were diluted in PBS and added to white 96-well plates. ACh was serially diluted and added to the wells to reach a final concentration as indicated. After incubation at 37 °C, luminescence was determined using the Nano-Glo Luciferase Assay System (Promega) following the manufacturer's instructions with the Varioskan ALF microplate reader (Thermo Scientific).

## Hydrodynamic transfection and bioluminescence imaging in mice

All animal procedures complied with USDA and NIH ethical regulations and were approved by the Stanford Institutional Animal Care and Use Committee of Stanford University. Hydrodynamic transfection was conducted in 6- to 8-week-old male nude mice (NU/J, strain #: 002019, the Jackson Laboratory) by injecting 20 mg of pAAV-CMV-ACh-NeuBI or pAAV-CMV-ACh-NeuBImut plasmids in 2 mL saline via the tail vein within 5–6 s. 18–24 h after hydrodynamic transfection, bioluminescence imaging was performed by intraperitoneal (i.p.) injection of Dulbecco Phosphate Buffered Saline (DPBS, without $Ca^{2+}$ or $Mg^{2+}$, Corning) containing 0.93 mmol of FFz in each mouse. Images were acquired in a Lago-X optical imaging system (Spectral Instruments) every 30 s for 3–4 min, under 1.5–2.5% isoflurane in oxygen for anesthesia. Once the signals plateaued, the imaging was paused, and mice were i.p. injected with either ACh (0.1–10 mg/kg) or saline and put back into the Lago-X optical imaging system to resume imaging. Imaging parameters were: open emission filter, 25-cm field of view, f/1.2

aperture, 2 × 2 binning, and 1–5 s exposure time depending on the brightness of the signals. Images were analyzed in Aura 4.0 software (Spectral Instruments). Data analysis were done using peak signals before and after the injection of ACh (0.1–10 mg/kg) or saline. The investigators were blinded to the identity of the protein variants or treatment dosage administered to the mice during both data acquisition and the subsequent image quantification. Cages were randomized before the start of the experiment to ensure unbiased group allocation.

## Housing and husbandry conditions

Animals were housed in groups of four to five per cage with free access to food and water. Cages were maintained and cleaned on a weekly schedule.

## Statistics

No formal statistical methods were used to predetermine sample size. Sample sizes were chosen based on prior studies in the field and practical considerations. Blinding was not performed as data analysis was conducted using automated scripts. Data were presented as mean ± SD or SEM as indicated in figure legends. Student's *t*-test and curve fitting by the "nonlinear regression (log(inhibitor) vs. response – variable slope)" model were performed in Prism 9 (GraphPad).

Figure 4 and synopsis graphics were created with BioRender.com.

# Data availability

The main data supporting the findings of this study are available within the article and supplementary information. This study includes no data deposited in external repositories.

The source data of this paper are collected in the following database record: biostudies:S-SCDT-10_1038-S44320-026-00210-z.

# Peer review information

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

## Acknowledgements

We thank the Stanford Center for Innovation in in vivo Imaging (SCi3) for instrument use and assistance with in vivo bioluminescent imaging, Dr. Yukun Alex Hao (Stanford) for providing suggestions for the MIDAS screening, and Dr. Zi Yao (UCSF) for reading the manuscript and offering valuable feedback. Illustrations were created with Biorender. Funding was provided by NIH grants 1R21NS122055 (MZL) and 1R21DA048252 (MZL), a Stanford Bio-X Interdisciplinary Initiatives Program Seed Grant (MZL), a Stanford Bio-X PhD Fellowship (YW), and a Stanford Wu Tsai Neurosciences Institute Interdisciplinary Graduate Fellowship (PW).

## Author contributions

**Yan Wu**: Data curation; Formal analysis; Validation; Investigation; Visualization; Methodology; Writing—original draft; Writing—review and editing. **Pengli Wang**: Data curation; Formal analysis; Validation; Investigation; Visualization; Methodology; Writing—review and editing. **Lan Xiang Liu**: Investigation. **Daesun Song**: Investigation. **Qin Qin**: Investigation. **Chao Gao**: Resources; Investigation. **Matt Hageman**: Resources. **Thomas A Kirkland**: Supervision. **Yichi Su**: Supervision; Investigation; Methodology. **Michael Z Lin**: Conceptualization; Supervision; Funding acquisition; Visualization; Methodology; Writing—original draft; Project administration; Writing—review and editing.

Source data underlying figure panels in this paper may have individual authorship assigned. Where available, figure panel/source data authorship is listed in the following database record: biostudies:S-SCDT-10_1038-S44320-026-00210-z.

## Disclosure and competing interests statement

Y.W. is a current employee of Genentech, but this work was performed at Stanford University independently of that affiliation. CG, MH, and TAK are employees of Promega Corporation. TAK is a patent inventor for NanoLuc substrates used in this study. MZL and PW are inventors on a patent application for the monotemplated microbe-independent deep assembly and screening methods described in this study. The remaining authors declare no competing interests.

# Expanded View Figures

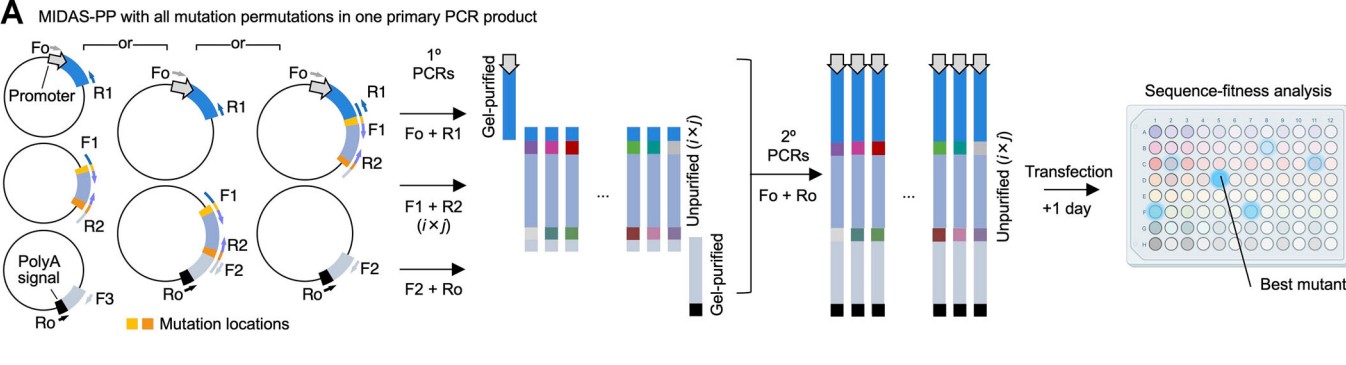

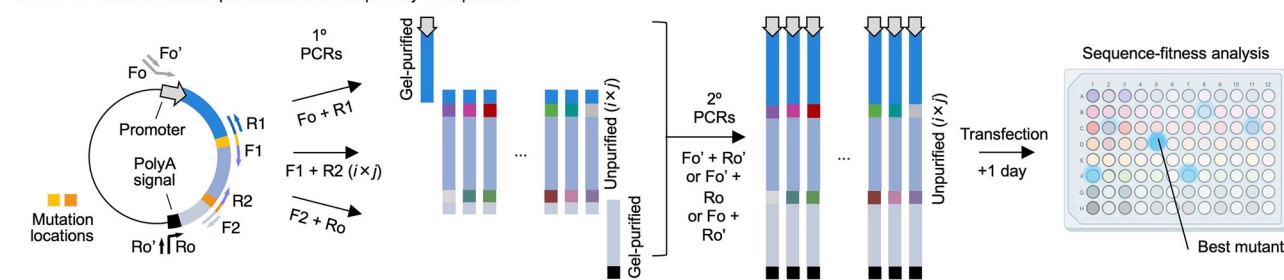

**Figure EV1. Alternative schemes for MIDAS-PP and MIDAS-MP.**

(**A**) MIDAS-PP with all mutation permutations in one primary PCR product. (**B**) MIDAS-MP with all mutation permutations in one primary PCR product. These schemes require a larger number of primary PCRs ($i \times j$) to cover all permutations at two locations compared to the scheme shown in Fig. 1. However, they may be preferable for robotic handling, as combinatorially mixing oligos to set up $i \times j$ primary PCRs and then proceeding one-by-one to the same number of secondary PCRs could be more efficient than combinatorially diluting completed primary PCR products to set up $i \times j$ secondary PCRs. Source data are available online for this figure.

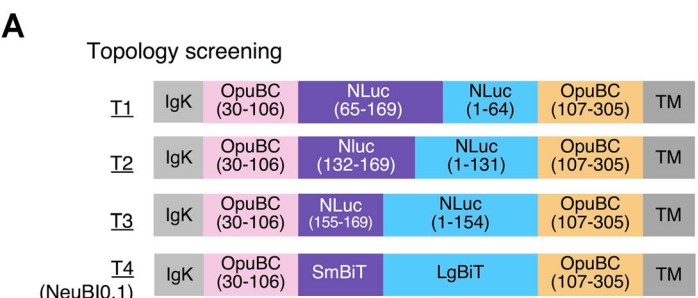

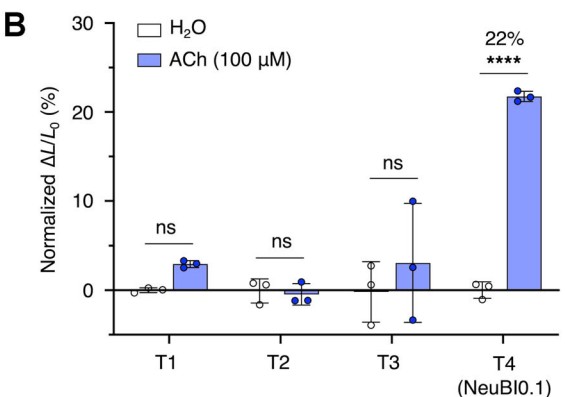

**Figure EV2.  Screening insertions of circularly permuted forms of NanoLuc into OpuBC.**

(A) Topologies T1–T4 testing various circular permutation sites of NanoLuc. (B) Luminescent signal change of T1–T4 upon the treatment of ACh (100 µM), normalized to the $H_2O$ control. $p = 4.29 \times 10^{-6}$. *$p < 0.05$; **$p < 0.01$; ***$p < 0.001$; ****$p < 0.0001$, by unpaired two-tailed Student's $t$-test compared to the $H_2O$ control. Data were presented as mean ± SD. Three technical replicates. Source data are available online for this figure.

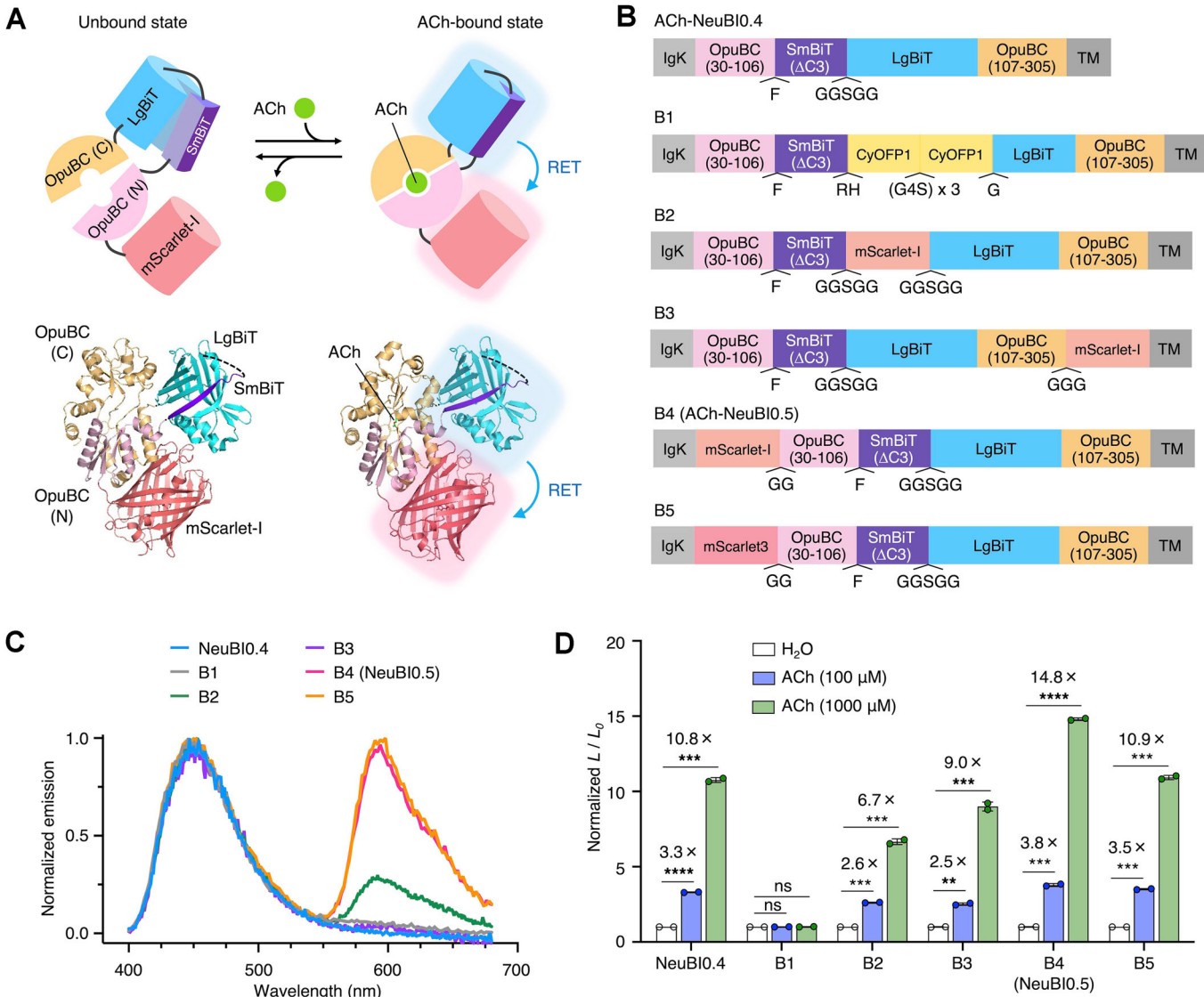

**Figure EV3. Engineering of ACh-NeuBI0.5.**

(A) Proposed mechanism of ACh-NeuBI0.5. Top, cartoon scheme of ACh-NeuBI0.5. Bottom, the structure model of ACh-NeuBI0.5, modified from PDB files 6URU (OpuBC), 5IBO (NanoLuc), and 5LK4 (mScarlet). ACh binding to OpuBC leads to an increase in the luminescence of cpNanoLuc and red light emission from mScarlet-I via resonance energy transfer (RET). (B) Topologies of constructs B1–B5 for red-shifting spectra. (C) Spectra of B1–B5, measured in the presence of ACh (1000 μM). (D) Fold of signal increase of B1–B5 in response to 100 μM ACh or 1000 μM ACh, normalized to the $H_2O$ control. NeuBI0.4: 100 μM: $p = 4.12 \times 10^{-5}$, 1000 μM: $p = 1.31 \times 10^{-4}$, B1: 100 μM: $p = 0.252$, 1000 μM: $p = 0.418$, B2: 100 μM: $p = 1.77 \times 10^{-4}$, 1000 μM: $p = 5.53 \times 10^{-4}$, B3: 100 μM: $p = 1.37 \times 10^{-3}$, 1000 μM: $p = 7.04 \times 10^{-4}$, B4: 100 μM: $p = 5.56 \times 10^{-4}$, 1000 μM: $p = 2 \times 10^{-5}$, B5: 100 μM: $p = 1.23 \times 10^{-4}$, 1000 μM: $p = 1.05 \times 10^{-4}$, ns not significant; *$p < 0.05$; **$p < 0.01$; ***$p < 0.001$; ****$p < 0.0001$, by unpaired two-tailed Student's *t*-test compared to the $H_2O$ control. Data were presented as mean ± SD. Three technical replicates. Source data are available online for this figure.

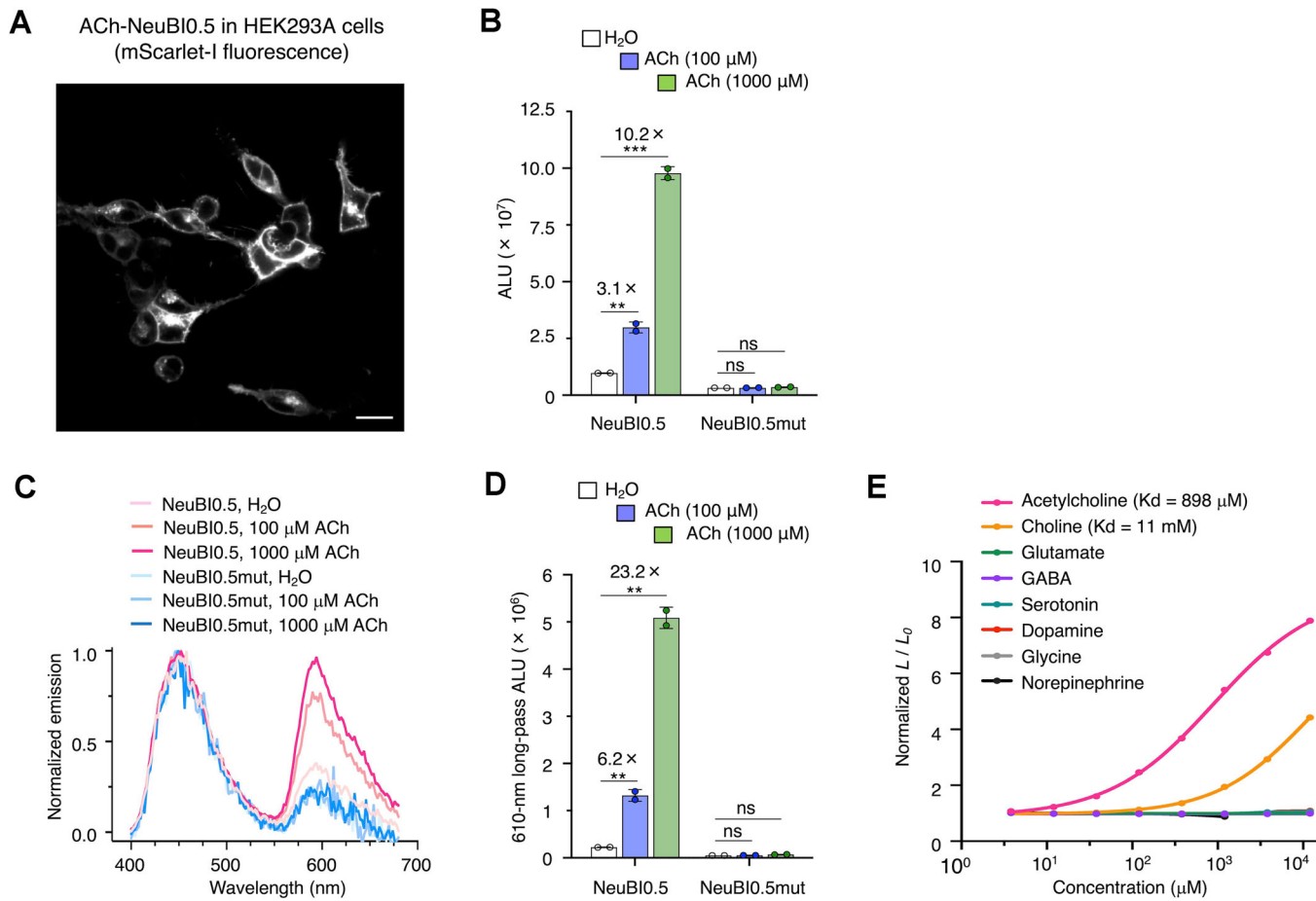

**Figure EV4. Characterization of ACh-NeuBI0.5.**

(A) Confocal microscopy images of mScarlet-I fluorescence of ACh-NeuBI0.5 in HEK293A cells. Scale bar, 20 μm. (B) Luminescence of ACh-NeuBI0.5 and ACh-NeuBI0.5mut in 0, 100, or 1000 μM ACh without filter. ALU, arbitrary luminescence units. NeuBI0.5: 100 μM: $p = 7 \times 10^{-3}$, 1000 μM: $p = 5.34 \times 10^{-4}$, ns not significant; *$p < 0.05$; **$p < 0.01$; ***$p < 0.001$; ****$p < 0.0001$, by unpaired two-tailed Student's $t$-test compared to the $H_2O$ control. Data were presented as mean ± SD. Two technical replicates. (C) Spectra of ACh-NeuBI0.5 and ACh-NeuBI0.5mut in 0, 100, or 1000 μM ACh. (D) Luminescence of ACh-NeuBI0.5 and ACh-NeuBI0.5mut as in (B) but with a 610 nm longpass filter. NeuBI0.5: 100 μM: $p = 6.3 \times 10^{-3}$, 1000 μM: $p = 1.08 \times 10^{-3}$, Data were presented as mean ± SD. Two technical replicates. (e) Fold signal increase of ACh-NeuBI0.5 in response to a panel of neurotransmitters at different concentrations on HEK293A cells. NE norepinephrine. Data were presented as mean ± SEM. Three technical replicates. Source data are available online for this figure.

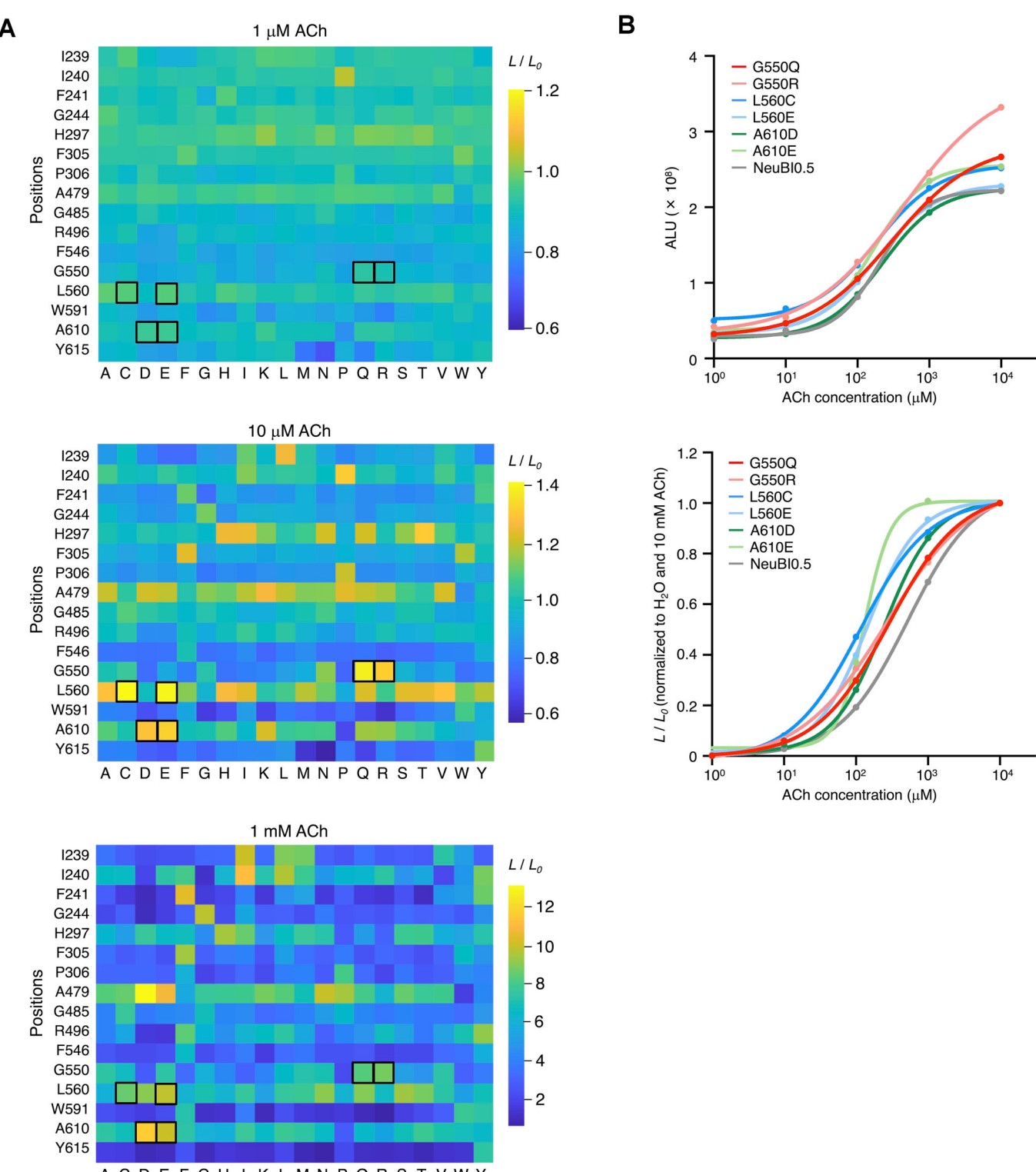

**Figure EV5. MIDAS results from saturation mutagenesis at 16 sites.**

(**A**) Heatmap showing MIDAS results from saturation mutagenesis at 16 sites in HEK293A cells, tested at 1 µM (top), 10 µM (middle), and 1 mM (bottom) ACh. Boxed cells indicate the top-performing mutants selected for further characterization. (**B**) Dose-response curves of the top mutants from the 16-site saturation mutagenesis in HEK293A cells, raw signals (top), or responses normalized to $H_2O$ and 10 mM ACh (bottom). All curves were fitted using the "nonlinear regression (log(inhibitor) vs. response – variable slope)" model in Prism. Data were presented as mean. Three technical replicates. Source data are available online for this figure.

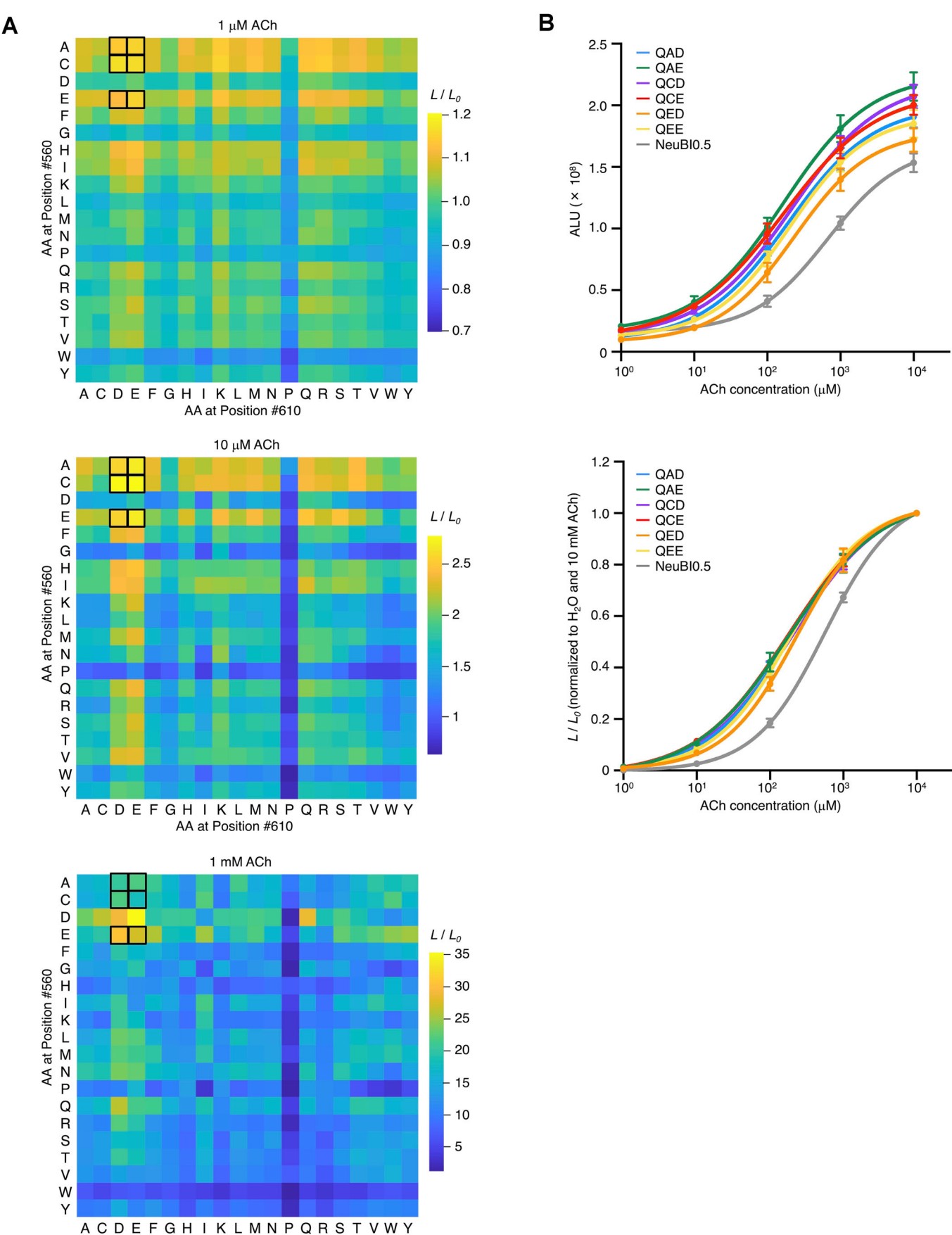

**Figure EV6. MIDAS results from 20 × 20 combinatorial mutagenesis at positions 560 and 610.**

(A) Heatmap showing MIDAS results from 20 × 20 combinatorial mutagenesis at positions 560 and 610 in HEK293A cells, tested at 1 µM (top), 10 µM (middle), and 1 mM (bottom) ACh. Boxed cells indicate the top-performing mutants selected for further characterization. (B) Dose-response curves of the top mutants from the 20 × 20 combinatorial mutagenesis in HEK293A cells, raw signals (top), or responses normalized to $H_2O$ and 10 mM ACh (bottom). All curves were fitted using the "nonlinear regression (log(inhibitor) vs. response – variable slope)" model in Prism. Data were presented as mean ± SD. Three technical replicates.

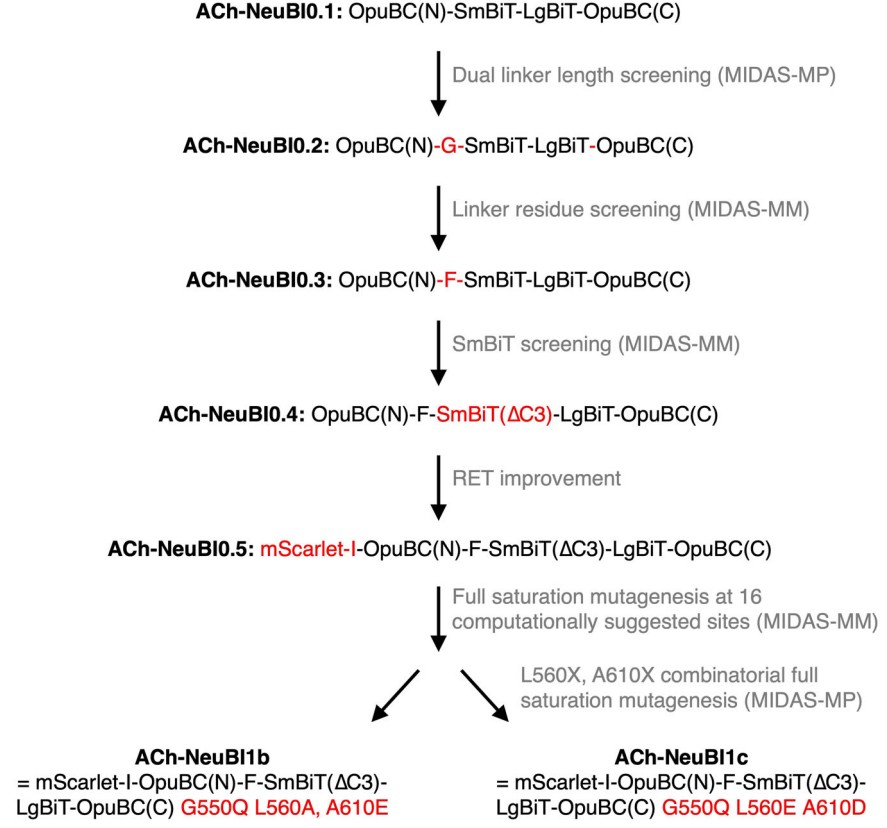

**Figure EV7. Evolutionary history of ACh-NeuBIs.**

The flowchart depicts the six engineering steps leading to ACh-NeuBI1b and ACh-NeuBI1c from ACh-NeuBI0.1. Molecular changes selected after each step are shown in red. Source data are available online for this figure.

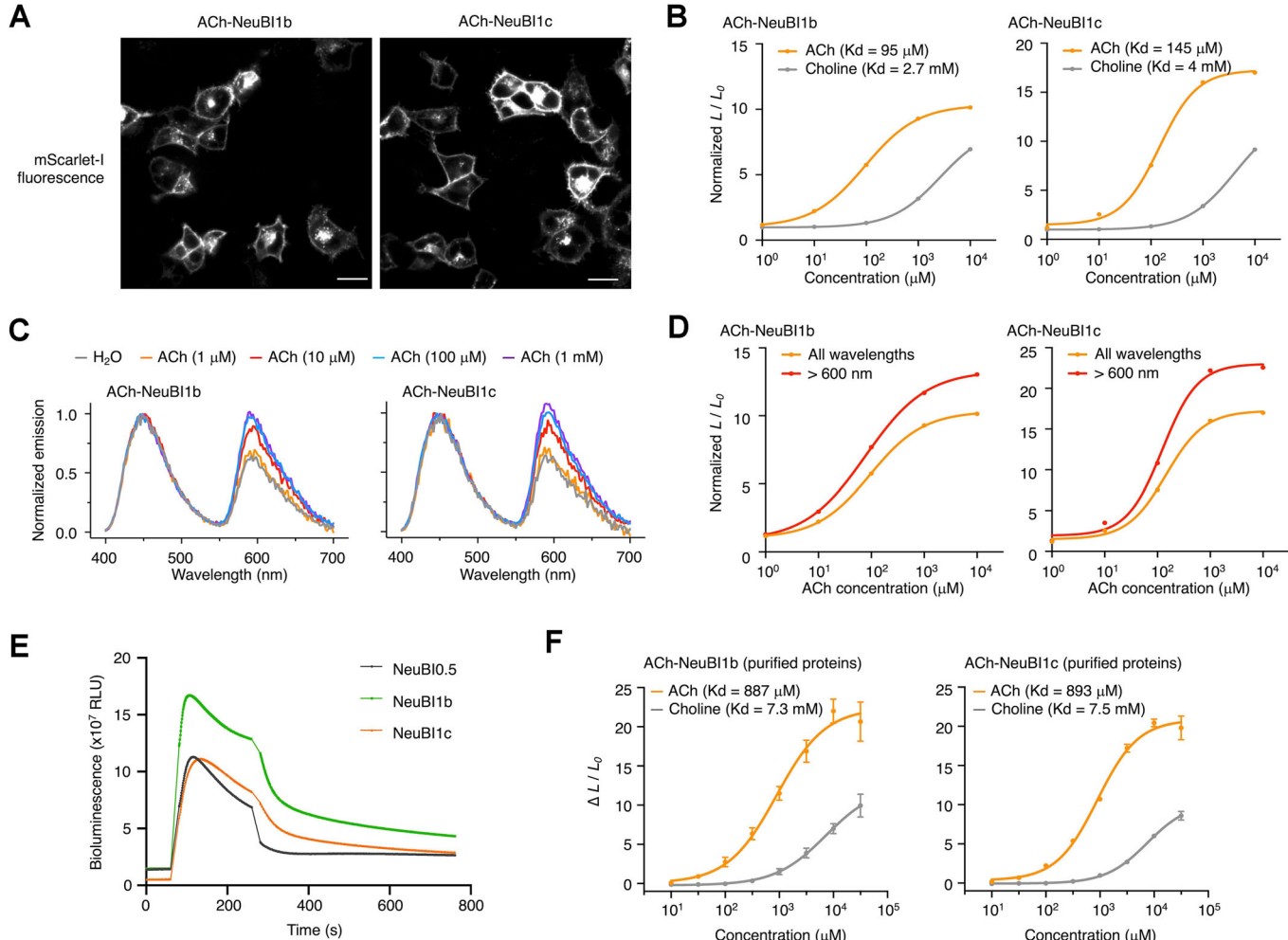

**Figure EV8. Characterization of ACh-NeuBI1b and NeuBI1c.**

(A) Confocal microscopy images of mScarlet-I fluorescence of NeuBI1b (left) and NeuBI1c (right) in HEK293A cells. Scale bar, 20 μm. (B) Fold of signal increase of NeuBI1b (left) and NeuBI1c (right) in response to ACh or choline at different concentrations in HEK293A cells, normalized to the H$_2$O control. (C) Spectra of NeuBI1b (left) and NeuBI1c (right), measured without ACh, or with ACh at various concentrations. (D) Fold of signal increase of NeuBI1b (left) and NeuBI1c (right) in response to ACh at different concentrations in HEK293A cells without a filter (orange line) or with a 610 nm longpass filter (red line). (E) Reversibility of ACh NeuBIs. HEK293A cells expressing individual ACh NeuBIs were imaged under sequential bioluminescence recording. Substrate-containing medium was applied during the initial 0–60 s. A 10 mM ACh solution was then added from 60–240 s to activate the sensors. After 240 s, the medium was replaced with a substrate-containing solution lacking ACh to assess signal reversibility. (F) Dose-dependent response of purified NeuBI1b (left) and NeuBI1c (right) proteins to ACh treatment. (B, D–F) Curves fitted by the "nonlinear regression (log(inhibitor) vs. response – variable slope)" model in Prism. Data were presented as mean ± SD. Three technical replicates. Source data are available online for this figure.

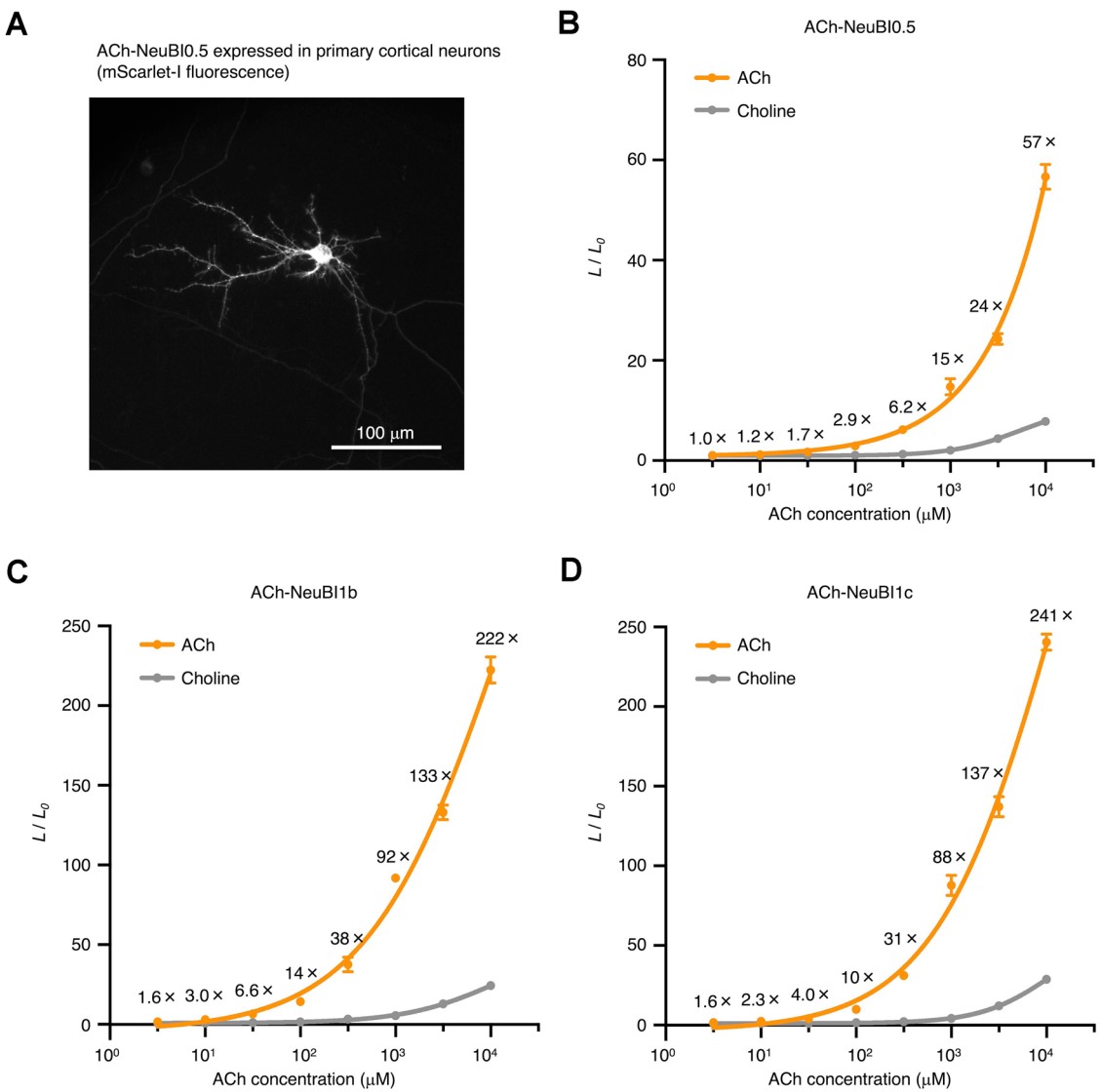

**Figure EV9. Characterization of ACh-NeuBIs in cultured neurons.**

(A) mScarlet-I fluorescence of ACh-NeuBI0.5 in primary cortical neurons. Scale bar, 100 μm. (B–D) Fold of signal increase of NeuBI0.5 (B), NeuBI1b (C), and NeuBI1c (D) in response to ACh or choline at different concentrations in primary cortical neurons. Error bars, SEMs. Curves fitted by the "nonlinear regression (log(inhibitor) vs. response – variable slope)" model in Prism. Three technical replicates. Source data are available online for this figure.

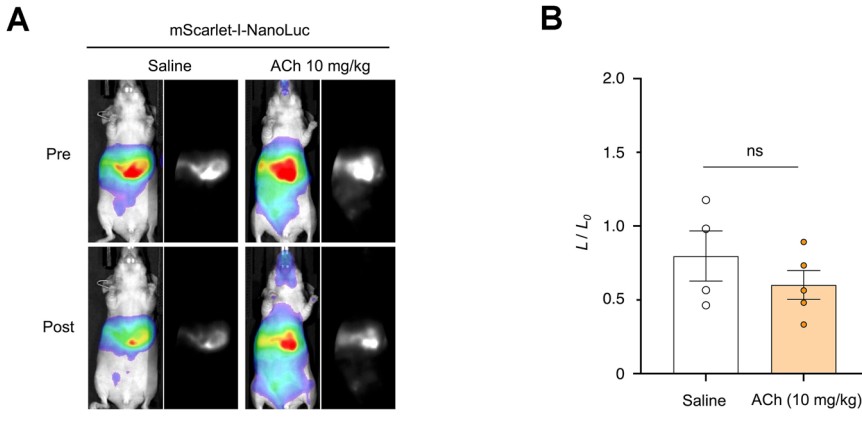

**A** mScarlet-I-NanoLuc

Saline   ACh 10 mg/kg

Pre

Post

0.1 ▮▮▮ 2.0 × 10⁹ p/s/cm²/sr

**B**

$L / L_0$

ns

Saline   ACh (10 mg/kg)

**Figure EV10. ACh does not affect NanoLuc's brightness in vivo.**

(**A**) Representative bioluminescence images acquired before and after the treatment of saline or ACh (10 mg/kg) in mScarlet-I-NanoLuc-expressing mice. (**B**) Fold of signal increase in response to ACh (10 mg/kg), normalized to the saline control. ns not significant, by unpaired two-tailed Student's *t*-test. Data were presented as mean ± SEM. One biological replicate and four-five technical replicates. Source data are available online for this figure.

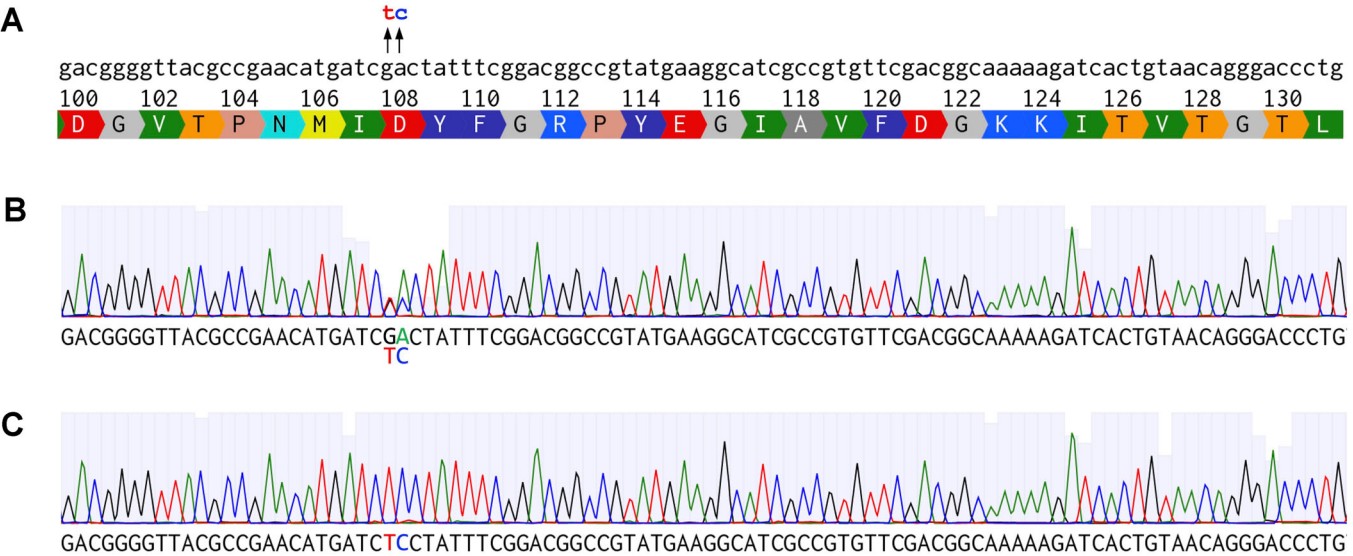

**Figure EV11.  Modified primer design in MIDAS-MM allows primary PCR dilution into the secondary PCR while avoiding plasmid template re-amplification.**

(A) DNA sequence and translated protein sequence surrounding two nucleotides targeted for mutation by MIDAS. (B) Representative sequencing reaction result of the secondary PCR product using flanking primers that also recognize gene elements in the template used for the mutagenic primary PCR. Primary PCRs were performed to generate a left-side invariant portion and an overlapping right-side mutated portion of a CMV-NanoLuc-BGHpA gene, where CMV is the cytomegalovirus promoter and where BGHpA is the bovine growth hormone gene polyadenylation signal. The forward and reverse primers for the left-side PCR were denoted as F1 and R1 primers, and the forward and reverse primers for the right-side PCR were denoted as F2 and R2 primers, and both primary PCRs used the same template plasmid. A 50-µL secondary PCR was then performed with F1 and R2 primers, and with gel-purified left-side fragment (1 µL of a 1 ng/uL eluate) and 1 µL of unpurified right-side PCR added to supply overlapping templates. When combined with direct secondary PCR transfection into mammalian cells, this method is referred to as MIDAS-PM. A representative Sanger sequencing reaction result is shown with the detected bases underneath. The chromatogram was visualized with the program Benchling (Benchling, Inc.), and bases at the mutated positions were colored to match the chromatogram. (C) The same procedure was performed but with two changes: unique sequence extensions were added to the 5' ends of the F1 and R2 primers used in the primary PCRs, and new F1' and R2' primers that primed off the 5' ends of the F1 and R2 primers, respectively, were used for the secondary PCR. When combined with direct secondary PCR transfection into mammalian cells, this protocol for monotemplated monofocal variant generation and expression is referred to as MIDAS-MM. Sanger sequencing detects only the desired GA to TC mutation.

