## [Peer Review File · Molecular Systems Biology]

Fast analysis and engineering of protein function by microbe-independent deep assembly and screening

Yan Wu, Pengli Wang, Lan Xiang Liu, Chao Gao, Daesun Song, Qin Qin, Matt Hageman, Thomas Kirkland, Yichi Su, and Michael Lin

Corresponding author(s): Michael Lin (mzlin@stanford.edu)

Review Timeline:

Submission Date:	11th May 25
Editorial Decision:	18th Jun 25
Revision Received:	20th Jan 26
Editorial Decision:	26th Feb 26
Revision Received:	17th Mar 26
Accepted:	2nd Apr 26

Editor: Poonam Bheda

Transaction Report:

18th Jun 2025

Manuscript Number: MSB-2025-13105-T

Title: Rapid optimization of protein function in mammalian cells via microbe-independent deep assembly and screening

Dear Prof Lin,

Thank you again for submitting your work to Molecular Systems Biology. We have now heard back from the three reviewers who agreed to evaluate your study. As you will see below, the reviewers appreciate that the proposed approach addresses a timely topic. However, they raise a series of concerns, which we would ask you to address in a major revision.

Without repeating all the comments listed below, some of the more fundamental issues raised are the following:

- provide clear comparisons between MIDAS method and other similar methods, including the PCR-based method in your previous Cell paper to highlight the advance of MIDAS
- clarify at what of scale screening the method can achieve
- include a more robust in vivo validation experiment for the final sensor
- show generalizability to another protein/system

All other issues raised would need to be satisfactorily addressed. Please let me know in case you would like to discuss in further detail any of the any of the reviewer comments or your proposed revisions, I would be happy to schedule a call.

We require:

- 1) A .docx formatted version of the manuscript text (including legends for main figures, EV figures and tables). Please make sure that the changes are highlighted to be clearly visible. Alternatively you may choose to submit your manuscript as a LaTeX file.
- 2) Individual production quality figure files as .eps, .tif, .jpg (one file per figure). For guidance, download the 'Figure Guide PDF' (<https://www.embopress.org/page/journal/17574684/authorguide#figureformat>).
- 3) At EMBO Press we ask authors to provide source data for the main figures. Our source data coordinator will contact you to discuss which figure panels we would need source data for and will also provide you with helpful tips on how to upload and organize the files.
- 4) A .docx formatted letter INCLUDING the reviewers' reports and your detailed point-by-point responses to their comments. As part of the EMBO Press transparent editorial process, the point-by-point response is part of the Peer Review File (PRF), which will be published alongside your paper.
- 5) A complete author checklist, which you can download from our author guidelines (<https://www.embopress.org/page/journal/17574684/authorguide#submissionofrevisions>). Please insert information in the checklist that is also reflected in the manuscript. The completed author checklist will also be part of the PRF.
- 6) Please note that all corresponding authors are required to supply an ORCID ID for their name upon submission of a revised manuscript.
- 7) It is mandatory to include a 'Data Availability' section after the Materials and Methods. Before submitting your revision, primary datasets produced in this study need to be deposited in an appropriate public database, and the accession numbers and database listed under 'Data Availability'. Please remember to provide a reviewer password if the datasets are not yet public (see <https://www.embopress.org/page/journal/17574684/authorguide#dataavailability>).

In case you have no data that requires deposition in a public database, please state so in this section as follows: "This study includes no data deposited in external repositories". Note that the Data Availability Section is restricted to new primary data that are part of this study.

8) All Materials and Methods need to be described in the main text using our 'Structured Methods' format, which is required for all research articles. According to this format, the Methods section includes a Reagents and Tools Table (listing key reagents, experimental models, software and relevant equipment and including their sources and relevant identifiers) followed by a Methods and Protocols section describing the methods using a step-by-step protocol format. The aim is to facilitate adoption of the methodologies across labs. Please upload the Reagents and Tools table as a separate document when submitting your

revised manuscript. More information on how to adhere to this format as well as a downloadable template (.docx) for the Reagents and Tools Table can be found in our author guidelines:

<https://www.embopress.org/page/journal/17444292/authorguide#structuredmethods>

9) For data quantification: please specify the name of the statistical test used to generate error bars and p-values, the number (n) of independent experiments (specify technical or biological replicates) underlying each data point and the test used to calculate p-values in each figure legend. The figure legends should contain a basic description of n, p-values and the test applied. Graphs must include a description of the bars and the error bars (s.d., s.e.m.). Please provide exact p-values (in either the figure or figure legend).

10) Our journal encourages inclusion of *data citations in the reference list* to directly cite datasets that were re-used and obtained from public databases. Data citations in the article text are distinct from normal bibliographical citations and should directly link to the database records from which the data can be accessed. In the main text, data citations are formatted as follows: "Data ref: Smith et al, 2001" or "Data ref: NCBI Sequence Read Archive PRJNA342805, 2017". In the Reference list, data citations must be labeled with "[DATASET]". A data reference must provide the database name, accession number/identifiers and a resolvable link to the landing page from which the data can be accessed at the end of the reference. Further instructions are available at .

11) We replaced Supplementary Information with Expanded View (EV) Figures and Tables that are collapsible/expandable online. EV Figures should be cited as 'Figure EV1, Figure EV2' etc... in the text and their respective legends should be included in the main text after the legends of regular figures.

- Additional Tables/Datasets should be labeled and referred to as Table EV1, Dataset EV1, etc. Legends should be provided in a separate tab in case of .xls files. Alternatively, the legend can be supplied as a separate text file (README) and zipped together with the Table/Dataset file.

<https://www.embopress.org/page/journal/17574684/authorguide#expandedview>

12) Author contributions: CRedit has replaced the traditional author contributions section because it offers a systematic machine-readable author contributions format that allows for more effective research assessment. Please remove the Authors Contributions from the manuscript and use the free text boxes beneath each contributing author's name in our system to add specific details on the author's contribution. More information is available in our guide to authors.

13) Disclosure statement and competing interests: We updated our journal's competing interests policy in January 2022 and request authors to consider both actual and perceived competing interests. Please review the policy

<https://www.embopress.org/competing-interests> and update your competing interests if necessary.

14) Every published paper now includes a 'Synopsis' to further enhance discoverability. Synopses are displayed on the journal webpage and are freely accessible to all readers. They include a short stand first (maximum of 300 characters, including space) as well as 2-5 one-sentences bullet points that summarizes the paper. Please write the bullet points to summarize the key NEW findings. They should be designed to be complementary to the abstract - i.e. not repeat the same text. We encourage inclusion of key acronyms and quantitative information (maximum of 30 words / bullet point). Please use the passive voice. Please attach these in a separate file or send them by email, we will incorporate them accordingly.

Please note that these would be the final versions and changes during proofing are usually not allowed.

15) As part of the EMBO Publications transparent editorial process initiative (see our policy here:

https://www.embopress.org/transparent-process#Review_Process), Molecular Systems Biology will publish online a Peer Review File (PRF) to accompany accepted manuscripts.

In the event of acceptance, this file will be published in conjunction with your paper and will include the anonymous referee reports, your point-by-point response and all pertinent correspondence relating to the manuscript. Let us know whether you agree with the publication of the PRF and as here, if you want to remove or not any figures from it prior to publication.

Please note that the Author checklist will be published at the end of the PRF.

Molecular Systems Biology has a "scooping protection" policy, whereby similar findings that are published by others during review or revision are not a criterion for rejection. Should you decide to submit a revised version, I do ask that you get in touch after three months if you have not completed it, to update us on the status.

Yours sincerely,

Poonam Bheda, PhD
Scientific Editor
Molecular Systems Biology

Reviewer #1:

The manuscript by Wu and colleagues presents a method for directed evolution and selection of proteins, which is fast and relatively low cost. Thus, it could potentially appeal to a number of labs who aim at smaller protein engineering projects. It is based on generating diversity by PCR and then directly transfecting DNA amplicons into mammalian cells, thus omitting any intermediate step that would involve growing and purifying DNA in and from bacteria. Overall, this could add a useful new facet to facilitate directed evolution projects and to interrogate functional consequences of diversification of limited numbers of selected residues.

Several issues still need to be clarified.

- It would be easier for the reader if the authors decided to include a scheme of the evolution, with the most important variants in each evolutionary step.
- There appear to be some inconsistencies with nomenclature: Orange Ach NeuBI, Ach NeuBI, Ach-NeuBI, NeuBI. Please revise one more time.
- responsiveness, specificity, and reversibility of optimized Ach-NeuBIs: reversibility should be better documented
- How well could PCR amplicons be retrieved and sequenced after evaluation of cells? If this is difficult, it should be clarified because it would mean that tracking all clones in the library throughout the project is an absolute requirement.

Reviewer #2:

In this manuscript, Wu et al. present Microbe-Independent Deep Assembly and Screening (MIDAS), a rapid, high-throughput method for optimizing protein function directly in mammalian cells. MIDAS integrates three streamlined steps: (1) multi-well-plate PCR to construct libraries of all possible variants at targeted sites, (2) direct transfection of unpurified linear PCR products into mammalian cells, and (3) cell-based phenotypic screening. Moreover, the approach enables deep sampling of machine learning-identified mutational hotspots by generating and screening each allelic variant uniquely, maximizing efficiency. Using MIDAS, the authors substantially enhanced the responsiveness, specificity, and dynamic range of AChNeuBI via iterative mutagenesis.

In summary, MIDAS circumvents cloning procedures, reducing the timeline from primer design to high-content assays from weeks to one day. While this work presents an intriguing advancement, several issues should be addressed prior to being considered for publication in Molecular Systems Biology. The following comments are given to strengthen the manuscript:

1. In Figure 1b, does each PCR product amplified by primers F2 and R2 contain only a single variant? If so, this approach would become prohibitively labor-intensive when screening mutant libraries comprising thousands of variants. Under such circumstances, MIDAS would seemingly offer limited advantages over conventional methods.
2. While the authors demonstrate MIDAS-enhanced performance of AChNeuBI, can this platform be applied to evolve other proteins (e.g., Cas9 or recombinases)? Crucially, any novel methodology must demonstrate potential generalizability and broad applicability to ensure widespread adoption by the research community.
3. In Figure 1d, the authors identify a linker length of 1 and 0 amino acids at linker positions 1 and 2, respectively, as optimal for connecting cpNanoLuc and OpuBC fragments. However, this conclusion cannot be considered definitive since they exclusively tested glycine residues. Given that alternative amino acid substitutions may yield differential effects (as illustrated in Figure 1e), comprehensive evaluation of diverse amino acid combinations would be essential to establish any conclusions.
4. Although the authors assert MIDAS offers enhanced safety over fluorescence-based methods, the absence of long-term stability data for AChNeuBI remains unaddressed. Would it be feasible to evaluate either *in vivo* expression toxicity or photostability of AChNeuBI to strengthen this claim and advance its clinical translational potential?
5. The optimized ACh-NeuBI successfully detected exogenous ACh noninvasively *in vivo*, with up to 100-fold responses. Would it be feasible to detect endogenous Ach *in vivo*?

Minor revisions:

6. The images (Figures S3a and S6a) are rather blurry and do not seem to support the conclusions. Can you provide clearer confocal microscopy images?
7. The term "H2O" in the legend of Figure S2 should be "H₂O".
8. The format of the references needs significant modification to ensure consistency, such as standardizing journal names and paper titles.

Reviewer #3:

The authors present MIDAS (Microbe-Independent Deep Assembly and Screening), a PCR-only platform that assembles linear gene constructs via overlap-extension in multiwell PCRs and directly uses them to transfect mammalian cells, bypassing traditional microbial cloning. They apply MIDAS to engineer a bioluminescent acetylcholine sensor by fusing OpuBC to split NanoLuc fragments, followed by systematic optimization of linkers, point mutations, and combinatorial variants to improve responsivity from ~22% to 2-3× signal increases (apparent k_d ~95 μ M). The final sensor variants, incorporating fluorescent proteins for red-shifting, exhibit significantly enhanced dynamic range and sensitivity, which are validated through robust bioluminescence expression in the liver following hydrodynamic tail-vein injection in mice.

Overall, this is a well-executed study of interest to researchers in protein engineering and neurotransmitter sensing. However, the novelty of MIDAS is somewhat overstated. The core concept-using PCR-assembled constructs with promoters and polyA signals for transient expression in mammalian cells-was previously demonstrated by the authors in their 2020 Cell paper. MIDAS represents a logical continuation and refinement of that earlier approach rather than a fundamentally new method. The authors should explicitly acknowledge the 2020 Cell paper and provide a clear comparison outlining similarities and differences between the two methods.

Major Concerns

1. In vivo Validation:

While the mouse experiments provide useful proof-of-concept, they rely exclusively on exogenous acetylcholine delivered by tail-vein injection. This setup does not mimic physiological cholinergic signaling, which is highly localized and temporally dynamic. To more convincingly demonstrate in vivo utility, the authors should include experiments that assess endogenous cholinergic activity-such as via optogenetic or chemogenetic stimulation of cholinergic neurons-to validate sensor performance under biologically relevant conditions.

2. Screening Scale and Throughput:

MIDAS is described as "rapid" and "high-throughput," but the demonstrated scale (hundreds of variants via multiwell PCR) is relatively modest. The authors should change the terms and clarify how MIDAS compares to other screening methods. What is the practical upper limit of MIDAS throughput (e.g., per 96- or 384-well plate, or per day)? Can the approach be scaled to larger libraries through pooled strategies? A discussion of trade-offs in coverage, labor, cost, and speed would enhance the utility of this method for broader adoption. Also, a time- or cost-based comparison between generating ~100 variants via MIDAS versus other methods would be particularly informative.

3. Normalization and Quantitative Comparisons:

Because MIDAS relies on transient transfection of linear PCR products, expression levels may vary across wells or constructs, potentially confounding functional comparisons. The authors should validate the expression variation, using co-transfection with a fluorescent reporter (e.g., mCherry) or a dual-luciferase system. This would be important information if the readers want to adapt the technology in their research.

4. Generality of the Method:

The study would be strengthened by demonstrating MIDAS on a second target protein or biosensor, even at a smaller scale, to establish its broader applicability. If such demonstrations are not feasible, the authors should clearly articulate MIDAS's current limitations. For example, MIDAS requires robust expression of PCR constructs and compatible optical assays, and its effectiveness depends on prior knowledge of functional sites for targeted mutagenesis. In cases lacking structural or functional information, alternative approaches such as random mutagenesis might be more suitable.

Minor Points

- The acronym MIDAS is defined both as "Microbial-Independent" and "Microbe-Independent." The authors should choose one and use it consistently throughout the manuscript.
- The meaning of " Δ C3" in reference to the SmBiT component should be clarified.
- Figure S3f: The K_D value is incorrectly listed as 898 mM; it should be 898 μ M.
- Figure 2b: Since variants with C, E, Q, T, and V at position 560 all outperform alanine, what is the rationale for using A in the final QAE variant?
- Units: Replace all instances of "uM" with the correct symbol " μ M."
- Figures S3f and S3g: If multiple neurotransmitters were tested, the x-axis label should be "Neurotransmitter concentration," not "ACh concentration."

- Figure S4b: Mutation A610E is referenced in the text, but the figure label shows A610?. This should be corrected for consistency.

In summary, this paper provides a useful demonstration of direct protein optimization in mammalian cells using PCR-assembled constructs. However, I suggest the authors better contextualize MIDAS in light of their prior work, improve the biological relevance of in vivo validation, and provide a clearer discussion of method scalability and generalizability.

Response to reviews for “Rapid optimization of protein function in mammalian cells via microbe-independent deep assembly and screening” (now titled “Fast analysis and engineering of protein function by microbe-independent deep assembly and screening”)

We are grateful to all the referees and editors for their careful reading of our manuscript, insightful comments, and helpful suggestions. We believe we thoroughly addressed all the concerns by adding more detailed explanations and additional experimental results. Our point-by-point responses are below in blue. New text in the revised manuscript is also shown in blue.

Reviewer #1:

The manuscript by Wu and colleagues presents a method for directed evolution and selection of proteins, which is fast and relatively low cost. Thus, it could potentially appeal to a number of labs who aim at smaller protein engineering projects. It is based on generating diversity by PCR and then directly transfecting DNA amplicons into mammalian cells, thus omitting any intermediate step that would involve growing and purifying DNA in and from bacteria. Overall, this could add a useful new facet to facilitate directed evolution projects and to interrogate functional consequences of diversification of limited numbers of selected residues.

We thank the reviewer for the positive and encouraging feedback on our manuscript. We greatly appreciate the thoughtful insights and the questions raised which have allowed us to significantly strengthen the manuscript.

Several issues still need to be clarified.

1. It would be easier for the reader if the authors decided to include a scheme of the evolution, with the most important variants in each evolutionary step.

We agree and have added a schematic summarizing the evolutionary trajectory of ACh-NeuBI, highlighting the key variants selected at each optimization step. This is now included as **Figure S6**, which presents a clear flowchart from ACh-NeuBI0.1 to ACh-NeuBI1b and ACh-NeuBI1c with the corresponding molecular changes.

2. There appear to be some inconsistencies with nomenclature: Orange Ach NeuBI, Ach NeuBI, Ach-NeuBI, NeuBI. Please revise one more time.

Thank you for pointing this out. We have carefully revised the manuscript to ensure consistent use of the name “ACh-NeuBI” throughout the text and figures.

3. Responsivity, specificity, and reversibility of optimized Ach-NeuBIs: reversibility should be better documented

We agree and have expanded our analysis of reversibility. We now include quantitative kinetic measurements showing signal decay following ACh washout for ACh-NeuBI0.5, ACh-NeuBI1b, and ACh-NeuBI1c. These data are presented in **Figure S7e** and described in the Results section. While higher-affinity variants exhibit slightly slower reversibility, all sensors demonstrate robust signal recovery on physiologically relevant timescales.

4. How well could PCR amplicons be retrieved and sequenced after evaluation of cells? If this is difficult, it should be clarified because it would mean that tracking all clones in the library throughout the project is an absolute requirement.

In MIDAS, variant identity is deterministically tracked by well position, rather than by post-hoc recovery and sequencing of PCR products. This design ensures a strict one-to-one mapping between sequence and function throughout screening. We have clarified this point in the Results, emphasizing that sequencing of PCR amplicons after screening is not required for tracking variants and that this well-based mapping is an intentional feature of the platform: “In the current MIDAS workflow, each PCR amplicon encodes a defined variant and is transfected into a separate well, allowing variant identity to be tracked by well position, which ensures accurate association between sequence and function throughout the screen.”

Reviewer #2:

In this manuscript, Wu et al. present Microbe-Independent Deep Assembly and Screening (MIDAS), a rapid, high-throughput method for optimizing protein function directly in mammalian cells. MIDAS integrates three streamlined steps: (1) multi-well-plate PCR to construct libraries of all possible variants at targeted sites, (2) direct transfection of unpurified linear PCR products into mammalian cells, and (3) cell-based phenotypic screening. Moreover, the approach enables deep sampling of machine learning-identified mutational hotspots by generating and screening each allelic variant uniquely, maximizing efficiency. Using MIDAS, the authors substantially enhanced the responsiveness, specificity, and dynamic range of AChNeuBI via iterative mutagenesis.

In summary, MIDAS circumvents cloning procedures, reducing the timeline from primer design to high-content assays from weeks to one day. While this work presents an intriguing advancement, several issues should be addressed prior to being considered for publication in *Molecular Systems Biology*. The following comments are given to strengthen the manuscript:

We thank the reviewer for carefully reading our manuscript and for the positive comments and useful suggestions.

1. In Figure 1b, does each PCR product amplified by primers F2 and R2 contain only a single variant? If so, this approach would become prohibitively labor-intensive when screening mutant libraries comprising thousands of variants. Under such circumstances, MIDAS would seemingly offer limited advantages over conventional methods.

We appreciate the reviewer's comment. As noted, each PCR product in this MIDAS workflow contains a single defined variant and is transfected individually. This format is not intended to replace pooled ultra-high-throughput methods, but instead is intended for medium-scale, targeted mutagenesis or structure-function mapping efforts, such as saturation mutagenesis at specific sites (including possibly all sites within a small protein) or systematic exploration of machine learning-predicted hotspots, where the performance of each variant is of interest and individual tracking is essential. We now cite several examples of groups cloning more than 800 variants to systematically evaluate the effects of mutations; MIDAS would allow the same types of studies at much lower cost and more rapidly.

Our results also demonstrate that MIDAS is not labor-intensive compared to alternative ways to perform true saturation mutagenesis. We should clarify that the two PCR steps are performed from oligos ordered in multiwell plates and pipetted into multiwell blocks using multichannel pipettors, and the resulting reactions are also transfected into cells in multiwell plates using multiwell pipettors. In our hands, with 384-well plates and multi-channel pipetting or automation, thousands of variants can be screened using just a few plates in parallel, which is practical and efficient. Thus the arrayed nature of MIDAS and the very few steps involved make it labor-efficient, as evidenced by the many improvement steps we performed to create improved proteins, including our added example of NanoLuc improvement.

In contrast, generating and validating thousands of plasmids by conventional cloning would be orders of magnitude more time- and resource-intensive. Specifically, plasmid cloning requires many sequential steps through tubes, more tubes, petri dishes, tubes, columns, and tubes again, executed over many days. MIDAS avoids this major bottleneck by bypassing bacterial cloning to enable rapid and microbe-independent screening directly in mammalian cells. Thus when plasmids need to be prepared, e.g. for deterministic assessment of mutational performance, MIDAS offers large advantages in time and cost.

MIDAS also offers advantages compared to methods where mutagenesis is performed "blind" and only the best-performing mutants are retrieved for identification. Each PCR product in MIDAS encodes a single defined variant, enabling deterministic and complete interrogation of targeted sequence space, which is often infeasible with random or pooled approaches in mammalian cells. In our study, most beneficial

mutations required multiple nucleotide changes and would be exceedingly unlikely to be recovered by error-prone PCR or pooled screening at practical library sizes. MIDAS ensures exact coverage of all desired variants with substantially lower noise by using arrayed, multiwell functional assays rather than noisy single-cell fluorescence-based methods such as FACS.

We have added the following in the discussion:

- “While MIDAS does not reach the ultra-high-throughput scale of pooled screening approaches, it allows the performance of all generated mutations to be assessed at a rate of thousands per day, making it a good fit for the testing of computationally predicted beneficial mutations.”
- “In our experience, generating 384 variants via MIDAS requires only ~4 hours of hands-on time and \$2000 in reagent costs. In contrast, an experienced researcher working in parallel on 24 samples at a time to clone the same number of variants would require 192 hours of time and ~\$20,000 in reagent costs. Thus, MIDAS is ~48 times faster and ~10 times less expensive than cloning-based approaches. While MIDAS does not reach the ultra-high-throughput scale of pooled screening approaches, it provides at least an order of magnitude improvement in time and expense for high-quality screening or characterization, making MIDAS especially useful for researchers who seek to explore specific sequence regions quickly, accurately, and cost-effectively.”
- “The deterministic construction of mutants at sites of interest, as performed in MIDAS, also offers a solution to the inefficiency of error-prone PCR in generating functional mutants with more than one nucleotide change. Six of the seven individual beneficial amino acid changes discovered in this study required two or three nucleotide mutations. Based on published measurements, functional clones with two mutations generated by whole-gene error-prone PCR would occur once in $> 7.1 \times 10^7$ clones, which can only be screened by noisy single-cell fluorescence-based methods such as FACS. If beneficial sites for mutation can be identified, then deterministic mutagenesis allows all possibilities to be generated and assayed in 20 wells for one site or 400 wells for two sites, allowing for multiwell-based measurements of any type and with less noise than FACS. Once sites for targeted mutagenesis are identified, MIDAS then tests them more quickly and efficiently than cloning with degenerate oligonucleotides.”

2. While the authors demonstrate MIDAS-enhanced performance of AChNeuBI, can this platform be applied to evolve other proteins (e.g., Cas9 or recombinases)? Crucially, any novel methodology must demonstrate potential generalizability and broad applicability to ensure widespread adoption by the research community.

We agree with the reviewer that generalizability is important. To directly demonstrate generalizability beyond biosensors, we added a second application: deep mutational scanning and optimization of NanoLuc luciferase activity toward three substrates using the MIDAS-MM protocol. These results are presented in **Figures 5 and 6** and show that MIDAS can improve enzyme catalysis, characterize mutational tolerance, and identify substrate-specific variants. This establishes that MIDAS generalizes to distinct protein classes and assay types (biosensors and enzymes), beyond a single target, and can be used for structure-function analyses, not only protein engineering.

3. In Figure 1d, the authors identify a linker length of 1 and 0 amino acids at linker positions 1 and 2, respectively, as optimal for connecting cpNanoLuc and OpuBC fragments. However, this conclusion cannot be considered definitive since they exclusively tested glycine residues. Given that alternative amino acid substitutions may yield differential effects (as illustrated in Figure 1e), comprehensive evaluation of diverse amino acid combinations would be essential to establish any conclusions.

We agree that the linker length screening in Fig. 1d (now Fig. 2d) was limited to glycine residues and does not establish a globally optimal configuration. Our goal at this stage was not to identify a universal optimum, but to improve the initial prototype in a practical and efficient manner. The sequential approach we used here, first optimizing length using glycine, a flexible and neutral residue, then refining composition in Fig. 1e (now Fig. 2e), was designed to balance screening effort and functional gains. A comprehensive screen

of all possible amino acid combinations across two linkers (let's say each up to 5 residues long) would theoretically require screening over 10^{13} variants ($20^{10} + 20^9 + 20^8 \dots + 20^1$), which is impractical.

We have changed the description of ACh-NeuBI0.2 from “this optimized variant” to simply “this variant” to avoid confusion about the degree of optimization. In agreement with the reviewer's point that linker sequence matters, our next experiment was to find the optimal amino acid at that single position, leading to Phe as already described in **Figure 2e**.

4. Although the authors assert MIDAS offers enhanced safety over fluorescence-based methods, the absence of long-term stability data for AChNeuBI remains unaddressed. Would it be feasible to evaluate either in vivo expression toxicity or photostability of AChNeuBI to strengthen this claim and advance its clinical translational potential?

We assume the reviewer means to ask about AChNeuBI rather than MIDAS, since MIDAS isn't an imaging method and thus can't be compared to fluorescence. Regarding toxicity from long-term expression, we note that related components have been safely used in vivo: for example, CaMBI (Su et al. doi: 10.1038/s41589-023-01265-x) and KiMBI (Wu et al. doi: 10.1038/s41589-025-01846-y), which are based on NanoLuc, have been expressed in mice safely, and PBP scaffold used here has also been successfully expressed in the brain (Borden et al. doi: 10.1101/2020.02.07.939504). Luciferases in general have been used for long-term tracking of cell populations without issue. There can be immune-based rejection, just as with fluorescent proteins, but this can be alleviated by the use of transgenic or immunodeficient mice. These are common issues for all proteins, and there is no reason to raise a concern for bioluminescent reporters as compared to fluorescent reporters. As for photostability, this is not a concern as NanoLuc operates via glow-type bioluminescence without the need for external excitation light, and the light output is orders of magnitude lower in energy density than the excitation light used in fluorescence.

5. The optimized ACh-NeuBI successfully detected exogenous ACh noninvasively in vivo, with up to 100-fold responses. Would it be feasible to detect endogenous ACh in vivo?

We thank the reviewer for this important question. Detecting endogenous ACh in vivo is indeed a compelling goal, and we agree that it would further demonstrate the sensor's utility. However, the main focus of this work is the development and validation of MIDAS as a rapid, cloning-free method for protein optimization in mammalian cells. The in vivo experiments in this study were designed to demonstrate that MIDAS-optimized sensor variants function robustly in a living animal context, as evidenced by dose-dependent, noninvasive detection of exogenous ACh with up to 100-fold signal increases.

Based on the EC_{50} values of the optimized ACh-NeuBI variants (~200 μ M) and the known physiological range of extracellular ACh concentrations [up to 1–10 mM at the synaptic clefts of neuromuscular junctions (Smart et al. doi: 10.1016/S0006-3495(98)77610-6) and the release sites of striatal cholinergic interneurons (Matityahu et al. doi: 10.1038/s41467-023-42311-5)], we expect the sensor can detect endogenous ACh in vivo under appropriate stimulation and expression conditions. However, such experiments would require additional optimization of targeting and experimental models, which are beyond the scope of this methods-focused manuscript.

Minor revisions:

6. The images (Figures S3a and S6a) are rather blurry and do not seem to support the conclusions. Can you provide clearer confocal microscopy images?

We appreciate the reviewer's feedback. We have replaced the images in **Figure S3a** and **S7a** with clearer confocal microscopy images to better illustrate membrane targeting, added “confocal microscopy images” in the corresponding legend, and added “confocal microscopy” in the Methods session.

Fig S3a:

ACh-NeuBI0.5 in HEK293A cells
(mScarlet-I fluorescence)

Fig. S7a:

ACh-NeuBI1b in HEK293A cells
(mScarlet-I fluorescence)

ACh-NeuBI1c in HEK293A cells
(mScarlet-I fluorescence)

7. The term "H2O" in the legend of Figure S2 should be "H₂O".

We have corrected the formatting of “H2O” to “H₂O” in the legend of Figure S2 in the revised manuscript.

8. The format of the references needs significant modification to ensure consistency, such as standardizing journal names and paper titles.

We appreciate the reviewer’s suggestion. We have carefully revised the reference list to ensure consistency in formatting, including standardization of journal names, article titles, and citation style in accordance with the journal’s guidelines.

Reviewer #3:

The authors present MIDAS (Microbe-Independent Deep Assembly and Screening), a PCR-only platform that assembles linear gene constructs via overlap-extension in multiwell PCRs and directly uses them to transfect mammalian cells, bypassing traditional microbial cloning. They apply MIDAS to engineer a bioluminescent acetylcholine sensor by fusing OpuBC to split NanoLuc fragments, followed by systematic optimization of linkers, point mutations, and combinatorial variants to improve responsivity from ~22% to 2-3× signal increases (apparent k_d ~95 μ M). The final sensor variants, incorporating fluorescent proteins for red-shifting, exhibit significantly enhanced dynamic range and sensitivity, which are validated through robust bioluminescence expression in the liver following hydrodynamic tail-vein injection in mice.

Overall, this is a well-executed study of interest to researchers in protein engineering and neurotransmitter sensing. However, the novelty of MIDAS is somewhat overstated. The core concept-using PCR-assembled constructs with promoters and polyA signals for transient expression in mammalian cells-was previously demonstrated by the authors in their 2020 Cell paper. MIDAS represents a logical continuation and refinement of that earlier approach rather than a fundamentally new method. The authors should explicitly acknowledge the 2020 Cell paper and provide a clear comparison outlining similarities and differences between the two methods.

We appreciate the reviewer's positive and encouraging feedback on our manuscript and suggestion to clarify the novelty of MIDAS in relation to our prior work (Villette et al., Cell, 2020). While in our previous work, we have showed that PCR-assembled fragments could be used to optimize to optimize one position or two adjacent positions already known to be important for protein function, MIDAS builds on this foundation and expands the use of this method in the following ways:

- Polyfocal and combinatorial mutagenesis
- Monotemplated MIDAS-MM to avoid plasmid contamination
- Enzyme optimization and structure-function mapping

We added a new **Figure 1** to describe in detail the new methods introduced in this study.

We have added the following to the discussion: "In previous work, we showed that PCR assembly of gene chimeras can be used to generate and screen mutations at one location for improved function¹⁷. The current study extends these results to additional mutational types and patterns. Specifically, we generalize the MIDAS method to multiple types of protein modifications not previously performed by PCR transfection: (1) deep screening of multiple sites throughout a functional surface of the protein as demonstrated by improving NanoLuc catalytic activity, (2) combinatorial linker length optimization, (3) testing of computational predictions for higher-affinity ligand binding; and (4) deep combinatorial mutagenesis of multiple sites located in discontinuous portions of the primary sequence. In addition, we introduce the modified MIDAS-MM protocol where the use of nested primers avoids re-amplification of parental plasmid in primary PCR reactions, allowing microbe-independent deep assembly and screening to be immediately applied to optimizing a protein of interest without requiring it to be first subcloned into multiple plasmids with different priming sites."

Major Concerns

1. In vivo Validation:

While the mouse experiments provide useful proof-of-concept, they rely exclusively on exogenous acetylcholine delivered by tail-vein injection. This setup does not mimic physiological cholinergic signaling, which is highly localized and temporally dynamic. To more convincingly demonstrate in vivo utility, the authors should include experiments that assess endogenous cholinergic activity-such as via optogenetic or chemogenetic stimulation of cholinergic neurons-to validate sensor performance under biologically relevant conditions.

We thank the reviewer for this important question. Detecting endogenous ACh *in vivo* is indeed a compelling goal, and we agree that it would further demonstrate the sensor's utility. However, the main focus of this work is the development and validation of MIDAS as a rapid, cloning-free method for protein optimization in mammalian cells. The *in vivo* experiments in this study were designed to demonstrate that MIDAS-optimized sensor variants function robustly in a living animal context, as evidenced by dose-dependent, noninvasive detection of exogenous ACh with up to 100-fold signal increases.

Based on the EC_{50} values of the optimized ACh-NeuBI variants (~200 μ M) and the known physiological range of extracellular ACh concentrations [up to 1–10 mM at the synaptic clefts of neuromuscular junctions (Smart et al. doi: 10.1016/S0006-3495(98)77610-6) and the release sites of striatal cholinergic interneurons (Matityahu et al. doi: 10.1038/s41467-023-42311-5)], we expect the sensor can detect endogenous ACh *in vivo* under appropriate stimulation and expression conditions. However, such experiments would require additional optimization of targeting and experimental models, which are beyond the scope of this methods-focused manuscript.

2. Screening Scale and Throughput:

MIDAS is described as "rapid" and "high-throughput," but the demonstrated scale (hundreds of variants via multiwell PCR) is relatively modest. The authors should change the terms and clarify how MIDAS compares to other screening methods. What is the practical upper limit of MIDAS throughput (e.g., per 96- or 384-well plate, or per day)? Can the approach be scaled to larger libraries through pooled strategies? A discussion of trade-offs in coverage, labor, cost, and speed would enhance the utility of this method for broader adoption. Also, a time- or cost-based comparison between generating ~100 variants via MIDAS versus other methods would be particularly informative.

We appreciate the reviewer's comment. We agree that while MIDAS is not designed to match the ultra-high-throughput capacity of pooled library approaches, it offers a practical, rapid, and cost-effective solution for medium-scale, targeted mutagenesis efforts, such as saturation mutagenesis at specific sites or systematic exploration of machine learning-predicted hotspots, where the performance of each variant is of interest and individual tracking is important. We have now revised manuscript and adjusted our language accordingly to avoid claiming MIDAS as "high-throughput". The current MIDAS flow can screen hundreds to thousands of variants in parallel using 384-well plates and multi-channel pipetting or automation. In our hands, a single researcher can practically generate and test up to ~1500 variants (four 384-well plates) in 2-3 days from primer delivery to data collection, without any bacterial cloning steps.

While the current implementation of MIDAS is well suited for arrayed formats where individual variant identity is tracked by well position, we agree that adapting MIDAS to pooled screening is an exciting future direction. In principle, MIDAS could be scaled to larger libraries by combining amplicons into pools and linking phenotypic output to genotype via sequencing. However, this would require additional optimization, such as efficient recovery of linear DNA from cells, barcode integration, and targeted amplification strategies to ensure accurate variant identification. We have noted this potential and its associated challenges in the revised discussion: "In principle, MIDAS could be adapted to pooled library screens by linking phenotype to genotype through barcoding or amplicon sequencing. This would require additional development to enable efficient DNA recovery and variant tracking, which would be an exciting avenue for future work."

We have now added a discussion of the trade-offs in coverage, labor, cost, and speed of MIDAS in the revised manuscript. "While MIDAS does not reach the ultra-high-throughput scale of pooled screening approaches, it offers a compelling balance of throughput, cost, and hands-on labor for medium-scale, targeted mutagenesis efforts, such as saturation mutagenesis at specific sites or systematic exploration of machine learning-predicted hotspots, where the performance of each variant is of interest and individual tracking is essential."

Regarding time and cost, we have added the following paragraph to the discussion: "We further consider the degree to which MIDAS accelerates protein engineering or mutational analysis. Traditionally,

performing these procedures in mammalian cells involves plasmid cloning. For example, to perform a deep sequence-function analysis of esterase function, more than 1000 variants were manually cloned, purified, and transfected into mammalian cells⁴⁶. Similarly, more than 800 plasmids were cloned, purified, and transfected into mammalian cells to develop jGCaMP8 calcium indicators¹². Cloning involves at a minimum PCR, vector digestion, transformation, colony picking, microbial culture, plasmid purification, and sequence verification. In contrast, MIDAS substitutes the entire cloning procedure with simply two PCR steps. In our experience, generating 384 variants via MIDAS requires only ~4 hours of hands-on time and \$2000 in reagent costs. In contrast, an experienced researcher working in parallel on 24 samples at a time to clone the same number of variants would require 192 hours of time and ~\$20,000 in reagent costs. Thus, MIDAS is ~48 times faster and ~10 times less expensive than cloning-based approaches.”

3. Normalization and Quantitative Comparisons:

Because MIDAS relies on transient transfection of linear PCR products, expression levels may vary across wells or constructs, potentially confounding functional comparisons. The authors should validate the expression variation, using co-transfection with a fluorescent reporter (e.g., mCherry) or a dual-luciferase system. This would be important information if the readers want to adapt the technology in their research.

We thank the reviewer for this thoughtful suggestion. In the ACh-NeuBI optimization described in this manuscript, we were primarily interested in fold change or affinity (or EC50), in which case we can obtain our improvement of interest by normalizing to baseline or peak luminescence after sigmoidal fit which allows for the response curve to obtain affinity.

In the new demonstration, we applied MIDAS to optimize NanoLuc catalytic activity, thus quantitative normalization of protein expression is essential because the goal is to compare photon output per molecule of enzyme. Accordingly, we implemented the strategy suggested by the reviewer by fusing NanoLuc to an orthogonal firefly luciferase, which serves as an internal reference for protein abundance. The NanoLuc/Firefly luminescence ratio therefore provides a direct measure of specific catalytic activity that is robust to well-to-well variation in expression. These results are presented in a new **Fig. 5**.

We believe this addition clarifies how expression variability can be handled in MIDAS and provides practical guidance for readers who wish to adapt the method to their own systems.

5. Generality of the Method:

The study would be strengthened by demonstrating MIDAS on a second target protein or biosensor, even at a smaller scale, to establish its broader applicability. If such demonstrations are not feasible, the authors should clearly articulate MIDAS's current limitations. For example, MIDAS requires robust expression of PCR constructs and compatible optical assays, and its effectiveness depends on prior knowledge of functional sites for targeted mutagenesis. In cases lacking structural or functional information, alternative approaches such as random mutagenesis might be more suitable.

We agree with the reviewer that generalizability is important. To directly demonstrate generalizability beyond biosensors, we added a second application: deep mutational scanning and optimization of NanoLuc luciferase activity toward three substrates using the MIDAS-MM protocol. These results are presented in **Figures 5 and 6** and show that MIDAS can improve enzyme catalysis, characterize mutational tolerance, and identify substrate-specific variants. This establishes that MIDAS generalizes to distinct protein classes and assay types (biosensors and enzymes), beyond a single target, and can be used for structure-function analyses, not only protein engineering.

Regarding applicability vs. random mutagenesis, we added to the discussion: “Random mutagenesis and ultra-high-throughput pooled screening approaches are still useful for identifying functional sites in proteins in a relatively unbiased manner, if an assay compatible with pooled screening exists. However, MIDAS could be used to mutate every residue to every possible side chain in smaller proteins, and is especially useful for exploring specific sequence regions quickly, accurately, and cost-effectively.”

Minor Points

- The acronym MIDAS is defined both as "Microbial-Independent" and "Microbe-Independent." The authors should choose one and use it consistently throughout the manuscript.

We thank the reviewer for catching this. We have carefully revised the manuscript to ensure consistent use of "Microbe-Independent" throughout the text.

- The meaning of " $\Delta C3$ " in reference to the SmBiT component should be clarified.

We now write "SmBiT ($\Delta C3$), a variant with a three-residue truncation at the C-terminus".

- Figure S3f: The K_D value is incorrectly listed as 898 mM; it should be 898 μM .

We thank the reviewer for catching this. We corrected the typo in the revised figures.

- Figure 2b: Since variants with C, E, Q, T, and V at position 560 all outperform alanine, what is the rationale for using A in the final QAE variant?

We thank the reviewer for this thoughtful question. **Figure 2b** (now **Figure 3b**) represents the results of saturation mutagenesis of 16 sites, in which each site was varied independently while other positions were held constant. Although several residues, including C, E, Q, T, and V, outperformed alanine in that context, these results do not necessarily predict optimal performance in combination with other substitutions.

In **Figure 3d**, we performed combinatorial mutagenesis across site 560 and 610, through this screen, we identified QAE as one of the top-performing variants, with the highest affinity and highest brightness at 10–100 μM ACh (**Fig. 3d**, **Fig. 3e**, **Fig. S5b**). The result highlights the non-additive nature of mutational effects. While Ala at 560 may not be the top performer on its own, its presence contributes positively in the context of the QAE background.

- Units: Replace all instances of "uM" with the correct symbol " μM ."

We have made this correction throughout the text.

- Figures S3f and S3g: If multiple neurotransmitters were tested, the x-axis label should be "Neurotransmitter concentration," not "ACh concentration."

We have changed "ACh concentration" to "concentration" in the revised **Figures S3e** and **S7e**.

- Figure S4b: Mutation A610E is referenced in the text, but the figure label shows A610?. This should be corrected for consistency.

We have corrected the typo in the revised **Figure S4b**.

In summary, this paper provides a useful demonstration of direct protein optimization in mammalian cells using PCR-assembled constructs. However, I suggest the authors better contextualize MIDAS in light of their prior work, improve the biological relevance of in vivo validation, and provide a clearer discussion of method scalability and generalizability.

We greatly appreciate the thoughtful insights and the questions raised which have allowed us to significantly strengthen the manuscript.

26th Feb 2026

Manuscript Number: MSB-2025-13105R

Title: Fast analysis and engineering of protein function by microbe-independent deep assembly and screening

Dear Prof Lin,

Thank you for the submission of your revised manuscript to Molecular Systems Biology. We have now received the enclosed reports from the referees that were asked to re-assess it. As you will see the reviewers are now globally supportive and I am pleased to inform you that we will be able to accept your manuscript pending the following final amendments:

- 1) The author list in the manuscript and submission system should match. Currently author Daesun Song is missing in the submission system. In addition there are name discrepancies that need to be rectified: Lan Liu vs Lan Xiang Liu, Thomas Kirkland vs Thomas A. Kirkland
- 2) In the main manuscript file, please include keywords to max. 5.
- 3) Please rename "Competing Interest" to "Disclosure and competing interests statement". Employment in a biotech company should also be included in the statement (Promega Corporation?). We updated our journal's competing interests policy in January 2022 and request authors to consider both actual and perceived competing interests. Please review the policy <https://link.springer.com/partners/embo-press/editorial-policies#Competing%20interest%20disclosures> and update your competing interests if necessary.
- 4) Author contributions: Please remove it from the manuscript and specify author contributions in our submission system. CRediT has replaced the traditional author contributions section because it offers a systematic machine-readable author contributions format that allows for more effective research assessment. You are encouraged to use the free text boxes beneath each contributing author's name to add specific details on the author's contribution. More information is available in our guide to authors: <https://www.embopress.org/page/journal/17574684/authorguide#authorshipguidelines>
- 5) References: Please correct the reference citation in the reference list to be alphabetical (not numerical). Where there are more than 10 authors on a paper, only the first 10 should be listed, followed by "et al.". Please check "Author Guidelines" for more information: <https://link.springer.com/journal/44320/submission-guidelines#cms-Reference-guidelines>
- 6) Our journal encourages inclusion of *data citations in the reference list* to directly cite datasets that were re-used and obtained from public databases. Data citations in the article text are distinct from normal bibliographical citations and should directly link to the database records from which the data can be accessed. In the main text, data citations are formatted as follows: "Data ref: Smith et al, 2001" or "Data ref: NCBI Sequence Read Archive PRJNA342805, 2017". In the Reference list, data citations must be labeled with "[DATASET]". A data reference must provide the database name, accession number/identifiers and a resolvable link to the landing page from which the data can be accessed at the end of the reference. Further instructions are available at .
- 7) In the Methods, please take care of the following:
 - Please acknowledge BioRender at the end of the Methods section.
 - Please include information on the mice and cultured rat neuronal cells in the Experimental Models section of the Reagents and Tools Table.
 - Please also be sure to include a sentence in the Methods as to whether or not the cell lines were recently authenticated and update the Author Checklist with this information and where it can be found in the manuscript.
 - Although you have indicated in the Author Checklist that a statement on whether or not blinding was done is included in the Methods, we were unable to locate this statement. Please update the manuscript to include this information in case it is missing.
- 8) Please upload the Reagents and Tools Table as a separate file choosing the file type "Reagent Table".
- 9) Please place individual sections of the manuscript in the following order: Title page - Abstract & Keywords - Introduction - Results - Discussion - Methods - Data Availability - Acknowledgements - Disclosure and Competing Interests Statement - References - Figure Legends - Expanded View Figure Legends.
- 10) For the figures and figure legends, please take care of the following:
 - Please remove all figures from main manuscript file and leave only main figure legends placed after the references.
 - Please make sure to update the callouts of all figures in the main manuscript text. Currently figure callouts are missing for Figure 1d and Figure 2b.
 - There are labels missing for some of the figure panels: Figure 2, panel F is not labeled in the figure itself, and Figure 3, panel F missing in figure caption.
 - Please define the annotated p values ****/**/*/* as well as provide the exact p-values for the same in the legend of figure 5D as appropriate.
 - Please note that the exact p values are not provided in the legends of figures 2B-E; 4C, EV1 B, EV2 D, EV3 B, D
 - Please indicate the statistical test used for data analysis in the legend of figure 5D
 - Please note that information related to n is missing in the legends of figures 5D, EV7 E,
 - Please note that the error bars are not defined in the legends of figures 5D, EV5 B
 - Please note that the measure of center for the error bars needs to be defined in the legends of figures 2B-E; EV2 D

11) In a routine figure check, we note that the picture of the Ach mice in Figure EV9A look quite similar between pre and post conditions. Please check to be sure that this is not the same mouse image with different heat maps.

12) Please remove Table EV1 from the manuscript and upload it separately.

13) Please ensure that all funding sources are entered into the manuscript submission system. Currently Stanford Bio-X Interdisciplinary Initiatives Program Seed Grant and Stanford Bio-X PhD Fellowship are missing.

14) Synopsis:

- Synopsis image: Please provide a graphic that summarises the main findings of the manuscript on a glance and upload it as a high-resolution jpeg file 550 pixels wide x (300-600) pixels high.

- Synopsis text: Please provide a separate word document including a short standfirst (maximum of 300 characters, including spaces) and up to 5 bullet points to summarise the key NEW findings. They should be designed to be complementary to the abstract - i.e. not repeat the same text. We encourage inclusion of key acronyms and quantitative information (maximum of 30 words / bullet point). Please use the passive voice.

15) As part of the EMBO Publications transparent editorial process initiative (see our policy here:

https://www.embopress.org/transparent-process#Review_Process), Molecular Systems Biology will publish online a Peer Review File (PRF) to accompany accepted manuscripts. This file will be published in conjunction with your paper and will include the anonymous referee reports, your point-by-point response and all pertinent correspondence relating to the manuscript. Let us know whether you agree with the publication of the PRF and as here, if you want to remove or not any figures from it prior to publication. Please note that the Authors checklist will be published at the end of the PRF.

16) After your paper is published, we may promote it on social media. If you have any handles or hashtags for Bluesky you would like included, please let us know.

17) Please provide a point-by-point letter INCLUDING my comments and your detailed responses (as Word file).

I look forward to reading a new revised version of your manuscript as soon as possible.

Yours sincerely,

Poonam Bheda, PhD
Scientific Editor
Molecular Systems Biology

Reviewer #1:

The authors have addressed my questions in a satisfying manner.

Reviewer #2:

The authors have meticulously revised the manuscript in response to my comments. I suggest the authors to make appropriate adjustments to the paper format according to MSB's publication guidelines. I think this manuscript is well-suited for publication in Molecular Systems Biology now.

Reviewer #3:

I have no additional comments. The authors have addressed my concerns.

Dear Dr. Bheda:

Thank you very much for your Decision Letter of February 26 accepting our manuscript "**Fast analysis and engineering of protein function by microbe-independent deep assembly and screening**" (MSB-2025-13105R"). We are elated to be publishing our paper in *Molecular Systems Biology*. We also greatly appreciate your careful reading of the manuscript that noticed some missing information, and your precise instructions for making corrections.

We are now submitting a revised version of the manuscript with the suggested changes in red. In addition, as instructed, we are including responses to each of the comments:

- 1) The author list in the manuscript and submission system should match. Currently author Daesun Song is missing in the submission system. In addition there are name discrepancies that need to be rectified: Lan Liu vs Lan Xiang Liu, Thomas Kirkland vs Thomas A. Kirkland.
 - We have made the corrections.
- 2) In the main manuscript file, please include keywords to max. 5.
 - Done. We have added 5 keywords.
- 3) Please rename "Competing Interest" to "Disclosure and competing interests statement". Employment in a biotech company should also be included in the statement (Promega Corporation?). We updated our journal's competing interests policy in January 2022 and request authors to consider both actual and perceived competing interests. Please review the policy (link.springer.com/partners/embopress/editorial-policies) and update your competing interests if necessary.
 - Done. We have added employer and patent information.
- 4) Author contributions: Please remove it from the manuscript and specify author contributions in our submission system. CRediT has replaced the traditional author contributions section because it offers a systematic machine-readable author contributions format that allows for more effective research assessment. You are encouraged to use the free text boxes beneath each contributing author's name to add specific details on the author's contribution. More information is available in our guide to authors: www.embopress.org/page/journal/17574684/authorguide
 - Done. We have removed the section.
- 5) References: Please correct the reference citation in the reference list to be alphabetical (not numerical). Where there are more than 10 authors on a paper, only the first 10 should be listed, followed by "et al.". Please check "Author Guidelines" for more information: <https://link.springer.com/journal/44320/submission-guidelines#cms-Reference-guidelines>
 - Done. We have changed the reference style.
- 6) Our journal encourages inclusion of *data citations in the reference list* to directly cite datasets that were re-used and obtained from public databases. Data citations in the article text are distinct from normal bibliographical citations and should directly link to the database records from which the data can be accessed. In the main text, data citations are formatted as follows: "Data ref: Smith et al, 2001" or "Data ref: NCBI Sequence Read Archive PRJNA342805, 2017". In the Reference list, data citations must be labeled with "[DATASET]". A data reference must provide the database name, accession number/identifiers and a resolvable link to the landing page from which the data can be accessed at the end of the reference. Further instructions are available at <https://www.embopress.org/page/journal/17574684/authorguide#referencesformat>.
 - We did not cite any datasets, so no changes were necessary.
- 7) In the Methods, please take care of the following:
 - Please acknowledge BioRender at the end of the Methods section.
 - We have added it into the software section of the reagent table.
 - Please include information on the mice and cultured rat neuronal cells in the Experimental Models section of the Reagents and Tools Table.
 - We have added cell and animal information to the reagent table.

- Please also be sure to include a sentence in the Methods as to whether or not the cell lines were recently authenticated and update the Author Checklist with this information and where it can be found in the manuscript.
 - We have added authentication statements in the Methods.
- Although you have indicated in the Author Checklist that a statement on whether or not blinding was done is included in the Methods, we were unable to locate this statement. Please update the manuscript to include this information in case it is missing.
 - We have added blinding statements in the Methods.
- 8) Please upload the Reagents and Tools Table as a separate file choosing the file type "Reagent Table".
 - Done; we have made the Reagents and Tools Table as a separate file.
- 9) Please place individual sections of the manuscript in the following order: Title page - Abstract & Keywords - Introduction - Results - Discussion - Methods - Data Availability - Acknowledgements - Disclosure and Competing Interests Statement - References - Figure Legends - Expanded View Figure Legends.
 - Done. We have reordered the manuscript.
- 10) For the figures and figure legends, please take care of the following:
 - Please remove all figures from main manuscript file and leave only main figure legends placed after the references.
 - Done. We have corrected this.
 - Please make sure to update the callouts of all figures in the main manuscript text. Currently figure callouts are missing for Figure 1d and Figure 2b.
 - Done. We have checked and made corrections.
 - There are labels missing for some of the figure panels: Figure 2, panel F is not labeled in the figure itself, and Figure 3, panel F missing in figure caption.
 - Done. We have checked and made corrections.
 - Please define the annotated p values ****/**/*/* as well as provide the exact p-values for the same in the legend of figure 5D as appropriate.
 - Done. We have checked and made corrections.
 - Please note that the exact p values are not provided in the legends of figures 2B-E; 4C, EV1B, EV2D, EV3B,D
 - Done. We have checked and made corrections.
 - Please indicate the statistical test used for data analysis in the legend of figure 5D.
 - Done. We have checked and made corrections.
 - Please note that information related to n is missing in the legends of figures 5D, EV7E.
 - Done. We have checked and made corrections.
 - Please note that the error bars are not defined in the legends of figures 5D, EV5B.
 - Done. We have checked and made corrections.
 - Please note that the measure of center for the error bars needs to be defined in the legends of figures 2B-E; EV2 D
 - Done. We have checked and made corrections.
- 11) In a routine figure check, we note that the picture of the Ach mice in Figure EV9A look quite similar between pre and post conditions. Please check to be sure that this is not the same mouse image with different heat maps.
 - We have double-checked. They are different images.
- 12) Please remove Table EV1 from the manuscript and upload it separately.
 - We have done this.
- 13) Please ensure that all funding sources are entered into the manuscript submission system. Currently Stanford Bio-X Interdisciplinary Initiatives Program Seed Grant and Stanford Bio-X PhD Fellowship are missing.
 - We have added the information into the system.
- 14) Synopsis:
 - Synopsis image: Please provide a graphic that summarises the main findings of the manuscript on a glance and upload it as a high-resolution jpeg file 550 pixels wide x (300-600) pixels high.
 - Done. We have made a graphical synopsis.

- Synopsis text: Please provide a separate word document including a short standfirst (maximum of 300 characters, including spaces) and up to 5 bullet points to summarise the key NEW findings. They should be designed to be complementary to the abstract - i.e. not repeat the same text. We encourage inclusion of key acronyms and quantitative information (maximum of 30 words / bullet point). Please use the passive voice.
 - Done. We have written a synopsis.
 - Please check your synopsis text and image before submission with your revised manuscript. Please be aware that in the proof stage minor corrections only are allowed (e.g., typos).
 - Done. We have checked the synopsis.
- 15) As part of the EMBO Publications transparent editorial process initiative (see www.embopress.org/transparent-process#Review_Process), Molecular Systems Biology will publish online a Peer Review File (PRF) to accompany accepted manuscripts. This file will be published in conjunction with your paper and will include the anonymous referee reports, your point-by-point response and all pertinent correspondence relating to the manuscript. Let us know whether you agree with the publication of the PRF and as here, if you want to remove or not any figures from it prior to publication. Please note that the Authors checklist will be published at the end of the PRF.
- We agree with the publication of the PRF.
- 16) After your paper is published, we may promote it on social media. If you have any handles or hashtags for Bluesky you would like included, please let us know.
- BlueSky: @michaelzlin.bsky.social
 - X.com: @michaellinlab
- 17) Please provide a point-by-point letter INCLUDING my comments and your detailed responses (as Word file).
- Responses are in this file.

In addition, we realized we could make the primer naming system universal across variations of MIDAS, which also would make the Methods section easier to understand. To do this, we changed the numbering system for the primers so that the outer primers are Fo and Ro and inner primers are Fi and Ri for the i-th targeted region. We have made the changes in the manuscript already and hope you agree that this is an improvement. We can change it back if you think the original numbering method was better. Finally, we added a new Figure EV1 and associated legend with alternate arrangements for the directionality of the mutagenic vs non-mutagenic internal primers. We believe this provides users with additional options; they may prefer the form in Figure 1 or the form in Figure 2, depending on whether they are performing manual or robotic pipetting. We have double-checked that the figure call-outs are correct after the insertion of this new figure.

2nd Apr 2026

Manuscript number: MSB-2025-13105RR

Title: Fast analysis and engineering of protein function by microbe-independent deep assembly and screening

Dear Prof Lin,

Congratulations on an excellent manuscript, I am pleased to inform you that your manuscript has been accepted for publication in Molecular Systems Biology. Thank you for your comprehensive response to referee concerns. It has been a pleasure to work with you to get this to the acceptance stage.

You may qualify for financial assistance for your publication charges - either via a Springer Nature fully open access agreement or an EMBO initiative. Check your eligibility: <https://link.springer.com/journal/44320/how-to-publish-with-us>

Yours sincerely,

Sincerely,

Poonam Bheda, PhD
Scientific Editor
Molecular Systems Biology

>>> Please note that it is Molecular Systems Biology policy for the transcript of the editorial process (containing referee reports and your response letter) to be published as an online supplement to each paper. If you do NOT want this, you will need to inform the Editorial Office via email immediately. More information is available here: <https://link.springer.com/partners/embo-press/editorial-policies#Peer%20review>